# Multi-task Linear Regression without Eigenvalue Lower Bounds: Adaptivity, Robustness, and Safety

**Seok-Jin Kim** [1]

## Abstract

We study the multi-task linear regression problem in the presence of contaminated tasks. We address the setting where the unknown parameters of a majority of tasks are close in the $\ell_2$-norm, while a fraction of tasks are arbitrary outliers. Existing theoretical frameworks for this problem rely heavily on the assumption that the empirical second moment of each task has a minimum eigenvalue bounded away from zero (order $\Omega(1)$). Crucially, this assumption fails in many high-dimensional scenarios, rendering prior guarantees vacuous. To overcome this limitation, we propose an estimator based on matrix-weighted norm regularization. We also introduce a relative balancedness condition, quantified by a balancedness constant, that compares each task's second moment with the average inlier geometry and relaxes the need for taskwise second-moment lower bounds. In favorable regimes with moderate balancedness, our prediction MSE bounds match the rate of Duan and Wang (2023) under substantially weaker spectral assumptions; the resulting task-overall MSE is minimax optimal up to logarithmic factors. Furthermore, we demonstrate that our estimator enjoys a safety guarantee: when the relevant balancedness constant is large or infinite, or when tasks are unrelated, the method performs no worse than independent task learning.

## 1. Introduction

Multi-task learning (MTL) aims to improve performance across multiple tasks simultaneously by exploiting their underlying relatedness (Zhang and Yang, 2018; 2021). This paradigm has achieved significant empirical success (Caru-

ana, 1997; Evgeniou and Pontil, 2004; Evgeniou et al., 2005) and has recently attracted growing theoretical interest (Asiaee et al., 2019; Wu et al., 2020; Tian et al., 2026; 2025; Huang et al., 2025). In this paper, we investigate MTL for linear and generalized linear models (GLMs) in a regime where the majority of tasks are similar, but a minority are arbitrary outliers. Crucially, neither the degree of similarity nor the fraction of outliers is known *a priori*, necessitating an adaptive learning approach. While outliers pose substantial theoretical challenges, robust estimators have recently been developed to address them (Tian et al., 2026; Duan and Wang, 2023; Tian et al., 2025).

**Problem Setup.** We consider a system of $m$ tasks. For the $j$-th task, we observe a dataset $\mathcal{D}_j := \{(x_{ji}, y_{ji})\}_{i=1}^{n_j}$, with covariates $x_{ji} \in \mathbb{R}^d$ and responses $y_{ji} \in \mathbb{R}$, generated by a linear model $\mathbb{E}[y_{ji} \mid x_{ji}] = x_{ji}^\top \theta_j^\star$, where $\theta_j^\star \in \mathbb{R}^d$ is the unknown parameter for task $j \in [m]$. Our analysis naturally extends to generalized linear models (GLMs). The objective is to estimate each $\theta_j^\star$ with small prediction mean squared error (MSE), also referred to as prediction risk. For each task $j$, let $\Sigma_j := \frac{1}{n_j} \sum_{i=1}^{n_j} x_{ji} x_{ji}^\top$ denote the *empirical second moment matrix*. We primarily focus on the balanced sample size case where $n_j = n$ for all $j$, deferring the heterogeneous setting to the appendix.

**Task Relatedness with Outliers.** We adopt the contamination model introduced by Duan and Wang (2023). We assume there exists a subset of "inlier" tasks $\mathbf{S} \subseteq [m]$ whose parameters $\{\theta_j^\star\}_{j \in \mathbf{S}}$ lie within an $\ell_2$-ball of radius $\delta$. The remaining tasks in $\mathbf{S}^c$ are outliers with no assumed structure, comprising a fraction $\frac{|\mathbf{S}^c|}{m} < \varepsilon$ of the total tasks. The parameters $\varepsilon$ and $\delta$ are unknown to the learner. This contaminated relatedness model has also been developed in subsequent work on semiparametric multi-task inference (Bhattacharya et al., 2025). This formulation generalizes standard $\ell_2$-closeness assumptions studied in the outlier-free regime (Soare et al., 2014; Wang et al., 2021; Sessa et al., 2023). Alternative notions of relatedness, such as shared low-rank representations, have been explored elsewhere (Tian et al., 2025); see Section 1.2 for a detailed review.

---

[1]Department of Industrial Engineering and Operations Research, Columbia University, New York, NY, United States. Correspondence to: Seok-Jin Kim <seok-jin.kim@columbia.edu>.

*Proceedings of the 43rd International Conference on Machine Learning*, Seoul, South Korea. PMLR 306, 2026. Copyright 2026 by the author(s).

**Relaxing Lower Boundedness of Second Moments.** A significant limitation of existing outlier-robust analyses (e.g., Duan and Wang, 2023; Tian et al., 2025) is the reliance on well-conditioned covariates. Specifically, these works typically assume the existence of constants $\rho = \Omega(1)$ and $L = \mathcal{O}(1)$ such that $\rho \mathbf{I}_d \preceq \boldsymbol{\Sigma}_j \preceq L\mathbf{I}_d$ for all $j \in [m]$. Under this *Lower Boundedness of Second Moments* (LBSM) condition, the analysis of Duan and Wang (2023) yields the following bounds for their estimator $\{\hat{\theta}_j\}_{j=1}^m$:

$$
\begin{aligned}
&\|\hat{\theta}_j - \theta_j^\star\|_2^2 \\
&\lesssim \frac{1}{\rho^2} \left( \frac{d}{mn} + \min\left( \frac{L^4}{\rho^2}\delta^2, \frac{d}{n} \right) + \varepsilon^2 \frac{d}{n} \right), \quad j \in \mathbf{S}, \quad (1) \\
&\|\hat{\theta}_j - \theta_j^\star\|_2^2 \lesssim \frac{1}{\rho^2}\frac{d}{n}, \quad j \in \mathbf{S}^c.
\end{aligned}
$$

While the upper bound $L = \mathcal{O}(1)$ is mild (Vershynin, 2018; Wainwright, 2019), the lower bound requirement $\rho = \Omega(1)$ is restrictive and often unrealistic in high-dimensional settings or for distributions with decaying spectra. For instance, covariates drawn uniformly from a sphere satisfy $\rho \asymp 1/d$, causing the bound in (1) to degrade significantly. This concern is not limited to passive data collection: in online learning settings such as linear bandits, the covariates or actions are selected adaptively by an algorithm, and the resulting taskwise empirical second moments need not satisfy uniform spectral regularity. From the standpoint of prediction MSE, it remains an open question whether this dependence on $\rho$ is fundamental or an artifact of the Euclidean-parameter analysis.

Single-task linear regression suggests that the LBSM requirement should not be necessary for prediction. Indeed, a fundamental baseline is *Independent-Task Learning* (ITL), which estimates $\theta_j^\star$ using only $\mathcal{D}_j$ and attains in-sample prediction MSE $\tilde{\mathcal{O}}(d/n)$ regardless of the condition number of $\boldsymbol{\Sigma}_j$ (Wainwright, 2019; Bach, 2024). This raises the central question of this work: *can multi-task linear regression exploit relatedness even under rapidly decaying spectra, while still preserving the ITL rate whenever transfer is unhelpful?* We answer this question affirmatively by replacing the standard $\ell_2$ regularization used in Duan and Wang (2023) (see Eq. (2)) with a *matrix-weighted regularization* induced by the norms $\|\cdot\|_{\boldsymbol{\Sigma}_j}$. As detailed in Section 3, this geometry relaxes LBSM: under the balancedness assumption introduced in Section 4, it enables beneficial transfer, while preserving the $\tilde{\mathcal{O}}(d/n)$ independent-task rate when transfer is unhelpful.

### 1.1. Overview of Results

Our approach replaces LBSM with a *balancedness assumption*, quantified by a balancedness constant $B$ that measures the similarity of second moment matrices across tasks rather than enforcing a uniform lower bound. Under LBSM,

$B \lesssim 1$, but our condition accommodates significantly broader scenarios, including rapidly decaying spectra. Under this relaxed assumption, we derive adaptive in-sample MSE guarantees and, once $n$ exceeds a mild threshold, corresponding population MSE bounds. Our results are summarized informally as follows:

**Theorem 1** (Informal). *Consider the multi-task linear regression problem with outliers (Definition 1). In favorable regimes with a moderate balancedness constant $B$, our estimator attains the same $m, n, \delta, \varepsilon$ MSE dependence as the robust MTL guarantee of Duan and Wang (2023), under substantially relaxed assumptions on the covariate eigenspectrum; moreover, the task-overall MSE is minimax optimal in these parameters, up to logarithmic and balancedness-dependent factors. Furthermore, the estimator exhibits:*

1. ***Robustness:** Efficient estimation in the presence of an unknown fraction $\varepsilon$ of outliers.*

2. ***Adaptivity:** Automatic utilization of task similarity under balancedness, without prior knowledge of $\delta$ or $\varepsilon$.*

3. ***Safety:** An MSE bound of $\tilde{\mathcal{O}}(d/n)$ holds even when no useful moderate balancedness constant exists or tasks are unrelated, matching the independent-task baseline.*

Thus, when $B$ is moderate, the estimator exploits task relatedness and recovers the robust MTL rate of Duan and Wang (2023) under weaker spectral assumptions. When $B$ is large or the relatedness structure is unfavorable, the same procedure falls back to the safe $\tilde{\mathcal{O}}(d/n)$ rate.

### 1.2. Background

Multi-task learning has been surveyed comprehensively in Zhang and Yang (2021). Within the context of linear models, MTL is a mature field (Evgeniou and Pontil, 2004; Evgeniou et al., 2005; Asiaee et al., 2019; Cai et al., 2022; Li et al., 2022). Classical approaches range from shared covariate representations (Breiman and Friedman, 1997; Evgeniou and Pontil, 2004; Evgeniou et al., 2005) to Stein-type shrinkage estimation (Chen et al., 2015). Theoretical characterizations have been established in simplified settings, such as two-task linear systems or single-hidden-layer networks (Mousavi Kalan et al., 2020; Wu et al., 2020).

Regarding task relatedness structures, sparsity-based methods (without outliers) are discussed in Cai et al. (2022); Li et al. (2022); Huang et al. (2025); Xu and Bastani (2025), while low-rank shared representations are examined in Hu et al. (2021); Du et al. (2023); Tian et al. (2025). Robust MTL has also been studied with irrelevant or outlier tasks in both linear and latent-variable settings (Chen et al., 2011; Tian et al., 2026). Our work focuses on $\ell_2$-closeness (Definition 1). The outlier-free variant was investigated by Soare et al. (2014); Wang et al. (2021); Sessa et al. (2023), while

the contaminated setting was formalized by Duan and Wang (2023) and later pursued in related semiparametric multi-task work (Bhattacharya et al., 2025).

Naive baselines include ITL and Data Pooling (DP). Both are suboptimal in our regime: ITL ignores task relatedness, while DP is sensitive to negative transfer from outliers (Duan and Wang, 2023). To address this, Duan and Wang (2023) proposed an adaptive robust method (ARMUL) minimizing the objective:

$$\mathcal{L}(\theta_1, \ldots, \theta_m, \beta) = \sum_{j \in [m]} \left( f_j(\theta_j) + \lambda \|\theta_j - \beta\|_2 \right), \quad (2)$$

where $f_j$ is the task-specific loss. However, their theoretical guarantees (e.g., Eq. 1) depend inversely on taskwise curvature parameters; in linear regression, this amounts to dependence on the minimum eigenvalue of $\mathbf{\Sigma}_j$. Our work relaxes these requirements by replacing the Euclidean regularization $\|\cdot\|_2$ with a matrix-weighted penalty $\|\theta_j - \beta\|_{\mathbf{\Sigma}_j}$, while avoiding the strong-convexity requirement used in prior analyses.

### 1.3. Notation

The constants $c_1, c_2, C_1, C_2, \ldots$ may differ from line to line. We use the symbol $[n]$ as a shorthand for $\{1, 2, \ldots, n\}$. For nonnegative sequences $\{a_n\}_{n=1}^{\infty}$ and $\{b_n\}_{n=1}^{\infty}$, we write $a_n \lesssim b_n$ or $a_n = \mathcal{O}(b_n)$ if there exists a positive constant $C$ such that $a_n \leq C b_n$ for all $n$. We use $\widetilde{\mathcal{O}}$ to hide polylogarithmic factors. In addition, we write $a_n \asymp b_n$ if $a_n \lesssim b_n$ and $b_n \lesssim a_n$. For a vector $x \in \mathbb{R}^d$, $\|x\|_p$ denotes its $\ell_p$-norm. For any positive semidefinite matrix $A \in \mathbb{R}^{d \times d}$, we define the induced norm $\|x\|_A := \sqrt{x^\top A x}$. For extended-real-valued functions $f, g : \mathbb{R}^d \to \mathbb{R} \cup \{+\infty\}$, their infimal convolution is

$$(f \star g)(\beta) := \inf_{\theta \in \mathbb{R}^d} \{f(\theta) + g(\beta - \theta)\}.$$

When the parameter space is restricted to a convex set $\mathbf{C}$, the infimum is taken over $\theta \in \mathbf{C}$.

## 2. Problem Setup

We consider a multi-task learning setting with $m$ tasks. For each task $j \in [m]$, we observe a dataset $\mathcal{D}_j := \{(x_{ji}, y_{ji})\}_{i=1}^{n_j}$ consisting of $d$-dimensional covariates and scalar responses. We assume the data generation process follows a linear model:

$$y_{ji} = x_{ji}^\top \theta_j^\star + \varepsilon_{ji}, \quad i \in [n_j], \quad (3)$$

where $\theta_j^\star \in \mathbb{R}^d$ is the unknown parameter for task $j$. Conditional on the design $\mathbf{X} := \{x_{ji}\}_{j,i}$, the noise terms $\varepsilon_{ji}$ are assumed to be independent and $\sigma$-sub-Gaussian. Without loss of generality, we set the sub-Gaussian parameter $\sigma = 1$.

While our framework extends to Generalized Linear Models (GLMs) as discussed in Section 6, we focus on the linear model for the primary analysis.

For each task $j$, let $\mathbf{X}_j := (x_{j1}, \ldots, x_{jn_j})^\top \in \mathbb{R}^{n_j \times d}$ denote the design matrix and $\mathbf{Y}_j := (y_{j1}, \ldots, y_{jn_j})^\top \in \mathbb{R}^{n_j}$ the response vector. We define the empirical Gram matrix as $\mathbf{V}_j := \mathbf{X}_j^\top \mathbf{X}_j = \sum_{i=1}^{n_j} x_{ji} x_{ji}^\top$ and the *empirical second moment matrix* as $\mathbf{\Sigma}_j := \frac{1}{n_j} \mathbf{V}_j$. To simplify the theoretical exposition, we assume a balanced sample size $n_j = n$ for all $j \in [m]$ in the main text; extensions to heterogeneous sample sizes are provided in Appendix F.

**Regularity Assumptions.** Following standard high-dimensional statistics literature (Vershynin, 2018; Wainwright, 2019), we assume that the covariate norms are bounded. Specifically, we assume $\|\mathbf{\Sigma}_j\|_{\mathrm{op}} \leq c$ for a universal constant $c$, which we normalize to $c = 1$ for simplicity. For instance, in the task-wise i.i.d. setting, this empirical bound holds with high probability when $\|x_{ji}\|_2$ is almost surely bounded or sub-exponential. *Crucially, we do not assume that the minimum eigenvalue of $\mathbf{\Sigma}_j$ is bounded away from zero.*

**Task Relatedness with Outliers.** We adopt the contamination framework introduced by Duan and Wang (2023) to model task relatedness in the presence of outliers.

**Definition 1** (Task Relatedness). A set of tasks is said to be $(\varepsilon, \delta)$-related if there exist a subset of indices $\mathbf{S} \subseteq [m]$ and a shared centroid parameter $\theta^\star \in \mathbb{R}^d$ such that:

$$\max_{j \in \mathbf{S}} \|\theta_j^\star - \theta^\star\|_2 \leq \delta \quad \text{and} \quad \frac{|\mathbf{S}^c|}{m} \leq \varepsilon.$$

Here, $\mathbf{S}$ represents the set of "inlier" tasks similar to $\theta^\star$, while $\mathbf{S}^c$ contains outlier tasks with arbitrary parameters. Our algorithm is fully adaptive and requires no prior knowledge of the radius $\delta$, the outlier fraction $\varepsilon$, or the index set $\mathbf{S}$.

**Performance Measures.** We evaluate the performance of our estimators using the MSE. Given the fixed design nature of our analysis, we primarily focus on the *in-sample MSE*. For an estimator $\hat{\theta}_j$, the in-sample MSE for task $j$ is defined as:

$$\mathcal{E}_j^{\mathrm{in}}(\hat{\theta}_j) := \|\hat{\theta}_j - \theta_j^\star\|_{\mathbf{\Sigma}_j}^2 = \frac{1}{n_j} \sum_{i=1}^{n_j} \left| x_{ji}^\top (\hat{\theta}_j - \theta_j^\star) \right|^2. \quad (4)$$

In the specific setting where covariates are *task-wise i.i.d.* (i.e., $x_{ji} \sim \mathcal{P}_j$ independently), we also consider the *population MSE*:

$$\mathcal{E}_j(\hat{\theta}_j) := \mathbb{E}_{x \sim \mathcal{P}_j} \left[ \left| x^\top (\hat{\theta}_j - \theta_j^\star) \right|^2 \right] = \|\hat{\theta}_j - \theta_j^\star\|_{\bar{\mathbf{\Sigma}}_j}^2, \quad (5)$$

where $\bar{\boldsymbol{\Sigma}}_j := \mathbb{E}[\boldsymbol{\Sigma}_j]$ is the population covariance matrix. This measure is standard in random design settings (Van de Geer, 2008; Bühlmann and Van De Geer, 2011). This metric differs from the squared Euclidean parameter error used in Duan and Wang (2023). Under LBSM, the two metrics are equivalent up to constants. Under general spectra without LBSM, however, Euclidean parameter error may fail to concentrate even when $\delta = \varepsilon = 0$, because weakly observed or unobserved directions are not statistically identifiable from prediction data. For this reason, prediction MSE, or risk, is widely used as a standard performance metric in regression with ill-conditioned or singular designs.

## 3. Methodologies

In this section, we present our proposed estimator. Although the main theoretical results are stated for the balanced case $n_j = n$, we present the estimator in a form that also covers heterogeneous sample sizes. We specialize to $n_j = n$ in the main analysis from Section 4 onward, and defer the full heterogeneous-sample-size guarantee to Appendix F. We define the collective parameter vector as $\Theta = (\theta_1, \ldots, \theta_m, \beta) \in (\mathbb{R}^d)^{m+1}$, where $\beta$ serves as a shared centroid parameter. For each task $j \in [m]$, let $f_j : \mathbb{R}^d \to \mathbb{R}$ be the task-specific convex loss function. For standard linear regression, we adopt the quadratic loss $f_j(\theta) = \frac{1}{2n_j}\|\mathbf{Y}_j - \mathbf{X}_j\theta\|_2^2$.

**Proposed Objective.** We propose to minimize a novel multi-task objective function that incorporates a *matrix-weighted regularization*. Given nonnegative weights $\{w_j\}_{j=1}^m$ and regularization parameters $\{\lambda_j\}_{j=1}^m$, our loss function is defined as:

$$\mathcal{L}(\Theta) := \sum_{j=1}^m w_j \left( f_j(\theta_j) + \lambda_j \|\theta_j - \beta\|_{\boldsymbol{\Sigma}_j} \right). \quad (6)$$

Here, the penalty term $\|\theta_j - \beta\|_{\boldsymbol{\Sigma}_j} := \sqrt{(\theta_j - \beta)^\top \boldsymbol{\Sigma}_j (\theta_j - \beta)}$ adapts the regularization geometry to the empirical second moment of each task. This is the primary distinction from prior work (Duan and Wang, 2023), which employs isotropic regularization (see Eq. (2)). Equivalently, $\|\theta_j - \beta\|_{\boldsymbol{\Sigma}_j} = \frac{1}{\sqrt{n_j}}\|\mathbf{X}_j(\theta_j - \beta)\|_2$, so the penalty measures disagreement in *prediction space* rather than raw parameter space. For Gaussian regression with common noise variance, this quantity is proportional to the Kullback–Leibler divergence between the task-wise predictive distributions induced by $\theta_j$ and $\beta$ on task $j$'s design. Equivalently, the analysis can be viewed as whitening by $\boldsymbol{\Sigma}_j^{1/2}$: the penalty becomes Euclidean in prediction-normalized coordinates, which is why weakly observed directions are not over-penalized. Under pairwise comparability of the task covariances, this whitening perspective would largely reduce to a common-geometry

argument; the main technical challenge here is to make it work under the weaker one-sided, average-based balancedness condition of Assumption 1. By contrast, an isotropic Euclidean penalty treats weak and well-observed directions equally and reintroduces the lower-eigenvalue dependence that we seek to avoid. Thus, by exploiting the curvature information in $\boldsymbol{\Sigma}_j$, our formulation remains effective even when the Gram matrices are ill-conditioned or singular.

Let $\mathbf{C} \subseteq \mathbb{R}^d$ denote the feasible set for the parameters $\theta_j^\star$ and $\theta^\star$. If a norm bound is known (e.g., $\|\theta^\star\|_2 \leq \xi$), we set $\mathbf{C} = \mathbf{B}(0, \xi)$; otherwise, we set $\mathbf{C} = \mathbb{R}^d$. Our estimator is obtained by solving the following joint convex optimization problem:

$$(\hat{\theta}_1, \ldots, \hat{\theta}_m, \hat{\beta}) \in \underset{\Theta \in \mathbf{C}^{m+1}}{\arg\min} \mathcal{L}(\theta_1, \ldots, \theta_m, \beta). \quad (7)$$

The complete procedure is outlined in Algorithm 1.

---

**Algorithm 1** `MTLR`: Matrix-Weighted Multi-task Linear Regression

---

1: **Input:** Datasets $\{\mathcal{D}_j\}_{j=1}^m$, task weights $\{w_j\}_{j=1}^m$, regularization parameters $\{\lambda_j\}_{j=1}^m$.
2: **Step 1:** Compute the empirical second moment matrices $\boldsymbol{\Sigma}_j \leftarrow \frac{1}{n_j}\mathbf{X}_j^\top \mathbf{X}_j$, for all $j \in [m]$.
3: **Step 2:** Solve the joint convex minimization problem:

$$(\hat{\theta}_1, \ldots, \hat{\theta}_m, \hat{\beta})$$
$$\leftarrow \underset{\Theta \in \mathbf{C}^{m+1}}{\arg\min} \sum_{j=1}^m w_j \left( f_j(\theta_j) + \lambda_j \|\theta_j - \beta\|_{\boldsymbol{\Sigma}_j} \right).$$

4: **Return:** The estimated task parameters $\{\hat{\theta}_j\}_{j=1}^m$.

---

**Parameter Selection.** As justified by our theoretical analysis (Section 5), we recommend scaling the regularization parameter as $\lambda_j \asymp \sqrt{d/n_j}$. This prescription uses only the observed dimension $d$ and sample sizes $n_j$; the remaining scalar multiplier can be fixed for a target confidence level or tuned by cross-validation, without knowing $\varepsilon$, $\delta$, or $\mathbf{S}$. Further guidance on hyperparameter tuning is provided in the experimental section.

**Specific Instantiations.** We detail the configuration of our loss function for different settings:

- **Linear Model (Equal Sample Size):** In the standard setting where $n_j = n$, we set $w_j = 1$ and $\lambda_j = \lambda$ for all $j$. The objective simplifies to:

$$\mathcal{L}(\Theta) = \sum_{j=1}^m \left( \frac{1}{2n}\|\mathbf{Y}_j - \mathbf{X}_j\theta_j\|_2^2 + \lambda\|\theta_j - \beta\|_{\boldsymbol{\Sigma}_j} \right).$$

- **GLM (Equal Sample Size):** For GLMs with equal sample sizes, we again set $w_j = 1$ and $\lambda_j = \lambda$ for all $j$, and define the task loss as the negative log-likelihood:

$$f_j(\theta) := \frac{1}{n} \sum_{i=1}^{n} \left( \psi(x_{ji}^\top \theta) - y_{ji} x_{ji}^\top \theta \right),$$

  where $\psi$ is the cumulant generating function (e.g., $\psi(t) = \log(1 + e^t)$ for logistic regression). See Section 6 for details.

- **General Sample Size:** For heterogeneous sample sizes, we balance the terms by setting $w_j = n_j$ and scaling $\lambda_j = \lambda / \sqrt{n_j}$. The resulting weighted loss for linear model is:

$$\mathcal{L}(\Theta) = \sum_{j=1}^{m} n_j \left( \frac{1}{2n_j} \|\mathbf{Y}_j - \mathbf{X}_j \theta_j\|_2^2 + \lambda_j \|\theta_j - \beta\|_{\mathbf{\Sigma}_j} \right).$$

  A detailed derivation is provided in Appendix F.

**Implementation via Reparameterization.** Since the objective (6) is jointly convex whenever the task losses $f_j$ and the feasible set $\mathbf{C}$ are convex, it can be solved efficiently. For practical implementation, it is often convenient to reparameterize the problem using deviations $v_j := \theta_j - \beta$. We optimize over $(\beta, v_1, \ldots, v_m)$ by solving:

$$\min_{\beta, \{v_j\}} \sum_{j=1}^{m} w_j \left( f_j(\beta + v_j) + \lambda_j \|v_j\|_{\mathbf{\Sigma}_j} \right).$$

The exact objective is nonsmooth only at zero-seminorm deviations, and is amenable to standard convex optimization algorithms such as proximal gradient descent or block coordinate descent. In our numerical experiments, we use L-BFGS-B on this reparameterized objective with a tiny numerical floor in the norm-gradient computation. For the problem sizes considered in our experiments, this implementation converged reliably and added only modest optimization overhead relative to the baseline methods.

## 4. Our Relaxed Assumption: Balancedness

We now introduce our core assumption regarding the spectral properties of the tasks. Unlike prior works that impose absolute bounds on eigenvalues, we propose a relative condition measuring the *balancedness* of information across tasks. From this section through the main linear and GLM results, we maintain the balanced-sample-size convention $n_j = n$; the heterogeneous-sample-size extension is given in Appendix F.

For any subset of tasks $\mathbf{I} \subseteq [m]$, let $\mathbf{V}_\mathbf{I} := \sum_{j \in \mathbf{I}} \mathbf{V}_j$ denote the aggregate Gram matrix, and define the average empirical

second moment matrix as:

$$\mathbf{\Sigma}_\mathbf{I} := \frac{1}{n|\mathbf{I}|} \mathbf{V}_\mathbf{I} = \frac{1}{|\mathbf{I}|} \sum_{j \in \mathbf{I}} \mathbf{\Sigma}_j.$$

$\mathbf{\Sigma}_\mathbf{I}$ captures the aggregate covariate geometry within the task cluster $\mathbf{I}$. Our assumption posits that the second moment of any individual task is not excessively disparate compared to the aggregate geometry of the similar tasks $\mathbf{S}$.

**Assumption 1** (Balancedness). There exists a constant $B > 0$ such that the following spectral inequality holds for all $j \in [m]$:

$$\mathbf{\Sigma}_j \preceq B \cdot \mathbf{\Sigma}_\mathbf{S}. \tag{8}$$

We refer to $B$ as the *balancedness constant*.

Strictly speaking, if extended values are allowed, the smallest such scalar always exists in $[1, \infty]$: when no finite scalar satisfies the Loewner inequality, we write $B = \infty$. Thus, throughout the paper, statements such as "favorable balancedness" mean that this extended balancedness constant is finite and moderate, not merely that a formal value of $B$ can be assigned. This distinction matters because there are examples with essentially disjoint informative directions where shared estimation offers little or no improvement over ITL; see Remark 1.

The role of $B$ is one-sided. It does not measure eigendecay itself, and it does not exclude zero eigenvalues: fast eigendecay and exact rank deficiency are both compatible with finite $B$. Nor is the point of the theory to claim optimal dependence on arbitrary $B$. Instead, $B$ quantifies how favorable the transfer geometry is once LBSM has been removed. The main theorem remains informative even when $B$ is large: the transfer terms weaken, but the safety statement still matches ITL.

Crucially, Assumption 1 is strictly weaker than the LBSM condition ($\mathbf{\Sigma}_j \succeq \rho \mathbf{I}$) assumed in Duan and Wang (2023). We highlight several key properties:

1. **Upper Bound Only:** We only impose an *upper bound* on $\mathbf{\Sigma}_j$. This allows $\mathbf{\Sigma}_j$ to be rank-deficient or have a rapidly decaying spectrum (i.e., zero or near-zero eigenvalues are permitted).

2. **Safety First:** The algorithm does not need to know $B$. Even if $B$ is large or infinite, Theorem 2 still yields the independent-task rate $\tilde{\mathcal{O}}(d/n)$ through its universal safety part.

3. **Favorable-Regime Refinement:** When $B$ is moderate, transfer becomes provably beneficial and the sharper multi-task terms appear.

4. **Relaxed Comparability:** It is weaker than pairwise comparability ($\Sigma_j \preceq B\Sigma_k$), as it compares an individual task to the *average* of many tasks. Pairwise comparability is implied by LBSM with an upper covariance bound and often appears in transfer-learning analyses through bounded density-ratio assumptions; our condition is one-sided and average-based.

*Remark* 1 (Why finite $B$ matters). A finite balancedness constant is not merely a technical artifact. If the inlier tasks are concentrated on essentially disjoint subspaces, then one can have $B \asymp |\mathbf{S}|$ while the average covariance $\Sigma_{\mathbf{S}}$ contains no direction that is uniformly informative across tasks. In such a regime, shared estimation offers little or no gain over ITL. Thus, some form of balancedness is genuinely necessary for nontrivial transfer, even though the safety part of our guarantee continues to hold without it.

**Connection to Covariate Shift in Linear Models.** The balancedness assumption has a similar one-sided flavor to coverage assumptions in linear-model transfer under covariate shift. There, source-to-target transfer is typically controlled by a Loewner comparison between the second moment matrices of the training and evaluation distributions: the directions that matter for the target risk must be sufficiently covered by the source design, while no uniform lower bound on every target covariance direction is required (Wang, 2026). Our condition plays the same role in the multi-task contaminated setting. The aggregate inlier covariance $\Sigma_{\mathbf{S}}$ acts as the shared source of information, and the requirement $\Sigma_j \preceq B\Sigma_{\mathbf{S}}$ says that each task's prediction geometry is covered by that aggregate inlier geometry up to the balancedness constant $B$. Thus the assumption is not a disguised eigenvalue lower bound; it is a one-sided coverage condition that identifies when information can be transferred safely from the good tasks while still permitting singular or highly anisotropic designs.

**Empirical Diagnostic.** Because Assumption 1 is stated in terms of empirical second moments, it also suggests a simple diagnostic:

$$B_{\text{emp}} := \max_{j \in [m]} \lambda_{\max}\left(\Sigma_{[m]}^{\dagger/2} \Sigma_j \Sigma_{[m]}^{\dagger/2}\right),$$

$$\Sigma_{[m]} := \frac{1}{m} \sum_{k=1}^{m} \Sigma_k.$$

When the outlier fraction is moderate, $\Sigma_{[m]}$ is a practical proxy for the unknown inlier average $\Sigma_{\mathbf{S}}$, so $B_{\text{emp}}$ provides a rough estimate of how favorable the transfer geometry is. Large values of $B_{\text{emp}}$ indicate that one or a few tasks have covariate directions poorly represented by the task average. This diagnostic is not used by the estimator, but it can help interpret a dataset or screen a small number of tasks that make transfer geometry unfavorable.

## 4.1. Examples of Assumption 1

We illustrate the generality of Assumption 1 through the following examples, covering both well-conditioned and degenerate regimes.

**Example 1 (LBSM Regime).** Suppose the standard eigenvalue bounds hold: $\rho\mathbf{I}_d \preceq \Sigma_j \preceq L\mathbf{I}_d$ for all $j$. Then $\Sigma_j \preceq L\mathbf{I}_d = \frac{L}{\rho}(\rho\mathbf{I}_d) \preceq \frac{L}{\rho}\Sigma_{\mathbf{S}}$. Thus, Assumption 1 holds with $B = L/\rho$. Our analysis therefore replaces direct lower-eigenvalue dependence by the relative constant $B$, which equals $L/\rho$ under this classical well-conditioned regime but remains meaningful far beyond it.

**Example 2 (Pairwise Comparable).** If $\Sigma_j \preceq B'\Sigma_k$ for all pairs $j, k$, then summing over $k \in \mathbf{S}$ and normalizing implies $\Sigma_j \preceq B'\Sigma_{\mathbf{S}}$. Thus, the assumption holds with $B = B'$.

**Example 3 (Degenerate/Low-Rank Covariates).** Consider a degenerate setting where $\Sigma_j = \mathbf{I}_d$ for a subset $\mathbf{I} \subset [m]$ and $\Sigma_j = \mathbf{0}$ otherwise. Provided that the inliers overlap with the informative tasks (i.e., $\mathbf{S} \cap \mathbf{I} \neq \emptyset$), a direct calculation shows that Assumption 1 holds with the finite constant $B = \frac{|\mathbf{S}|}{|\mathbf{S} \cap \mathbf{I}|}$.

## 4.2. Balancedness for Task-wise i.i.d. Covariates

While our analysis primarily relies on the empirical quantities $\Sigma_j$, it is instructive to relate Assumption 1 to population properties when covariates are stochastic. Consider the setting where $x_{ji} \overset{\text{i.i.d.}}{\sim} \mathcal{P}_j$ for each task $j$, and let $\bar{\Sigma}_j := \mathbb{E}_{x \sim \mathcal{P}_j}[xx^\top]$.

We define the *comparability measure* $\nu_j \geq 1$ as the smallest constant satisfying:

$$\nu_j^{-1}\Sigma_j \preceq \bar{\Sigma}_j \preceq \nu_j\Sigma_j. \tag{9}$$

This constant captures the concentration of the empirical covariance around its mean. Standard matrix concentration theory guarantees that $\nu_j \approx 1$ with high probability once the sample size $n$ satisfies mild conditions.

**Sample Complexity for Sub-Exponential Covariates.** If $x_{ji}$ follows a strongly sub-Gaussian or sub-exponential distribution with parameter $K = \mathcal{O}(1)$, then by Theorem 4.7.1 in Vershynin (2018); Ding and Zheng (2024), we typically have $\nu_j \lesssim 1$ once $n = \tilde{\Omega}(d)$. Standard distributions such as Gaussian and Uniform satisfy this condition. Specifically, if the coordinates of $\bar{\Sigma}_j^{-1/2}x_{ji}$ have $\mathcal{O}(1)$ ninth moments, or if $\bar{\Sigma}_j^{-1/2}x_{ji}$ itself satisfies a $\tilde{\mathcal{O}}(1)$ high-probability $\ell_2$-norm bound, then a sample complexity of $\tilde{\mathcal{O}}(d)$ suffices (Oliveira, 2016). We provide a more detailed discussion regarding general sub-exponential covariates in Section H and Lemma 7.

Even for general sub-exponential covariates, $\nu_j = \tilde{\mathcal{O}}(1)$ holds provided that $n$ satisfies a mild sample complexity requirement. Furthermore, if $\bar{\Sigma}_j$ satisfies the LBSM condition, standard matrix concentration results imply $\nu_j = \mathcal{O}(1)$ as long as $n = \tilde{\Omega}(d)$ (Vershynin, 2018).

Using this measure, we can reformulate the balancedness assumption in terms of population quantities:

*Remark* 2 (Population Balancedness). Define the average population covariance $\bar{\Sigma}_{\mathbf{S}} := \frac{1}{|\mathbf{S}|} \sum_{j \in \mathbf{S}} \bar{\Sigma}_j$. If the population covariances satisfy $\bar{\Sigma}_j \preceq \bar{B}\bar{\Sigma}_{\mathbf{S}}$, then the empirical Assumption 1 holds with $B \lesssim (\max_j \nu_j)^2 \bar{B}$.

# 5. Results for Linear Model

In this section, we derive theoretical guarantees for the linear model. Unlike prior analyses, our guarantees depend on the relaxed balancedness constant $B$ rather than the minimum eigenvalue of $\Sigma_j$. They should be read in two layers: a universal safety bound, and a sharper transfer bound when the geometry is favorable.

We fix a failure probability $\kappa \in (0, 1)$ and define $\zeta := \log(16m/\kappa)$. Throughout this section, we maintain the balanced-sample-size convention $n_j = n$; the general case is treated in Appendix F.

## 5.1. In-sample MSE Bound

We analyze Algorithm 1 with the regularization parameter scaled as $\lambda_j = q\sqrt{d\zeta/n}$ for a sufficiently large universal constant $q \gtrsim 1$. The following theorem establishes the in-sample performance.

**Theorem 2** (In-sample MSE). *Suppose the task relatedness condition (Definition 1) holds. With probability at least $1 - \kappa$, the estimator $\hat{\theta}_j$ obtained by Algorithm 1 with $\mathbf{C} = \mathbb{R}^d$ satisfies the following bounds simultaneously for all $j \in [m]$:*

- *Safety Guarantee: Regardless of the balancedness constant $B$ or the parameters $\varepsilon$ and $\delta$,*

$$\mathcal{E}_j^{\text{in}}(\hat{\theta}_j) \lesssim q^2 \frac{d}{n}\zeta.$$

  *This matches the minimax rate for independent-task linear regression.*

- *Adaptivity and Robustness: If Assumption 1 holds with $B \lesssim \min(1/\varepsilon, m)$, then for all inlier tasks $j \in \mathbf{S}$:*

$$\mathcal{E}_j^{\text{in}}(\hat{\theta}_j) \lesssim \left( \frac{Bd}{mn} + \min\left(B\delta^2, q^2\frac{d}{n}\right) + q^2 B^2 \varepsilon^2 \frac{d}{n} \right)\zeta.$$

**Discussion.** Compared to Duan and Wang (2023), our result avoids dependence on the minimum eigenvalue of the second moments. Part (a) is a universal safety statement: regardless of $B$, $\varepsilon$, or $\delta$, the estimator never pays more than the independent-task rate up to logarithmic factors. Part (b) becomes active only when $B$ is moderate and tasks are related. In this favorable regime, the inlier guarantee, together with the safety bound for outlier tasks, recovers the same $m, n, \delta, \varepsilon$ rate structure as the robust MTL guarantee of Duan and Wang (2023), but without imposing LBSM; the covariate-geometry dependence enters through the relative balancedness constant $B$ rather than through a taskwise lower eigenvalue. Under classical LBSM with $B = L/\rho$, this replaces the lower-eigenvalue-driven factors in the prior Euclidean-parameter analysis by the relative balancedness factor $B$, which can remain finite even when no uniform lower eigenvalue exists. Specializing back to the LBSM regime, our prediction-MSE bound has a milder $\rho$-dependence than the MSE bound obtained from the analysis of Duan and Wang (2023).

**Discussion of Optimality.** Indeed, averaging the theorem bounds over all tasks gives

$$\frac{1}{m}\sum_{j=1}^{m} \mathcal{E}_j^{\text{in}}(\hat{\theta}_j)$$
$$\lesssim \left( \frac{Bd}{mn} + \min\left(B\delta^2, q^2\frac{d}{n}\right) + q^2 B^2 \varepsilon^2 \frac{d}{n} + \varepsilon q^2 \frac{d}{n} \right)\zeta,$$

which, in favorable regimes with moderate $B$, has the same $m, n, \delta, \varepsilon$ dependence as the robust MTL benchmark of Duan and Wang (2023) when viewed through prediction risk. The matching lower-bound comparison is most naturally stated in the fixed-design setting and is already witnessed by an orthogonal-design submodel: when $\mathbf{X}_j^\top \mathbf{X}_j = n\mathbf{I}_d$ and the noise is Gaussian, the observations reduce to independent Gaussian sequence observations $n^{-1}\mathbf{X}_j^\top \mathbf{Y}_j = \theta_j^\star + N(0, \mathbf{I}_d/n)$, with $B = 1$. Specializing the minimax lower-bound construction of Duan and Wang (2023) to this submodel yields the additive lower bound

$$\frac{d}{mn} + \min\left\{\delta^2, \frac{d}{n}\right\} + \varepsilon\frac{d}{n}.$$

Since in-sample prediction MSE coincides with squared Euclidean parameter error in this orthogonal submodel, the task-overall in-sample MSE above is minimax optimal with respect to $m, n, \delta, \varepsilon$ in the favorable balanced regime, up to logarithmic and balancedness-dependent factors.

## 5.2. Population MSE Bound

We now extend our guarantees to the population MSE $\mathcal{E}_j(\hat{\theta}_j)$ (defined in (5)) under the assumption that covariates are task-wise i.i.d. Recall the definition of the comparability

constant $\nu_j$ from (9), which relates empirical and population covariances ($\bar{\boldsymbol{\Sigma}}_j \preceq \nu_j \boldsymbol{\Sigma}_j$).

**Theorem 3** (Population MSE). *In the task-wise i.i.d. setting, under the conditions of Theorem 2, the following population bounds hold with probability at least $1 - \kappa$:*

1. ***Via Comparability:*** *For all tasks,*

$$\mathcal{E}_j(\hat{\theta}_j) \ \leq \ \nu_j \cdot \mathcal{E}_j^{\mathrm{in}}(\hat{\theta}_j).$$

2. ***Via Intrinsic Dimension (Safety):*** *Let $U_j$ be a high-probability upper bound on $\|x_{ji}\|_2$ in the sense formalized in Definition 3 of Appendix H. If the parameter domain is known to be bounded, $\mathbf{C} = \mathbf{B}(0, \xi)$, and $\theta_j^\star \in \mathbf{C}$, then:*

$$\mathcal{E}_j(\hat{\theta}_j^\xi) \ \lesssim \ \mathcal{E}_j^{\mathrm{in}}(\hat{\theta}_j) + \tilde{\mathcal{O}}\big(\frac{\xi^2 U_j^2}{n}\big),$$

*where $\hat{\theta}_j^\xi$ is the $\|\cdot\|_{\boldsymbol{\Sigma}_j}$-norm projection of $\hat{\theta}_j$ onto $\mathbf{B}(0, \xi)$.*

**Discussion.** The population bound introduces an additional empirical-to-population comparability factor, $\nu_j$. When $\nu_j \lesssim 1$, which holds for many sub-Gaussian or sub-exponential designs once $n \gtrsim d$, the favorable in-sample transfer guarantee carries over directly to population MSE. Even when $\nu_j$ is large, the intrinsic-dimension bound still provides a second safety layer through $\mathcal{E}_j(\hat{\theta}_j^\xi) \lesssim \mathcal{E}_j^{\mathrm{in}}(\hat{\theta}_j) + \xi^2 U_j^2/n$. The high-probability norm parameter $U_j$ is defined precisely in Appendix H: it is a tail scale for $\|x_{ji}\|_2$. The bounded-domain condition is extra prior information on the parameter scale; it is not needed for the in-sample theorem, but it gives a population fallback when empirical-to-population comparability is unfavorable. For the linear model, the projection post-processing is only used for this fallback: since it is taken in the same $\|\cdot\|_{\boldsymbol{\Sigma}_j}$-norm as the in-sample risk and $\theta_j^\star \in \mathbf{B}(0, \xi)$, it does not increase the in-sample prediction error, i.e., $\mathcal{E}_j^{\mathrm{in}}(\hat{\theta}_j^\xi) \leq \mathcal{E}_j^{\mathrm{in}}(\hat{\theta}_j)$. Thus, when prior knowledge of the bounded parameter set $\mathbf{C}$ is available, the projected estimator is automatically no worse for in-sample MSE, while Theorem 3 provides an additional safety guarantee for population MSE. For sub-Gaussian covariates and constant $\xi$, standard norm concentration gives $U_j = \tilde{\mathcal{O}}(\sqrt{d})$, so this term still matches the ITL rate up to logarithmic factors. Moreover, whenever $\nu_j$ is moderate, or more generally whenever $U_j^2 \ll d$ because the design is effectively low-dimensional, the resulting population bound is strictly sharper than ITL. In this sense, $\nu_j$ determines when the transfer improvement lifts to population risk, while the $U_j$-based bound preserves safety even outside that regime.

# 6. Results for Generalized Linear Model

We now extend the analysis to generalized linear models (GLMs) (Van de Geer, 2008). Throughout this section, we keep the balanced-sample-size convention $n_j = n$.

For each task $j \in [m]$ and sample $i \in [n]$, the conditional distribution is given by:

$$\mathbb{P}\big[y_{ji} \mid x_{ji}\big] = \rho(y_{ji}) \exp\big(y_{ji} \cdot x_{ji}^\top \theta_j^\star - \psi(x_{ji}^\top \theta_j^\star)\big),$$

where $\psi$ is a convex function, and we define the link function as $\psi'(z) = m(z)$. This implies $\mathbb{E}[y_{ji} \mid x_{ji}] = m(x_{ji}^\top \theta_j^\star)$. We define the stochastic noise as $\varepsilon_{ji} := y_{ji} - m(x_{ji}^\top \theta_j^\star)$. We assume that the pairs $(x_{ji}, y_{ji})$ for $i, j \geq 1$ are sampled independently. We adopt standard GLM assumptions (Oh et al., 2021; Tian et al., 2025) and, for simplicity, take $\varepsilon_{ji}$ to be 1-sub-Gaussian.

**Assumption 2.** We assume that $\theta_j^\star, \theta^\star \in \mathbf{B}(0, \xi)$ for some constant $\xi > 0$ known to the learner. Furthermore, we assume there exists an interval $\mathcal{I} \subset \mathbb{R}$ such that for all tasks $j$, samples $i$, and parameters $\theta \in \mathbf{C} = \mathbf{B}(0, \xi)$, the linear predictor satisfies $x_{ji}^\top \theta \in \mathcal{I}$. On this interval, the link curvature is uniformly bounded:

$$\alpha_\ell \leq m'(z) = \psi''(z) \leq \alpha_u, \qquad \forall z \in \mathcal{I},$$

for universal constants $\alpha_\ell, \alpha_u > 0$.

**In-Sample and Population MSE.** Define the GLM in-sample prediction error by $\mathcal{R}_j^{\mathrm{in}}(\theta) := \frac{1}{n} \sum_{i=1}^n |m(x_{ji}^\top \theta_j^\star) - m(x_{ji}^\top \theta)|^2$. As in the linear case, when covariates are task-wise i.i.d. (i.e., $x_{ji} \sim \mathcal{P}_j$ independently within each task), we also define the population prediction error by $\mathcal{R}_j(\theta) := \mathbb{E}_{x_{ji} \sim \mathcal{P}_j} |m(x_{ji}^\top \theta_j^\star) - m(x_{ji}^\top \theta)|^2$. Under Assumption 2, the mean value theorem gives

$$\mathcal{R}_j^{\mathrm{in}}(\hat{\theta}_j) \lesssim \mathcal{E}_j^{\mathrm{in}}(\hat{\theta}_j), \quad \mathcal{R}_j(\hat{\theta}_j) \lesssim \mathcal{E}_j(\hat{\theta}_j).$$

Thus it suffices to control $\mathcal{E}_j^{\mathrm{in}}(\hat{\theta}_j)$ and $\mathcal{E}_j(\hat{\theta}_j)$.

**Loss Function and Algorithm for GLM.** In Algorithm 1, for the GLM setup, we construct the loss function $\mathcal{L}$ using:

$$f_j(\theta) := \frac{1}{n} \sum_{i=1}^n \Big(\psi(x_{ji}^\top \theta) - y_{ji} x_{ji}^\top \theta\Big).$$

We set the domain of the parameter to $\mathbf{C} = \mathbf{B}(0, \xi)$. Apart from these changes, the procedure follows the methodology outlined in Section 3.

**In-Sample MSE Bounds.** The GLM analogue of Theorem 2 is as follows. Recall that $\zeta := \log(16m/\kappa)$.

**Theorem 4** (In-sample MSE: GLM). *Suppose the task relatedness condition (Definition 1) and GLM regularity assumptions (Assumption 2) hold. With probability at least $1 - \kappa$,*

*the estimator $\hat{\theta}_j$ obtained by Algorithm 1 with $\mathbf{C} = \mathbf{B}(0, \xi)$ and $\lambda_j = q\sqrt{d\zeta/n}$ ($q \gtrsim 1$) satisfies the following bounds simultaneously for all $j \in [m]$:*

- **Safety Guarantee (Universal):** *Regardless of the balancedness constant $B$ or the parameters $\varepsilon$ and $\delta$,*

$$\mathcal{E}_j^{\mathrm{in}}(\hat{\theta}_j) \lesssim q^2 \frac{d}{n}\zeta.$$

  *This matches the known independent-task GLM rate.*

- **Adaptivity to Relatedness (Inliers):** *If Assumption 1 holds with $B \lesssim \min(1/\varepsilon, m)$, then for all inlier tasks $j \in \mathbf{S}$:*

$$\mathcal{E}_j^{\mathrm{in}}(\hat{\theta}_j) \lesssim \left( \frac{Bd}{mn} + \min\left( B^2\delta^2, q^2\frac{d}{n} \right) \right. $$
$$\left. + q^2 B^2 \varepsilon^2 \frac{d}{n} \right)\zeta.$$

This mirrors the linear-model message: part (a) gives universal safety, while part (b) yields sharper transfer only in the favorable regime.

**Population MSE Bounds.** Combining Theorem 4 with Theorem 3 yields the corresponding GLM population guarantee: safety when $B$ is large or infinite, or when $\nu_j$ is large, and sharper transfer when both are moderate. For GLMs, the algorithm is already run over the bounded domain $\mathbf{C} = \mathbf{B}(0, \xi)$, so the intrinsic-dimension comparison in Theorem 3 applies directly without any additional post-processing.

## 7. Experiments

We empirically validate our method on both synthetic data and the real-world Human Activity Recognition (HAR) dataset from Anguita et al. (2013), which has also been used in prior robust MTL work (Duan and Wang, 2023; Tian et al., 2025). We benchmark against three baselines: Data Pooling (DP), Independent Task Learning (ITL), and the adaptive robust method (ARMUL) of Duan and Wang (2023). Full experimental details, including data generation and hyperparameter tuning protocols, are provided in Appendix A.

**Synthetic Data.** Our synthetic experiments sweep the inlier radius $\delta$, the outlier fraction $\varepsilon$, the eigendecay exponent, and a large-$B$ stress test based on common-operator-norm spiked covariances, while reporting MSE separately on all, related, and outlier tasks. Appendix A reports the detailed numbers and split-specific plots; the overall pattern is that our method sharpens the related-task risk in favorable regimes and falls back safely when transfer becomes unfavorable.

**Real-data: HAR Dataset.** On the HAR dataset (Anguita et al., 2013), we use the 561-dimensional features without PCA and formulate a multi-task logistic regression problem to classify the "standing" activity against all others. Appendix A reports the full protocol and results, where our method outperforms DP, ITL, and ARMUL.

## 8. Conclusion

In this paper, we have established an efficient multi-task linear regression framework that operates under strictly relaxed assumptions on the covariate spectrum. By introducing a matrix-weighted regularization scheme, we recover the $m, n, \delta, \varepsilon$ prediction-MSE rates of Duan and Wang (2023) under substantially weaker spectral assumptions. In the favorable balanced regime, the resulting task-overall MSE is minimax optimal up to logarithmic and balancedness-dependent factors. Crucially, our estimator provides a *safety guarantee*: it performs robustly when tasks are related and the balancedness constant $B$ is moderate, yet gracefully degrades to the independent-task rate when $B$ is large or infinite.

Promising directions for future research include investigating the tightness of the dependence on the balancedness constant $B$ and extending this geometric framework to infinite-dimensional settings such as reproducing kernel Hilbert spaces (RKHS) and overparameterized neural-network regimes, including neural tangent kernel limits.

## Acknowledgments

We thank the reviewers for their thoughtful and constructive feedback, which helped improve the presentation of this work. We are also grateful to Kaizheng Wang for helpful discussions.

## Impact Statement

This paper presents work whose goal is to advance the field of Machine Learning. There are many potential societal consequences of our work, none of which we feel must be specifically highlighted here.

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

# A. Numerical Experiments

The experiments below are theory-aligned diagnostics for contaminated linear and generalized linear multi-task learning. Accordingly, we compare against data pooling (DP), independent-task learning (ITL), and ARMUL (Duan and Wang, 2023). Code for reproducing the numerical experiments is available at https://github.com/seokjinkim0428/ Multi-task-Linear-Regression. The synthetic studies vary the quantities appearing in our theory: task related-ness, contamination, eigendecay, and balancedness. We also include a real-data experiment on the UCI Human Activity Recognition (HAR) dataset to illustrate the same matrix-weighted approach in a multi-task logistic regression setting.

## A.1. Synthetic Data

We benchmark our method against DP, ITL, and ARMUL. Unless otherwise stated, the synthetic experiments use $n = 100$, $m = 30$, and $d = 30$, and we evaluate population MSE for each task. The covariates $x_{ji}$ are sampled from the unit sphere and then scaled coordinatewise by $k^{-\alpha}$, where $\alpha$ controls eigendecay. For related tasks $j \in \mathbf{S}$, we set $\theta_j^\star = \theta^\star + \delta\Delta_j$, where $\Delta_j$ is sampled uniformly from the upper unit sphere; for outlier tasks $j \in \mathbf{S}^c$, we sample $\theta_j^\star$ from the same upper-hemisphere geometry with Euclidean radius $r_{\text{out}} = 10$. Responses are generated as $y_{ji} = x_{ji}^\top\theta_j^\star + \varepsilon_{ji}$, with $\varepsilon_{ji} \sim \mathcal{N}(0, 1)$.

We report MSE separately on all, related, and outlier tasks. For our method and ARMUL, we tune the multiplicative regularization parameter using 5-fold cross-validation over

$$\mathcal{Q}_{\text{synth}} = \{0.1, 0.4, 0.7, 1.0, 2.0, 4.0, 8.0, 16.0\},$$

and every point averages over 30 Monte Carlo repetitions. DP and ITL do not require hyperparameter tuning. Throughout this subsection, we use $\bar{B}$ to denote the population balancedness constant from Remark 2, namely the smallest scalar such that $\bar{\Sigma}_j \preceq \bar{B}\bar{\Sigma}_{\mathbf{S}}$ for all inlier tasks $j \in \mathbf{S}$. Our large-balancedness DGP is calibrated directly at this population-covariance level. In Gaussian random-design settings, standard covariance concentration implies that the empirical balancedness constant $B$ closely tracks $\bar{B}$ up to small fluctuations, so sweeping $\bar{B}$ is a natural proxy for sweeping $B$.

**Sweep of Task Relatedness $\delta$.** We vary $\delta \in \{0.2, 0.4, 0.8, 1.6, 3.2\}$ while keeping $\varepsilon = 0.1$, $\alpha = 1$, and the covariance geometry shared across tasks, so $\bar{B} = 1$. Across this favorable-transfer regime, our estimator consistently attains the smallest all-task MSE and the clearest gains on related tasks. Its all-task MSE ranges from $0.0138$ to $0.0259$, while the best competing baseline remains above $0.041$. On related tasks, our method remains well below the transfer baselines, with errors between $0.0022$ and $0.0136$. As expected, the outlier-only split is not where transfer helps most, and ITL remains competitive there.

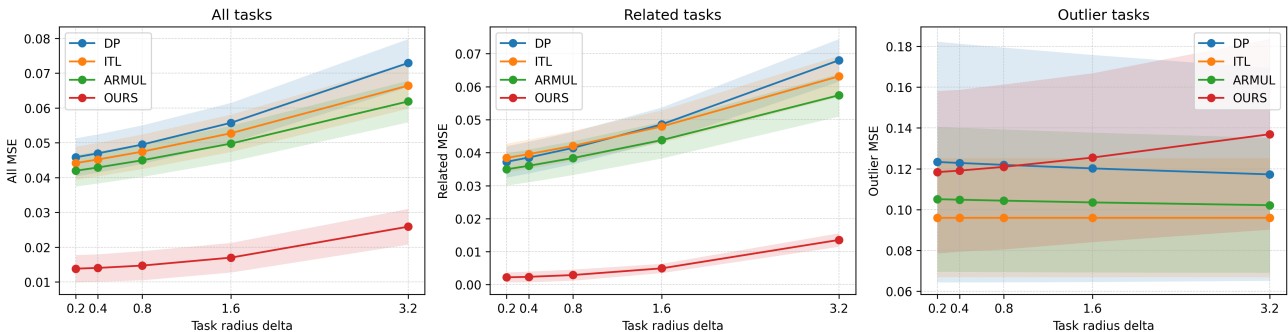

*Figure 1.* Synthetic sweep over the inlier radius $\delta$. Rows show all-task, related-task, and outlier-task MSE.

**Sweep of Outlier Fraction $\varepsilon$.** We next vary $\varepsilon \in \{0.05, 0.1, 0.2, 0.3, 0.4\}$, again under $\bar{B} = 1$. This directly isolates contamination while preserving favorable covariance alignment. Our method remains best on all-task MSE throughout the sweep and preserves a large advantage on related tasks even as the fraction of arbitrary outliers increases. The all-task error increases smoothly from $0.0062$ to $0.0460$, rather than showing an abrupt failure mode. On the outlier-only split, ITL is typically the strongest baseline, which is natural because those tasks are deliberately unrelated.

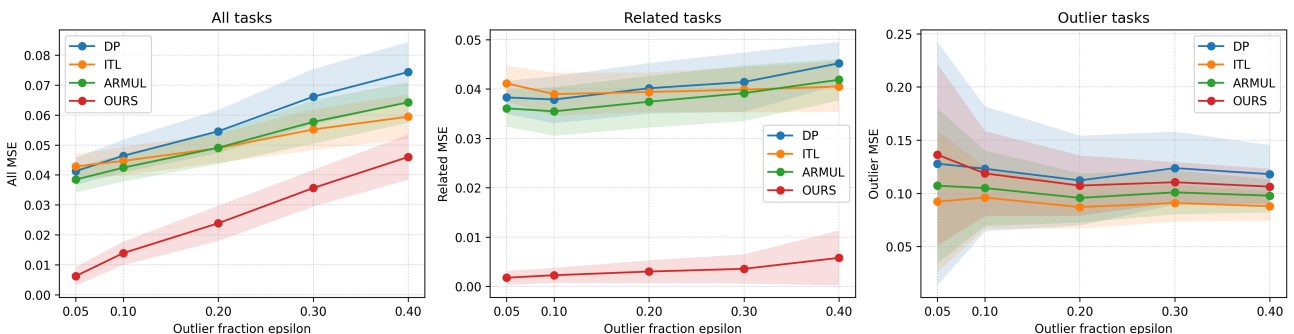

*Figure 2.* Synthetic sweep over the outlier fraction $\varepsilon$. Rows show all-task, related-task, and outlier-task MSE.

**Sweep of Eigendecay $\alpha$.** To stress the spectral assumptions directly, we keep $\bar{B} = 1$ and vary the eigendecay exponent over

$$\alpha \in \{0, 0.5, 1.0, 1.5, 2.0\}.$$

This experiment is the cleanest demonstration that our method does not require a uniform eigenvalue lower bound. Even when the spectrum decays sharply, our estimator continues to dominate on all-task and related-task error. In particular, the gap is already substantial at $\alpha = 0$, and it remains visible through the rapidly decaying cases $\alpha = 1.5$ and $2.0$.

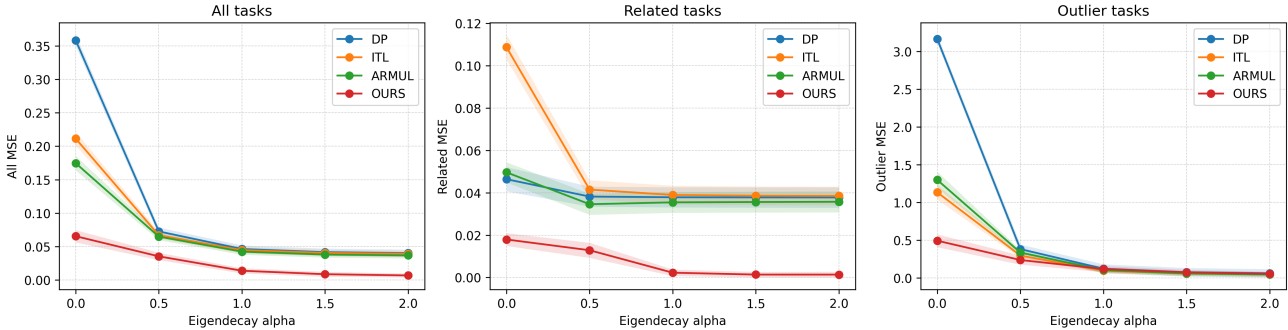

*Figure 3.* Synthetic sweep over the eigendecay exponent $\alpha$. Rows show all-task, related-task, and outlier-task MSE.

**Sweep of Population Balancedness $\bar{B}$.** Our large-$\bar{B}$ stress test uses a different DGP. For this section only, we set $m = 50$, $\varepsilon = 0.5$, and $\delta = 0.5$. We first sample the outlier set uniformly at random. The remaining tasks are inliers, and their covariances are assigned from a two-group spiked family:

$$\mathbf{\Sigma}_j = \eta \mathbf{I} + (1 - \eta) e_{g(j)} e_{g(j)}^\top.$$

Here $e_1, e_2$ are two coordinate directions and $g(j) \in \{1, 2\}$ records the spike group assigned to task $j$. We sweep target values $W \in \{5, 10, 15, 20\}$, where $W$ is the desired population balancedness level $\bar{B}$. Given $W$, let $r = \lfloor |\mathbf{S}|/W \rfloor$ and $p = r/|\mathbf{S}|$. Conditional on the random inlier set, we select $r$ inlier tasks uniformly without replacement and set $g(j) = 2$ for these minority-spike tasks; all remaining inlier tasks have $g(j) = 1$. Outlier tasks also use the majority-spike covariance $g(j) = 1$, but their regression parameters are sampled independently from the same outlier construction as above, with Euclidean radius $r_{\text{out}} = 10$. We choose

$$\eta = \frac{1/W - p}{1 - p},$$

clipped to $[0, 1]$. This choice calibrates the inlier average covariance along the minority spike direction. Indeed, along direction $e_2$, a minority task has variance 1, while the inlier average has variance $p + (1 - p)\eta$. Therefore the one-sided comparison ratio for a minority-spike task is

$$\frac{1}{p + (1 - p)\eta} = W,$$

up to the integer rounding in $r$. Thus the population balancedness constant is calibrated to $\bar{B} \approx W$. Since the design is Gaussian, standard covariance concentration implies that the empirical balancedness constant $B$ is close to this population value $\bar{B}$ once $n \gtrsim d$, up to logarithmic factors.

The resulting pattern is especially informative. When $\bar{B} = 5$, our estimator is the best overall, with all-task MSE $0.0074$. As $\bar{B}$ increases, ITL becomes the best method on all-task error, which is precisely the regime where aggressive transfer should lose its advantage. In this large-$\bar{B}$, high-contamination regime, our method tracks the ITL baseline closely and does not exhibit catastrophic negative transfer. Because this experiment uses the severe contamination level $\varepsilon = 0.5$, we view the all-task curve as the primary diagnostic; the related-task split is less informative here than in the earlier sweeps. We do not expect the errors to vary monotonically with $\bar{B}$, since changing $\bar{B}$ changes the underlying spiked-covariance mixture rather than merely increasing a single scalar difficulty parameter.

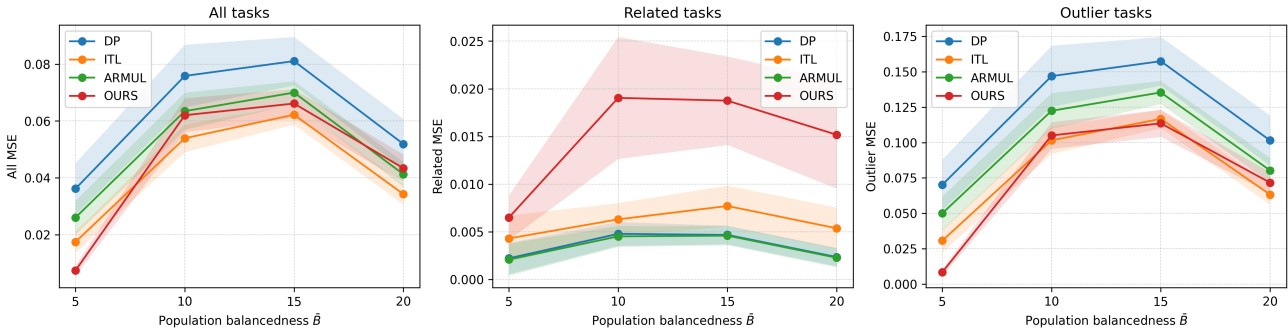

*Figure 4.* Large-$\bar{B}$ synthetic stress test based on common-operator-norm spiked covariances. Rows show all-task, related-task, and outlier-task MSE.

**Rank-Deficient Stress.** This same sweep also probes the rank-deficient regime. The calibrated floor $\eta$ is zero at the $\bar{B} = 5$ endpoint, so the inlier covariance matrices are singular rank-one spiked covariances; at the remaining sweep values the floor is small, giving near-singular designs. Thus the experiment tests both large-balancedness behavior and the singular/near-singular covariance regime that violates a uniform eigenvalue lower bound.

### A.2. Real-World Data

We evaluated the performance of our proposed algorithm against various baselines using the HAR dataset (Anguita et al., 2013). This dataset comprises motion data collected from 30 participants wearing waist-mounted smartphones equipped with inertial sensors. The dataset covers six distinct daily activities: walking, walking upstairs, walking downstairs, sitting, standing, and laying. The subject-level sample sizes range from 281 to 409 observations, with an average of about 343 observations per volunteer. Each data point is represented by a 561-dimensional feature vector, capturing characteristics in both the time and frequency domains.

Following Duan and Wang (2023), we formulated the problem as a multi-task logistic regression with $m = 30$ tasks (one task per volunteer). Unlike their HAR experiment, which distinguished `sitting` from the other activities, we instead used the label $y = 1$ for `standing` and $y = 0$ for all other activities. We randomly selected 20% of the data from each task for testing and trained logistic models on the remaining data. We intentionally do not apply PCA. The purpose of our method is precisely to remain stable when some covariate directions have small empirical variance, so discarding low-variance directions is not necessary for numerical stability and may remove useful information. This means that the HAR numbers below should be read as a theory-aligned real-data illustration, not as a direct reproduction of the HAR table in Duan and Wang (2023), whose label construction and PCA preprocessing differ from ours.

We compared our method with ARMUL, DP, and ITL, utilizing the logistic regression loss for all methods. For our method and ARMUL, we tuned the regularization parameter $q$—the multiplicative factor in $\lambda_j = q \frac{\sqrt{d}}{\sqrt{n_j}}$—using 5-fold cross-validation over the grid $q \in \{0.05, 0.10, 0.15, \ldots, 0.50\}$. We repeated the experiment over 30 independent random splits. As shown in Table 1, our method strictly outperformed DP, ITL, and ARMUL.

*Table 1.* Results for the real-world data experiment. Comparison of classification error rates (percentage). Standard deviations are reported in parentheses.

| Method | Mean Error (SD) |
|---|---|
| **Ours** | 1.25 (0.32) |
| DP | 7.61 (0.46) |
| ITL | 4.67 (0.51) |
| ARMUL (Duan and Wang, 2023) | 5.24 (0.43) |

As a diagnostic, the empirical balancedness constant on the raw HAR subject tasks is about $B_{\mathrm{emp}} \approx 30$: not a trivial $B \approx 1$ regime, but also far from a completely degenerate transfer geometry. Thus, the HAR experiment is not in a $B = \mathcal{O}(1)$ regime; rather, it illustrates that the proposed matrix-weighted procedure can remain empirically effective even when both the ambient dimension and the balancedness diagnostic are large. One plausible interpretation is that DP is vulnerable to subject heterogeneity because it forces all participants to share a single parameter, while ARMUL is disadvantaged by the failure of LBSM in this high-dimensional setting. The improvement over ITL suggests that balancedness is mainly a sufficient condition for our sharp theory: even when $B_{\mathrm{emp}}$ is large, matrix-weighted regularization can still exploit useful shared structure while remaining stable.

## B. Loss Geometry

The next results isolate the single-task $n$-sample geometry that underlies our multi-task analysis. Unlike the main contaminated multi-task setup, every statement in this section concerns a single regression problem with empirical second moment $\boldsymbol{\Sigma}$ and the matrix-weighted infimal convolution $f \star \lambda \| \cdot \|_{\boldsymbol{\Sigma}}$. We separate the linear and GLM cases because the domain and curvature assumptions differ. For a loss $f$, feasible set $\mathbf{C}$, and seminorm $\| \cdot \|_{\boldsymbol{\Sigma}}$, we use the convention

$$(f \star \lambda \| \cdot \|_{\boldsymbol{\Sigma}})(\beta) := \inf_{\theta \in \mathbf{C}} \left\{ f(\theta) + \lambda \|\theta - \beta\|_{\boldsymbol{\Sigma}} \right\}.$$

This definition is used in both the linear and GLM arguments below. In the linear case we take $\mathbf{C} = \mathbb{R}^d$, so the constraint is omitted; in the GLM case we take $\mathbf{C} = \mathbf{B}(0, \xi)$.

### B.1. Linear Regression

We first consider a single fixed-design linear regression problem with $n$ samples $\{(x_i, y_i)\}_{i=1}^n$, model

$$y_i = x_i^\top \theta^\star + \varepsilon_i,$$

design matrix $\mathbf{X} \in \mathbb{R}^{n \times d}$, response vector $\mathbf{Y} \in \mathbb{R}^n$, empirical second moment $\boldsymbol{\Sigma} := \frac{1}{n} \mathbf{X}^\top \mathbf{X}$, and squared loss

$$f(\theta) := \frac{1}{2n} \|\mathbf{Y} - \mathbf{X}\theta\|_2^2, \qquad g(\beta) := (f \star \lambda \| \cdot \|_{\boldsymbol{\Sigma}})(\beta).$$

The first ingredient identifies a regime where the regularized task objective agrees with the original loss and a complementary safety bound showing that regularization cannot hurt by more than the single-task rate.

**Whitened Notation.** When $\boldsymbol{\Sigma}$ is full-rank, we define the whitened design matrix by

$$\tilde{\mathbf{X}} := \mathbf{X}\boldsymbol{\Sigma}^{-1/2},$$

and, for any vector $v \in \mathbb{R}^d$, we define its whitened image by

$$\bar{v} := \boldsymbol{\Sigma}^{1/2}v.$$

For any function $h : \mathbb{R}^d \to \mathbb{R}$, we define its whitened version by

$$\bar{h}(\bar{v}) := h(\boldsymbol{\Sigma}^{-1/2}\bar{v}).$$

Thus $\bar{\theta}$, $\bar{\beta}$, $\bar{\theta}^\star$, $\bar{\bar{\theta}}$, and $\bar{\bar{\theta}}$ are all instances of the same map $v \mapsto \bar{v}$. In particular, the transformed loss is

$$\bar{f}(\bar{v}) := f(\mathbf{\Sigma}^{-1/2}\bar{v}) = \frac{1}{2n}\|\mathbf{Y} - \tilde{\mathbf{X}}\bar{v}\|_2^2.$$

When $\mathbf{\Sigma}$ is rank-deficient, the barred notation is interpreted in the projected row space in the rank-deficient case below.

**Proposition 1** (Infimal Convolution Invariant Region: Linear)**.** *If the regularization parameter satisfies*

$$\lambda > \|\theta^\star - \beta\|_{\mathbf{\Sigma}} + c_0 \frac{\sqrt{d\log(1/\kappa)}}{\sqrt{n}},$$

*then with probability at least $1 - \kappa$, we have $f(\beta) = g(\beta)$. Furthermore, any minimizer $\beta'$ of the objective $f(\theta) + \lambda\|\theta - \beta\|_{\mathbf{\Sigma}}$ satisfies $\|\beta' - \beta\|_{\mathbf{\Sigma}} = 0$.*

*Proof.* We first prove the result for the case where $\mathbf{\Sigma}$ is full-rank, and then extend it to the general case.

**Case 1: Full-Rank $\mathbf{\Sigma}$.** Note that $\frac{1}{n}\tilde{\mathbf{X}}^\top\tilde{\mathbf{X}} = \mathbf{I}_d$. The optimization problem can be rewritten in the transformed space as:

$$\min_\theta \left(\frac{1}{2n}\|\mathbf{Y} - \mathbf{X}\theta\|_2^2 + \lambda\|\theta - \beta\|_{\mathbf{\Sigma}}\right) = \min_{\bar{\theta}} \left(\frac{1}{2n}\|\mathbf{Y} - \tilde{\mathbf{X}}\bar{\theta}\|_2^2 + \lambda\|\bar{\theta} - \bar{\beta}\|_2\right).$$

By the first-order optimality condition for the $\ell_2$-regularized problem (or Lemma 13), the minimizer is exactly $\bar{\beta}$ if $\lambda > \|\nabla\bar{f}(\bar{\beta})\|_2$.

We calculate the gradient norm at $\bar{\beta}$:

$$\begin{aligned}
\|\nabla\bar{f}(\bar{\beta})\|_2 &= \left\|\frac{1}{n}\tilde{\mathbf{X}}^\top(\tilde{\mathbf{X}}\bar{\beta} - \mathbf{Y})\right\|_2 \\
&= \left\|\frac{1}{n}\tilde{\mathbf{X}}^\top(\tilde{\mathbf{X}}\bar{\beta} - \tilde{\mathbf{X}}\bar{\theta}^\star - \varepsilon)\right\|_2 \\
&\leq \left\|\frac{1}{n}\tilde{\mathbf{X}}^\top\tilde{\mathbf{X}}(\bar{\beta} - \bar{\theta}^\star)\right\|_2 + \left\|\frac{1}{n}\tilde{\mathbf{X}}^\top\varepsilon\right\|_2 \\
&= \|\bar{\beta} - \bar{\theta}^\star\|_2 + \left\|\frac{1}{n}\tilde{\mathbf{X}}^\top\varepsilon\right\|_2.
\end{aligned}$$

The first term equals $\|\beta - \theta^\star\|_{\mathbf{\Sigma}}$. For the second term, since the rows of $\tilde{\mathbf{X}}$ are isotropic, the Hanson-Wright inequality (Lemma 16) implies that with probability at least $1 - \kappa$:

$$\left\|\frac{1}{n}\tilde{\mathbf{X}}^\top\varepsilon\right\|_2 \leq c_0 \frac{\sqrt{d\log(1/\kappa)}}{\sqrt{n}}.$$

Thus, the condition on $\lambda$ ensures $\bar{\beta}$ is the minimizer, implying $\beta$ minimizes the original objective.

**Case 2: Rank-Deficient $\mathbf{\Sigma}$.** Let $\mathcal{R} := \text{Range}(\mathbf{\Sigma}) = \text{row}(\mathbf{X})$, and let $\mathbf{P}$ be the orthogonal projection onto $\mathcal{R}$. Since $\text{Null}(\mathbf{\Sigma}) = \text{Null}(\mathbf{X})$, for every $v \in \mathbb{R}^d$,

$$\mathbf{X}v = \mathbf{X}\mathbf{P}v, \qquad \|v\|_{\mathbf{\Sigma}} = \|\mathbf{P}v\|_{\mathbf{\Sigma}}.$$

Hence both the loss and the penalty in the linear objective depend on $\theta$ only through $\mathbf{P}\theta$, with $\beta$ entering only through $\mathbf{P}\beta$:

$$f(\theta) = f(\mathbf{P}\theta), \qquad \|\theta - \beta\|_{\mathbf{\Sigma}} = \|\mathbf{P}\theta - \mathbf{P}\beta\|_{\mathbf{\Sigma}}.$$

Because the linear case is unconstrained, the original problem is exactly equivalent to the reduced problem over $u \in \mathcal{R}$,

$$\min_{u\in\mathcal{R}}\{f(u) + \lambda\|u - \mathbf{P}\beta\|_{\mathbf{\Sigma}}\}.$$

On $\mathcal{R}$, the restricted operator $\boldsymbol{\Sigma}|_{\mathcal{R}}$ is positive definite, so Case 1 applies to the reduced $r = \text{rank}(\boldsymbol{\Sigma})$-dimensional problem with projected parameters $\mathbf{P}\theta^\star$ and $\mathbf{P}\beta$. The stochastic term is of order $\sqrt{r/n}$, which is bounded by the displayed $\sqrt{d/n}$ term. Since $\|\mathbf{P}\beta - \mathbf{P}\theta^\star\|_{\boldsymbol{\Sigma}} = \|\beta - \theta^\star\|_{\boldsymbol{\Sigma}}$, the assumed lower bound on $\lambda$ implies that $\mathbf{P}\beta$ is a minimizer of the reduced problem. Therefore $g(\beta) = f(\beta)$. Moreover, if $\beta'$ is any minimizer of the original objective, then $\mathbf{P}\beta'$ is a minimizer of the reduced problem, so $\|\mathbf{P}\beta' - \mathbf{P}\beta\|_{\boldsymbol{\Sigma}} = 0$. Equivalently, $\|\beta' - \beta\|_{\boldsymbol{\Sigma}} = 0$. $\qquad\square$

**Proposition 2** (Personalization: Linear). *With probability at least $1 - \kappa$, the following holds uniformly over all $\beta \in \mathbb{R}^d$: for every minimizer $\hat{\theta}(\beta)$ of $f(\theta) + \lambda\|\theta - \beta\|_{\boldsymbol{\Sigma}}$,*

$$\|\hat{\theta}(\beta) - \theta^\star\|_{\boldsymbol{\Sigma}} \leq \lambda + c_0 \sqrt{\frac{d}{n} \log\left(\frac{1}{\kappa}\right)}.$$

*Proof.* If $\boldsymbol{\Sigma}$ is rank-deficient, let $\mathcal{R} := \text{Range}(\boldsymbol{\Sigma}) = \text{row}(\mathbf{X})$ and let $\mathbf{P}$ be the orthogonal projection onto $\mathcal{R}$. As in the previous proposition, $f(\theta) = f(\mathbf{P}\theta)$ and $\|\theta - \beta\|_{\boldsymbol{\Sigma}} = \|\mathbf{P}\theta - \mathbf{P}\beta\|_{\boldsymbol{\Sigma}}$. Thus the unconstrained problem reduces exactly to $\mathcal{R}$, where $\boldsymbol{\Sigma}|_{\mathcal{R}}$ is positive definite; the stochastic bound on this $r = \text{rank}(\boldsymbol{\Sigma})$-dimensional space is no larger than the displayed $\sqrt{d}$ bound. It therefore suffices to prove the result on the working subspace, and we write the proof as if $\boldsymbol{\Sigma}$ were full-rank.

Fix an arbitrary $\beta \in \mathbb{R}^d$, and let $\hat{\theta} = \hat{\theta}(\beta)$. Let $\tilde{\theta}$ be the minimizer of $f(\theta)$ (OLS). The estimators in the transformed space, $\bar{\hat{\theta}} := \boldsymbol{\Sigma}^{1/2}\hat{\theta}$ and $\bar{\tilde{\theta}} := \boldsymbol{\Sigma}^{1/2}\tilde{\theta}$, are the minimizers of

$$\min_{\bar{\theta}} \left( \bar{f}(\bar{\theta}) + \lambda\|\bar{\theta} - \bar{\beta}\|_2 \right) \quad \text{and} \quad \min_{\bar{\theta}} \bar{f}(\bar{\theta}),$$

respectively.

**Step 1: Perturbation via Regularization.** Since $\nabla^2 \bar{f}(\bar{\theta}) = \mathbf{I}_d$, the function $\bar{f}$ is 1-strongly convex. The regularization term $h(\bar{\theta}) := \lambda\|\bar{\theta} - \bar{\beta}\|_2$ is convex and $\lambda$-Lipschitz with respect to $\|\cdot\|_2$. By Lemma 11,

$$\|\bar{\hat{\theta}} - \bar{\tilde{\theta}}\|_2 \leq \lambda,$$

which is equivalent to $\|\hat{\theta} - \tilde{\theta}\|_{\boldsymbol{\Sigma}} \leq \lambda$.

**Step 2: OLS Estimation Error.** The unregularized estimator $\tilde{\theta}$ is the OLS solution, and

$$\|\bar{\tilde{\theta}} - \bar{\theta}^\star\|_2 \leq \|\nabla \bar{f}(\bar{\theta}^\star)\|_2 = \left\|\frac{1}{n}\sum_{i=1}^n \varepsilon_i \tilde{x}_i\right\|_2,$$

where $\tilde{x}_i = \boldsymbol{\Sigma}^{-1/2} x_i$ are isotropic. Applying the Hanson-Wright inequality (Lemma 16), with probability at least $1 - \kappa$:

$$\|\bar{\tilde{\theta}} - \bar{\theta}^\star\|_2 \leq c_0 \sqrt{\frac{d}{n} \log\left(\frac{1}{\kappa}\right)}.$$

This is equivalent to $\|\tilde{\theta} - \theta^\star\|_{\boldsymbol{\Sigma}} \leq c_0 \sqrt{\frac{d}{n} \log(\frac{1}{\kappa})}$.

**Conclusion.** Combining the two bounds yields

$$\|\hat{\theta} - \theta^\star\|_{\boldsymbol{\Sigma}} \leq \|\hat{\theta} - \tilde{\theta}\|_{\boldsymbol{\Sigma}} + \|\tilde{\theta} - \theta^\star\|_{\boldsymbol{\Sigma}}$$

$$\leq \lambda + c_0 \sqrt{\frac{d}{n} \log\left(\frac{1}{\kappa}\right)}.$$

The only random event used above is the OLS noise event in Step 2, which does not depend on $\beta$. Hence the bound holds simultaneously for every $\beta$ and every corresponding minimizer $\hat{\theta}(\beta)$. $\qquad\square$

## B.2. Generalized Linear Models

We next consider a single fixed-design GLM with $n$ samples $\{(x_i, y_i)\}_{i=1}^n$, empirical second moment

$$\Sigma := \frac{1}{n} \sum_{i=1}^n x_i x_i^\top,$$

parameter domain $\mathbf{C} = \mathbf{B}(0, \xi)$ from Assumption 2, negative log-likelihood

$$f(\theta) := \frac{1}{n} \sum_{i=1}^n \left(-y_i x_i^\top \theta + \psi(x_i^\top \theta)\right), \qquad g(\beta) := (f \star \lambda \|\cdot\|_\Sigma)(\beta),$$

and again study the corresponding matrix-weighted infimal convolution on this single-task $n$-sample problem.

**Whitened Notation.** When $\Sigma$ is full-rank, define the whitened covariates by

$$\tilde{x}_i := \Sigma^{-1/2} x_i,$$

for any vector $v \in \mathbb{R}^d$, define its whitened image by

$$\bar{v} := \Sigma^{1/2} v,$$

and define the transformed domain by

$$\bar{\mathbf{C}} := \{\bar{v} : v \in \mathbf{C}\}.$$

For any function $h : \mathbb{R}^d \to \mathbb{R}$, define its whitened version by

$$\bar{h}(\bar{v}) := h(\Sigma^{-1/2} \bar{v}).$$

Thus $\bar{\theta}$, $\bar{\beta}$, $\bar{\theta}^\star$, $\bar{\hat{\theta}}$, and $\bar{\bar{\theta}}$ are all instances of the same map $v \mapsto \bar{v}$. In particular, the transformed loss is

$$\bar{f}(\bar{v}) := f(\Sigma^{-1/2} \bar{v}) = \frac{1}{n} \sum_{i=1}^n \left(-y_i \tilde{x}_i^\top \bar{v} + \psi(\tilde{x}_i^\top \bar{v})\right).$$

When $\Sigma$ is rank-deficient, the barred notation is interpreted on the projected row space, with the projected feasible set described in the proof below.

**Proposition 3** (Infimal Convolution Invariant Region: GLM)**.** *If $\beta \in \mathbf{C}$ and $\lambda > \alpha_u \|\beta - \theta^\star\|_\Sigma + c_0 \frac{\sqrt{d \log(1/\kappa)}}{\sqrt{n}}$, then with probability at least $1 - \kappa$,*

$$\beta \in \arg\min_{\theta \in \mathbf{C}} \left(f(\theta) + \lambda \|\theta - \beta\|_\Sigma\right).$$

*Consequently, $f(\beta) = (f \star \lambda \|\cdot\|_\Sigma)(\beta)$. Furthermore, any minimizer $\beta'$ satisfies $\|\beta' - \beta\|_\Sigma = 0$.*

*Proof.* We first consider the case where $\Sigma$ is full-rank. The original optimization problem is equivalent to the transformed problem:

$$\beta \in \arg\min_{\theta \in \mathbf{C}} \left(f(\theta) + \lambda \|\theta - \beta\|_\Sigma\right) \iff \bar{\beta} \in \arg\min_{\bar{\theta} \in \bar{\mathbf{C}}} \left(\bar{f}(\bar{\theta}) + \lambda \|\bar{\theta} - \bar{\beta}\|_2\right).$$

Since $\bar{f}$ is convex on $\bar{\mathbf{C}}$ and $\bar{\beta} \in \bar{\mathbf{C}}$, by Lemma 14, it suffices to show that $\|\nabla \bar{f}(\bar{\beta})\|_2 < \lambda$.

We compute the gradient $\nabla \bar{f}(\bar{\beta})$:

$$\nabla \bar{f}(\bar{\beta}) = \frac{1}{n} \sum_{i=1}^n \left(-y_i \tilde{x}_i + \psi'(\tilde{x}_i^\top \bar{\beta}) \tilde{x}_i\right).$$

Substituting the model equation $y_i = \psi'(\tilde{x}_i^\top \theta^\star) + \varepsilon_i$, we decompose the gradient into a noise term and a bias term:

$$\nabla \bar{f}(\bar{\beta}) = -\frac{1}{n} \sum_{i=1}^n \varepsilon_i \tilde{x}_i + \frac{1}{n} \sum_{i=1}^n \left( \psi'(\tilde{x}_i^\top \bar{\beta}) - \psi'(\tilde{x}_i^\top \theta^\star) \right) \tilde{x}_i.$$

By the Mean Value Theorem, there exists $\xi_i$ between $\tilde{x}_i^\top \bar{\beta}$ and $\tilde{x}_i^\top \theta^\star$ such that:

$$\psi'(\tilde{x}_i^\top \bar{\beta}) - \psi'(\tilde{x}_i^\top \theta^\star) = \psi''(\xi_i) \tilde{x}_i^\top (\bar{\beta} - \theta^\star).$$

Substituting this back, we obtain:

$$\nabla \bar{f}(\bar{\beta}) = -\frac{1}{n} \sum_{i=1}^n \varepsilon_i \tilde{x}_i + \left( \frac{1}{n} \sum_{i=1}^n \psi''(\xi_i) \tilde{x}_i \tilde{x}_i^\top \right) (\bar{\beta} - \theta^\star).$$

Let $\tilde{\mathbf{Q}} := \frac{1}{n} \sum_{i=1}^n \psi''(\xi_i) \tilde{x}_i \tilde{x}_i^\top$. Under Assumption 2, we have $\psi''(\cdot) \in [\alpha_\ell, \alpha_u]$. Since $\frac{1}{n} \sum_{i=1}^n \tilde{x}_i \tilde{x}_i^\top = \mathbf{I}_d$, the matrix $\tilde{\mathbf{Q}}$ satisfies:

$$\alpha_\ell \mathbf{I}_d \preceq \tilde{\mathbf{Q}} \preceq \alpha_u \mathbf{I}_d \implies \|\tilde{\mathbf{Q}}\|_{\mathrm{op}} \leq \alpha_u.$$

We now bound the norm of the gradient. Using the triangle inequality:

$$\|\nabla \bar{f}(\bar{\beta})\|_2 \leq \left\| \frac{1}{n} \sum_{i=1}^n \varepsilon_i \tilde{x}_i \right\|_2 + \|\tilde{\mathbf{Q}}\|_{\mathrm{op}} \|\bar{\beta} - \theta^\star\|_2$$

$$\leq \left\| \frac{1}{n} \sum_{i=1}^n \varepsilon_i \tilde{x}_i \right\|_2 + \alpha_u \|\beta - \theta^\star\|_{\boldsymbol{\Sigma}}.$$

Since the whitened covariates $\tilde{x}_i$ are isotropic, we apply the Hanson-Wright inequality (Lemma 16). With probability at least $1 - \kappa$:

$$\left\| \frac{1}{n} \sum_{i=1}^n \varepsilon_i \tilde{x}_i \right\|_2 \leq c_0 \frac{\sqrt{d \log(1/\kappa)}}{\sqrt{n}}.$$

Thus, provided that $\lambda > \alpha_u \|\beta - \theta^\star\|_{\boldsymbol{\Sigma}} + c_0 \frac{\sqrt{d \log(1/\kappa)}}{\sqrt{n}}$, the condition $\|\nabla \bar{f}(\bar{\beta})\|_2 < \lambda$ holds, implying $\beta$ is a minimizer over $\mathbf{C}$.

For the case where $\boldsymbol{\Sigma}$ is rank-deficient, let $\mathcal{R} := \mathrm{Range}(\boldsymbol{\Sigma}) = \mathrm{row}(\mathbf{X})$, and let $\mathbf{P}$ be the orthogonal projection onto $\mathcal{R}$. As above, $\mathrm{Null}(\boldsymbol{\Sigma}) = \mathrm{Null}(\mathbf{X})$, so the GLM linear predictors and the seminorm are unchanged by projection:

$$x_i^\top \theta = x_i^\top \mathbf{P}\theta, \qquad \|\theta - \beta\|_{\boldsymbol{\Sigma}} = \|\mathbf{P}\theta - \mathbf{P}\beta\|_{\boldsymbol{\Sigma}}.$$

The constrained domain causes no further difficulty. Here $\mathbf{C} = \mathbf{B}(0, \xi)$, and orthogonal projections are contractions; hence

$$\mathbf{PC} = \mathbf{C} \cap \mathcal{R} \subseteq \mathbf{C}.$$

Thus replacing any feasible $\theta$ by $\mathbf{P}\theta$ preserves all GLM linear predictors and the $\boldsymbol{\Sigma}$-seminorm penalty while keeping the parameter feasible. The original constrained problem is therefore equivalent, in objective value and minimizer projections, to the reduced problem over $u \in \mathbf{PC} = \mathbf{C} \cap \mathcal{R}$, with $\beta$ and $\theta^\star$ replaced by $\mathbf{P}\beta$ and $\mathbf{P}\theta^\star$. Because the reduced feasible set remains inside $\mathbf{C}$, the curvature condition in Assumption 2 is valid throughout the reduced problem. On $\mathcal{R}$, the restricted operator $\boldsymbol{\Sigma}|_{\mathcal{R}}$ is positive definite, so the full-rank argument applies in dimension $r = \mathrm{rank}(\boldsymbol{\Sigma})$, with the stochastic $\sqrt{r}$ term bounded by the displayed $\sqrt{d}$ term. Since $\|\mathbf{P}\beta - \mathbf{P}\theta^\star\|_{\boldsymbol{\Sigma}} = \|\beta - \theta^\star\|_{\boldsymbol{\Sigma}}$, the same lower bound on $\lambda$ implies that $\mathbf{P}\beta$ minimizes the reduced problem. Equivalently, $\beta$ is a minimizer of the original constrained problem, and any minimizer $\beta'$ satisfies $\|\mathbf{P}\beta' - \mathbf{P}\beta\|_{\boldsymbol{\Sigma}} = 0$, i.e., $\|\beta' - \beta\|_{\boldsymbol{\Sigma}} = 0$. $\square$

**Lemma 1** (Strong Convexity and Growth Condition: GLM). *Suppose $\boldsymbol{\Sigma}$ is full-rank, and let $\bar{f}$ be the transformed loss defined in Proposition 3 on the domain $\bar{\mathbf{C}}$. The function $\bar{f}$ is $\alpha_\ell$-strongly convex. Consequently, let $\bar{\bar{\theta}}$ be the minimizer of $\bar{f}$ over the convex set $\bar{\mathbf{C}}$. Then for any $\bar{\theta} \in \bar{\mathbf{C}}$:*

$$\bar{f}(\bar{\theta}) \geq \bar{f}(\bar{\bar{\theta}}) + \frac{\alpha_\ell}{2} \|\bar{\theta} - \bar{\bar{\theta}}\|_2^2.$$

*Proof.* First, we establish the uniform lower bound on the Hessian. The Hessian of the transformed loss is given by:

$$\nabla^2 \bar{f}(\bar{\theta}) = \frac{1}{n} \sum_{i=1}^{n} \psi''(\tilde{x}_i^\top \bar{\theta}) \tilde{x}_i \tilde{x}_i^\top.$$

Under Assumption 2, the second derivative of the cumulant generating function satisfies $\psi''(z) \geq \alpha_\ell$ for all $z$ in the domain. Additionally, by the construction of the whitened covariates $\tilde{x}_i = \Sigma^{-1/2} x_i$, the empirical covariance is isotropic:

$$\frac{1}{n} \sum_{i=1}^{n} \tilde{x}_i \tilde{x}_i^\top = \Sigma^{-1/2} \left( \frac{1}{n} \sum_{i=1}^{n} x_i x_i^\top \right) \Sigma^{-1/2} = \Sigma^{-1/2} \Sigma \Sigma^{-1/2} = \mathbf{I}_d.$$

This implies $\nabla^2 \bar{f}(\bar{\theta}) \succeq \alpha_\ell \mathbf{I}_d$ for all $\bar{\theta} \in \bar{\mathbf{C}}$, establishing $\alpha_\ell$-strong convexity.

Now, applying the definition of strong convexity at the minimizer $\bar{\bar{\theta}}$:

$$\bar{f}(\bar{\theta}) \geq \bar{f}(\bar{\bar{\theta}}) + \nabla \bar{f}(\bar{\bar{\theta}})^\top (\bar{\theta} - \bar{\bar{\theta}}) + \frac{\alpha_\ell}{2} \|\bar{\theta} - \bar{\bar{\theta}}\|_2^2.$$

Since $\bar{\mathbf{C}}$ is a convex set and $\bar{\bar{\theta}}$ is the minimizer over $\bar{\mathbf{C}}$, the first-order optimality condition states that $\langle \nabla \bar{f}(\bar{\bar{\theta}}), \bar{\theta} - \bar{\bar{\theta}} \rangle \geq 0$ for all $\bar{\theta} \in \bar{\mathbf{C}}$. Dropping the non-negative linear term yields the desired result:

$$\bar{f}(\bar{\theta}) \geq \bar{f}(\bar{\bar{\theta}}) + \frac{\alpha_\ell}{2} \|\bar{\theta} - \bar{\bar{\theta}}\|_2^2.$$

$\square$

**Proposition 4** (Personalization: GLM). *Let $\hat{\theta}$ be a minimizer of*

$$\min_{\theta \in \mathbf{C}} \left\{ f(\theta) + \lambda \|\theta - \beta\|_\Sigma \right\},$$

*and let $\tilde{\theta}$ be the minimizer of the unregularized loss $f(\theta)$. Then for any $\lambda > 0$, we have:*

$$\|\hat{\theta} - \tilde{\theta}\|_\Sigma \leq \frac{2\lambda}{\alpha_\ell}.$$

*Consequently, with probability at least $1 - \kappa$:*

$$\|\hat{\theta} - \theta^\star\|_\Sigma \leq \frac{2\lambda}{\alpha_\ell} + \frac{c_0}{\alpha_\ell} \sqrt{\frac{d \log(2/\kappa)}{n}}.$$

*Proof.* If $\Sigma$ is rank-deficient, let $\mathcal{R} := \text{Range}(\Sigma) = \text{row}(\mathbf{X})$, and let $\mathbf{P}$ be the orthogonal projection onto $\mathcal{R}$. For any $\theta \in \mathbf{C}$, the GLM loss and the matrix seminorm depend only on $\mathbf{P}\theta$, and the projected feasible set is

$$\mathbf{PC} = \mathbf{C} \cap \mathcal{R} \subseteq \mathbf{C},$$

because $\mathbf{C} = \mathbf{B}(0, \xi)$. Therefore the regularized and unregularized constrained problems reduce exactly to $\mathbf{PC}$ on $\mathcal{R}$, with $\beta$ replaced by $\mathbf{P}\beta$. The curvature bounds remain valid on this reduced feasible set, and $\Sigma|_\mathcal{R}$ is positive definite. The proof below applies on $\mathcal{R}$; converting back gives the same $\Sigma$-seminorm bounds, and the stochastic term is controlled by $\sqrt{\text{rank}(\Sigma)} \leq \sqrt{d}$. We therefore write the argument in full-rank notation.

We proceed by analyzing the problem in the whitened domain. Let $\bar{\theta} := \Sigma^{1/2}\theta$ and define the transformed domain $\bar{\mathbf{C}} := \{\Sigma^{1/2}\theta \mid \theta \in \mathbf{C}\}$. The transformed loss function is given by $\bar{f}(\bar{\theta}) := f(\Sigma^{-1/2}\bar{\theta})$.

Let $\hat{\bar{\theta}} = \Sigma^{1/2}\hat{\theta}$ and $\bar{\bar{\theta}} = \Sigma^{1/2}\tilde{\theta}$ be the minimizers in $\bar{\mathbf{C}}$ for the regularized and unregularized problems, respectively.

**Step 1: Lower Bound via Strong Convexity.** By Lemma 1, $\bar{f}$ is $\alpha_\ell$-strongly convex on $\bar{\mathbf{C}}$. Using the property of strongly convex functions and the first-order optimality condition for $\bar{\bar{\theta}}$ (i.e., $\langle \nabla \bar{f}(\bar{\bar{\theta}}), \bar{\theta} - \bar{\bar{\theta}} \rangle \geq 0$ for all $\bar{\theta} \in \bar{\mathbf{C}}$), we have:

$$\bar{f}(\hat{\bar{\theta}}) - \bar{f}(\bar{\bar{\theta}}) \geq \langle \nabla \bar{f}(\bar{\bar{\theta}}), \hat{\bar{\theta}} - \bar{\bar{\theta}} \rangle + \frac{\alpha_\ell}{2} \|\hat{\bar{\theta}} - \bar{\bar{\theta}}\|_2^2$$

$$\geq \frac{\alpha_\ell}{2} \|\hat{\bar{\theta}} - \bar{\bar{\theta}}\|_2^2.$$

**Step 2: Upper Bound via Regularization.** From the optimality of $\bar{\hat{\theta}}$ for the regularized objective:

$$\bar{f}(\bar{\hat{\theta}}) + \lambda\|\bar{\hat{\theta}} - \bar{\beta}\|_2 \leq \bar{f}(\bar{\tilde{\theta}}) + \lambda\|\bar{\tilde{\theta}} - \bar{\beta}\|_2.$$

Rearranging terms and applying the reverse triangle inequality:

$$\bar{f}(\bar{\hat{\theta}}) - \bar{f}(\bar{\tilde{\theta}}) \leq \lambda\left(\|\bar{\tilde{\theta}} - \bar{\beta}\|_2 - \|\bar{\hat{\theta}} - \bar{\beta}\|_2\right)$$
$$\leq \lambda\|\bar{\hat{\theta}} - \bar{\tilde{\theta}}\|_2.$$

**Step 3: Combining Bounds.** Combining the inequalities from Steps 1 and 2:

$$\frac{\alpha_\ell}{2}\|\bar{\hat{\theta}} - \bar{\tilde{\theta}}\|_2^2 \leq \lambda\|\bar{\hat{\theta}} - \bar{\tilde{\theta}}\|_2.$$

Assuming $\|\bar{\hat{\theta}} - \bar{\tilde{\theta}}\|_2 > 0$ (otherwise the bound holds trivially), we divide by the norm to obtain:

$$\|\bar{\hat{\theta}} - \bar{\tilde{\theta}}\|_2 \leq \frac{2\lambda}{\alpha_\ell}.$$

Converting back to the original space using $\|\bar{v}\|_2 = \|v\|_{\boldsymbol{\Sigma}}$, we get $\|\hat{\theta} - \tilde{\theta}\|_{\boldsymbol{\Sigma}} \leq \frac{2\lambda}{\alpha_\ell}$.

**Step 4: Final Error Bound.** The unregularized GLM estimator $\tilde{\theta}$ satisfies the standard concentration bound (derived from the $\alpha_\ell$-strong convexity and Hanson-Wright inequality on the gradient):

$$\|\tilde{\theta} - \theta^\star\|_{\boldsymbol{\Sigma}} \leq \frac{1}{\alpha_\ell}\|\nabla\bar{f}(\bar{\theta}^\star)\|_2 \leq \frac{c_0}{\alpha_\ell}\sqrt{\frac{d\log(2/\kappa)}{n}}.$$

Applying the triangle inequality $\|\hat{\theta} - \theta^\star\|_{\boldsymbol{\Sigma}} \leq \|\hat{\theta} - \tilde{\theta}\|_{\boldsymbol{\Sigma}} + \|\tilde{\theta} - \theta^\star\|_{\boldsymbol{\Sigma}}$ yields the final result. □

## C. Proofs for the Linear Model

We now turn to the genuinely multi-task part of the linear-model proof. The single-task $n$-sample loss geometry was isolated in Section B; here we set up the pooled objectives, analyze the inlier pooling estimator, control the outlier perturbation step, and conclude the proof of Theorem 2.

### C.1. Multi-Task Setup and Notation

Throughout the transfer proof, all quantities involving the inlier average $\boldsymbol{\Sigma}_{\mathbf{S}}$ are interpreted after projecting onto $\mathrm{Range}(\boldsymbol{\Sigma}_{\mathbf{S}})$. On this projected subspace, the displayed inverses are ordinary inverses. This entails no loss for in-sample risk, because Assumption 1 implies that any direction in $\mathrm{Null}(\boldsymbol{\Sigma}_{\mathbf{S}})$ is also in $\mathrm{Null}(\boldsymbol{\Sigma}_j)$ for every inlier task $j \in \mathbf{S}$. Thus, in the proof below, we work on this projected subspace and treat $\boldsymbol{\Sigma}_{\mathbf{S}}$ as full-rank. For the linear model, we work on the unconstrained domain $\mathbf{C} = \mathbb{R}^d$.

For any subset of tasks $\mathbf{I} \subseteq [m]$, let

$$N_{\mathbf{I}} := \sum_{j\in\mathbf{I}} n_j, \qquad \mathbf{V}_{\mathbf{I}} := \sum_{j\in\mathbf{I}} \mathbf{V}_j, \qquad \boldsymbol{\Sigma}_{\mathbf{I}} := \frac{1}{N_{\mathbf{I}}}\mathbf{V}_{\mathbf{I}}.$$

In the balanced case analyzed in the main theorem, $n_j = n$ and hence $N_{\mathbf{S}} = n|\mathbf{S}|$. We define the task loss

$$f_j(\beta) := \frac{1}{2n}\sum_{i=1}^n (y_{ji} - x_{ji}^\top\beta)^2$$

and its matrix-weighted infimal convolution

$$g_j(\beta) := \left(f_j \star \lambda\|\cdot\|_{\boldsymbol{\Sigma}_j}\right)(\beta) = \inf_{u\in\mathbb{R}^d}\left\{f_j(u) + \lambda\|u - \beta\|_{\boldsymbol{\Sigma}_j}\right\}.$$

We further write

$$F(\beta) := \sum_{j \in [m]} f_j(\beta), \qquad\qquad G(\beta) := \sum_{j \in [m]} g_j(\beta),$$

$$F_{\mathbf{S}}(\beta) := \sum_{j \in \mathbf{S}} f_j(\beta), \qquad\qquad G_{\mathbf{S}}(\beta) := \sum_{j \in \mathbf{S}} g_j(\beta).$$

We also denote the aggregate outlier contribution by

$$L(\beta) := \sum_{j \in \mathbf{S}^c} g_j(\beta), \qquad \text{so that} \qquad G(\beta) = G_{\mathbf{S}}(\beta) + L(\beta).$$

Let $\hat{\beta}_F$, $\hat{\beta}_{F_{\mathbf{S}}}$, $\hat{\beta}_G$, and $\hat{\beta}_{G_{\mathbf{S}}}$ denote minimizers of these objectives. We also define the invariant region

$$R_j := \{\beta \in \mathbf{C} \mid f_j(\beta) = g_j(\beta)\}, \qquad j \in [m],$$

and, for each inlier task, $\eta_j^\star := \theta_j^\star - \theta^\star$.

**Pooled Whitening Notation.** On $\mathrm{Range}(\boldsymbol{\Sigma}_{\mathbf{S}})$, we define the pooled whitening map by

$$\bar{v} := \boldsymbol{\Sigma}_{\mathbf{S}}^{1/2} v, \qquad v \in \mathbb{R}^d.$$

For any function $h : \mathbb{R}^d \to \mathbb{R}$, we define its pooled-whitened version by

$$\bar{h}(\bar{v}) := h(\boldsymbol{\Sigma}_{\mathbf{S}}^{-1/2} \bar{v}),$$

where $\boldsymbol{\Sigma}_{\mathbf{S}}^{-1/2}$ denotes the inverse on $\mathrm{Range}(\boldsymbol{\Sigma}_{\mathbf{S}})$. Thus $\bar{\theta}$, $\bar{\beta}$, $\bar{\theta}^\star$, $\bar{\theta}_j^\star$, $\bar{\hat{\beta}}_F$, $\bar{\hat{\beta}}_{F_{\mathbf{S}}}$, $\bar{\hat{\beta}}_G$, and $\bar{\hat{\beta}}_{G_{\mathbf{S}}}$ are all instances of the same pooled whitening map. In particular, we write $\bar{F}$, $\bar{G}$, $\bar{F}_{\mathbf{S}}$, $\bar{G}_{\mathbf{S}}$, and $\bar{L}$ for the transformed pooled objectives.

### C.2. The Inlier Pooling Estimator

The second ingredient is a high-probability description of the inlier pooling estimator. This identifies the statistical rate of the ideal inlier-only procedure and characterizes a regime in which the regularized inlier objective collapses to the same minimizer.

**Lemma 2** (Error Bound of Pooling Estimator: Linear)**.** *Let $\hat{\beta}_{F_{\mathbf{S}}}$ be the minimizer of $F_{\mathbf{S}}$. Under Assumption 1, with probability at least $1 - \frac{\kappa}{4}$, simultaneously for all $j \in \mathbf{S}$,*

$$\|\hat{\beta}_{F_{\mathbf{S}}} - \theta^\star\|_{\boldsymbol{\Sigma}_j} \le \sqrt{B}\delta + c_0 \sqrt{\frac{Bd\zeta}{N_{\mathbf{S}}}}.$$

*Proof.* Recall the closed-form solution for the pooling estimator:

$$\hat{\beta}_{F_{\mathbf{S}}} = \mathbf{V}_{\mathbf{S}}^{-1} \sum_{k \in \mathbf{S}} \mathbf{X}_k^\top \mathbf{Y}_k = \theta^\star + \mathbf{V}_{\mathbf{S}}^{-1} \sum_{k \in \mathbf{S}} \mathbf{X}_k^\top (\mathbf{X}_k \eta_k^\star + \varepsilon_k).$$

We analyze the error in the $\boldsymbol{\Sigma}_j$-norm, decomposing it into a bias term and a variance term:

$$\|\hat{\beta}_{F_{\mathbf{S}}} - \theta^\star\|_{\boldsymbol{\Sigma}_j} \le \underbrace{\left\| \mathbf{V}_{\mathbf{S}}^{-1} \sum_{k \in \mathbf{S}} \mathbf{V}_k \eta_k^\star \right\|_{\boldsymbol{\Sigma}_j}}_{\text{Bias}} + \underbrace{\left\| \mathbf{V}_{\mathbf{S}}^{-1} \sum_{k \in \mathbf{S}} \mathbf{X}_k^\top \varepsilon_k \right\|_{\boldsymbol{\Sigma}_j}}_{\text{Variance}}.$$

**Bias Term.** Using the balancedness assumption $\boldsymbol{\Sigma}_j \preceq B\boldsymbol{\Sigma}_{\mathbf{S}} = \frac{B}{N_{\mathbf{S}}} \mathbf{V}_{\mathbf{S}}$, we have $\boldsymbol{\Sigma}_j^{1/2} \mathbf{V}_{\mathbf{S}}^{-1} \boldsymbol{\Sigma}_j^{1/2} \preceq \frac{B}{N_{\mathbf{S}}} \mathbf{I}$. Let $\mathbf{e}$ be the stacked vector of $\{\mathbf{X}_k \eta_k^\star\}_{k \in \mathbf{S}} \in \mathbb{R}^{N_{\mathbf{S}}}$. Then

$$(\text{Bias})^2 \le \frac{B}{N_{\mathbf{S}}} \mathbf{e}^\top \mathbf{X}_{\mathbf{S}} (\mathbf{X}_{\mathbf{S}}^\top \mathbf{X}_{\mathbf{S}})^{-1} \mathbf{X}_{\mathbf{S}}^\top \mathbf{e} \le \frac{B}{N_{\mathbf{S}}} \|\mathbf{e}\|_2^2.$$

Since $\|\mathbf{e}\|_2^2 \le N_{\mathbf{S}}\delta^2$, we obtain Bias $\le \sqrt{B}\delta$.

**Variance Term.** Let $\mathbf{X_S}$ be the stacked inlier design and let $\varepsilon_{\mathbf{S}}$ be the stacked inlier noise vector. The squared variance term is the quadratic form

$$\varepsilon_{\mathbf{S}}^{\top} \mathbf{X_S} \mathbf{V_S}^{-1} \Sigma_j \mathbf{V_S}^{-1} \mathbf{X_S}^{\top} \varepsilon_{\mathbf{S}}.$$

Since $\mathbf{V_S} = N_{\mathbf{S}} \Sigma_{\mathbf{S}}$, Assumption 1 gives

$$\mathbf{V_S}^{-1/2} \Sigma_j \mathbf{V_S}^{-1/2} \preceq \frac{B}{N_{\mathbf{S}}} \mathbf{I}.$$

Thus the quadratic-form matrix has trace at most $Bd/N_{\mathbf{S}}$ and operator norm at most $B/N_{\mathbf{S}}$. Applying the Hanson-Wright inequality (Lemma 16) with failure level $\kappa/(4m)$ for each task $j$ gives, on a union-bound event of probability at least $1 - \kappa/4$,

$$\text{Variance} \leq c_0 \sqrt{\frac{Bd\zeta}{N_{\mathbf{S}}}}.$$

Combining both terms yields the result. $\qquad\square$

**Proposition 5** (Equivalence of Pooling Estimators: Linear). *If the regularization parameter satisfies*

$$\lambda > (\sqrt{B} + 1)\delta + c_0 \sqrt{\frac{d\zeta}{n}} + c_0 \sqrt{\frac{Bd\zeta}{N_{\mathbf{S}}}},$$

*then with probability at least $1 - \frac{\kappa}{2}$, we have $\hat{\beta}_{G_{\mathbf{S}}} = \hat{\beta}_{F_{\mathbf{S}}}$ and $\hat{\beta}_{F_{\mathbf{S}}} \in R_j$ for all $j \in \mathbf{S}$.*

*Proof.* Let $\mathcal{E}_{\text{pool}}$ be the pooling-estimator event from Lemma 2; it has probability at least $1 - \kappa/4$. Let $\mathcal{E}_{\text{inv}}$ be the event that the taskwise noise bound used in Proposition 1 holds for every $j \in \mathbf{S}$, each invoked with failure level $\kappa/(4m)$. By a union bound, $\mathbb{P}(\mathcal{E}_{\text{inv}}) \geq 1 - \kappa/4$, and hence $\mathbb{P}(\mathcal{E}_{\text{pool}} \cap \mathcal{E}_{\text{inv}}) \geq 1 - \kappa/2$. On this intersection, a sufficient condition for $\beta \in R_j$ is $\lambda > \|\theta_j^{\star} - \beta\|_{\Sigma_j} + c_0 \sqrt{\frac{d\zeta}{n}}$. Substituting $\beta = \hat{\beta}_{F_{\mathbf{S}}}$:

$$\|\theta_j^{\star} - \hat{\beta}_{F_{\mathbf{S}}}\|_{\Sigma_j} \leq \|\theta_j^{\star} - \theta^{\star}\|_{\Sigma_j} + \|\theta^{\star} - \hat{\beta}_{F_{\mathbf{S}}}\|_{\Sigma_j}$$
$$\leq \delta + \left( \sqrt{B}\delta + c_0 \sqrt{\frac{Bd\zeta}{N_{\mathbf{S}}}} \right),$$

where we used Lemma 2. The assumed lower bound on $\lambda$ ensures this condition holds strictly for all $j \in \mathbf{S}$. Thus, $\hat{\beta}_{F_{\mathbf{S}}}$ lies in the invariant region of every task $j \in \mathbf{S}$. For each $j \in \mathbf{S}$, define the positive slack

$$\Delta_j := \lambda - \left( \|\theta_j^{\star} - \hat{\beta}_{F_{\mathbf{S}}}\|_{\Sigma_j} + c_0 \sqrt{\frac{d\zeta}{n}} \right) > 0.$$

Since $\beta \mapsto \|\theta_j^{\star} - \beta\|_{\Sigma_j}$ is continuous and the stochastic term in Proposition 1 does not depend on $\beta$, there exists $r_j > 0$ such that

$$\|\beta - \hat{\beta}_{F_{\mathbf{S}}}\|_2 < r_j \quad \Longrightarrow \quad \|\theta_j^{\star} - \beta\|_{\Sigma_j} + c_0 \sqrt{\frac{d\zeta}{n}} < \lambda.$$

Hence Proposition 1 implies $f_j(\beta) = g_j(\beta)$ throughout the Euclidean ball $B_{r_j}(\hat{\beta}_{F_{\mathbf{S}}})$. Let $r := \min_{j \in \mathbf{S}} r_j > 0$. Then

$$F_{\mathbf{S}}(\beta) = G_{\mathbf{S}}(\beta) \qquad \text{for all } \beta \in B_r(\hat{\beta}_{F_{\mathbf{S}}}).$$

On $\text{Range}(\Sigma_{\mathbf{S}})$, $F_{\mathbf{S}}$ is strictly convex, so $\hat{\beta}_{F_{\mathbf{S}}}$ is its unique minimizer in the prediction-relevant subspace and therefore a local minimizer of $G_{\mathbf{S}}$ as well. Finally, $G_{\mathbf{S}}$ is convex as a sum of infimal convolutions of convex functions, so any local minimizer is global. Hence $\hat{\beta}_{F_{\mathbf{S}}}$ minimizes $G_{\mathbf{S}}$. $\qquad\square$

## C.3. A Perturbation Bound for Outliers

The third ingredient controls the shift from the inlier-only objective $G_{\mathbf{S}}$ to the full objective $G = G_{\mathbf{S}} + L$, where $L$ denotes the aggregate outlier contribution introduced above, by treating this outlier term as a Lipschitz perturbation in the whitened geometry.

**Claim 1** (Local Inlier Curvature: Linear). *Suppose Assumption 1 holds and*

$$\lambda > 2(\sqrt{B} + 1)\delta + 2c_0\sqrt{\frac{d\zeta}{n}} + 2c_0\sqrt{\frac{d\zeta B}{N_{\mathbf{S}}}}.$$

*Conditioned on the event where Lemma 2, Proposition 5, and the taskwise noise bounds used in Proposition 1 hold for all $j \in \mathbf{S}$, every $\beta$ satisfying*

$$\|\bar{\beta} - \hat{\bar{\beta}}_{G_{\mathbf{S}}}\|_2 \leq \frac{\lambda}{2\sqrt{B}},$$

*belongs to the invariant region $R_j$ for all $j \in \mathbf{S}$. Consequently, in this neighborhood,*

$$\nabla^2 \bar{G}_{\mathbf{S}}(\bar{\beta}) = |\mathbf{S}|\mathbf{I}_d.$$

*Proof.* Under the stated event, $\hat{\beta}_{G_{\mathbf{S}}} = \hat{\beta}_{F_{\mathbf{S}}}$. Moreover, by Lemma 2 and the assumed lower bound on $\lambda$, for every $j \in \mathbf{S}$,

$$\|\theta_j^\star - \hat{\beta}_{G_{\mathbf{S}}}\|_{\boldsymbol{\Sigma}_j} + c_0\sqrt{\frac{d\zeta}{n}} \leq (\sqrt{B} + 1)\delta + c_0\sqrt{\frac{Bd\zeta}{N_{\mathbf{S}}}} + c_0\sqrt{\frac{d\zeta}{n}} < \frac{\lambda}{2}.$$

If $\|\bar{\beta} - \hat{\bar{\beta}}_{G_{\mathbf{S}}}\|_2 \leq \lambda/(2\sqrt{B})$, then Assumption 1 gives

$$\|\beta - \hat{\beta}_{G_{\mathbf{S}}}\|_{\boldsymbol{\Sigma}_j} \leq \sqrt{B}\|\beta - \hat{\beta}_{G_{\mathbf{S}}}\|_{\boldsymbol{\Sigma}_{\mathbf{S}}} = \sqrt{B}\|\bar{\beta} - \hat{\bar{\beta}}_{G_{\mathbf{S}}}\|_2 \leq \frac{\lambda}{2}.$$

Hence

$$\|\theta_j^\star - \beta\|_{\boldsymbol{\Sigma}_j} + c_0\sqrt{\frac{d\zeta}{n}} < \lambda,$$

so Proposition 1 implies $\beta \in R_j$ for all $j \in \mathbf{S}$. Since the stochastic term in Proposition 1 is precisely the taskwise noise bound included in the conditioning event and does not depend on $\beta$, this implication holds simultaneously throughout the displayed neighborhood.

Therefore $G_{\mathbf{S}}$ and $F_{\mathbf{S}}$ agree throughout this neighborhood. For the linear loss,

$$\nabla^2 \bar{f}_j(\bar{\beta}) = \boldsymbol{\Sigma}_{\mathbf{S}}^{-1/2}\boldsymbol{\Sigma}_j\boldsymbol{\Sigma}_{\mathbf{S}}^{-1/2},$$

which is independent of $\bar{\beta}$. Summing over $j \in \mathbf{S}$ and using $\boldsymbol{\Sigma}_{\mathbf{S}} = |\mathbf{S}|^{-1}\sum_{j \in \mathbf{S}}\boldsymbol{\Sigma}_j$ yields

$$\nabla^2 \bar{G}_{\mathbf{S}}(\bar{\beta}) = \nabla^2 \bar{F}_{\mathbf{S}}(\bar{\beta}) = \sum_{j \in \mathbf{S}}\boldsymbol{\Sigma}_{\mathbf{S}}^{-1/2}\boldsymbol{\Sigma}_j\boldsymbol{\Sigma}_{\mathbf{S}}^{-1/2} = |\mathbf{S}|\mathbf{I}_d.$$

$\square$

**Proposition 6** (Key Proposition: Linear Model). *Suppose Assumption 1 holds with $B < \frac{1}{3\varepsilon}$ and the regularization parameter satisfies:*

$$\lambda > 2(\sqrt{B} + 1)\delta + 2c_0\sqrt{\frac{d\zeta}{n}} + 2c_0\sqrt{\frac{d\zeta B}{N_{\mathbf{S}}}}.$$

*Then, with probability at least $1 - \kappa$, the following holds simultaneously for all inlier tasks $j \in \mathbf{S}$:*

$$\|\hat{\beta}_G - \theta_j^\star\|_{\boldsymbol{\Sigma}_j} \leq (\sqrt{B} + 1)\delta + c_0\sqrt{\frac{Bd\zeta}{N_{\mathbf{S}}}} + \frac{\varepsilon}{1 - \varepsilon}B\lambda.$$

*Furthermore, under the same event, we have*

$$\|\hat{\theta}_j - \hat{\beta}_G\|_{\boldsymbol{\Sigma}_j} = 0, \qquad \|\hat{\theta}_j - \theta_j^\star\|_{\boldsymbol{\Sigma}_j} = \|\hat{\beta}_G - \theta_j^\star\|_{\boldsymbol{\Sigma}_j}$$

*for all $j \in \mathbf{S}$.*

*Proof.* Using the pooled whitening notation introduced above, let $\hat{\bar{\beta}}_G := \boldsymbol{\Sigma}_{\mathbf{S}}^{1/2}\hat{\beta}_G$ and $\hat{\bar{\beta}}_{G_{\mathbf{S}}} := \boldsymbol{\Sigma}_{\mathbf{S}}^{1/2}\hat{\beta}_{G_{\mathbf{S}}}$. Minimizing $G$ is equivalent to minimizing $\bar{G} = \bar{G}_{\mathbf{S}} + \bar{L}$.

**Step 1: Pooling Range and Strong Convexity.** Let $\mathcal{E}_{\text{pool}}$ denote the event where Lemma 2, Proposition 5, and the taskwise invariant-region noise bounds hold. By the failure-budget convention above, the pooling-estimator event is allocated failure $\kappa/4$ and the taskwise noise events are allocated total failure $\kappa/4$, so $\mathbb{P}(\mathcal{E}_{\text{pool}}) \geq 1 - \kappa/2 \geq 1 - \kappa$. By Proposition 5, $\hat{\beta}_{F_{\mathbf{S}}} = \hat{\beta}_{G_{\mathbf{S}}}$ on this event. Moreover, Claim 1 shows that for all $\beta$ in the neighborhood

$$\|\bar{\beta} - \bar{\hat{\beta}}_{G_{\mathbf{S}}}\|_2 \leq \frac{\lambda}{2\sqrt{B}},$$

the transformed inlier objective has exact quadratic curvature:

$$\nabla^2 \bar{G}_{\mathbf{S}}(\bar{\beta}) = \nabla^2 \bar{F}_{\mathbf{S}}(\bar{\beta}) = |\mathbf{S}|\mathbf{I}_d.$$

**Step 2: Lipschitz Continuity of Outliers.** Since $\varepsilon < \frac{1}{3B} \leq \frac{1}{2B+1}$, Lemma 3 ensures that

$$\text{Lip}(\bar{L}) \leq \sum_{j \in \mathbf{S}^c} \text{Lip}(\bar{g}_j) \leq m\varepsilon\sqrt{B}\lambda.$$

We apply Lemma 11 with parameters $r = \frac{\lambda}{2\sqrt{B}}$, $\rho = |\mathbf{S}|$, and $\lambda' = m\varepsilon\sqrt{B}\lambda$. Since $m\varepsilon\sqrt{B}\lambda < (1-\varepsilon)m\frac{\lambda}{2\sqrt{B}} \leq \rho r$, this yields

$$\|\bar{\hat{\beta}}_G - \bar{\hat{\beta}}_{G_{\mathbf{S}}}\|_2 \leq \frac{m\varepsilon\sqrt{B}\lambda}{|\mathbf{S}|} \leq \frac{\varepsilon}{1-\varepsilon}\sqrt{B}\lambda.$$

Translating back to the original domain, for any $j \in \mathbf{S}$,

$$\|\hat{\beta}_G - \hat{\beta}_{G_{\mathbf{S}}}\|_{\mathbf{\Sigma}_j} \leq \frac{\varepsilon}{1-\varepsilon}B\lambda.$$

**Step 3: Total Error Bound.** Combining this perturbation bound with Lemma 2, on $\mathcal{E}_{\text{pool}}$ we obtain

$$
\begin{aligned}
\|\hat{\beta}_G - \theta_j^\star\|_{\mathbf{\Sigma}_j} &\leq \|\hat{\beta}_{G_{\mathbf{S}}} - \theta_j^\star\|_{\mathbf{\Sigma}_j} + \|\hat{\beta}_G - \hat{\beta}_{G_{\mathbf{S}}}\|_{\mathbf{\Sigma}_j} \\
&\leq (\sqrt{B}+1)\delta + c_0\sqrt{\frac{Bd\zeta}{N_{\mathbf{S}}}} + \frac{\varepsilon}{1-\varepsilon}B\lambda \\
&\leq (\sqrt{B}+1)\delta + c_0\sqrt{\frac{Bd\zeta}{N_{\mathbf{S}}}} + \frac{1}{2}\lambda.
\end{aligned}
\tag{10}
$$

**Step 4: Invariant Region Check.** To verify that $\hat{\beta}_G \in R_j$, Proposition 1 requires

$$\|\hat{\beta}_G - \theta_j^\star\|_{\mathbf{\Sigma}_j} + c_0\sqrt{\frac{d\zeta}{n}} < \lambda.$$

Substituting (10), the left-hand side is bounded by

$$(\sqrt{B}+1)\delta + c_0\sqrt{\frac{Bd\zeta}{N_{\mathbf{S}}}} + \frac{1}{2}\lambda + c_0\sqrt{\frac{d\zeta}{n}},$$

which is strictly smaller than $\lambda$ by the assumed lower bound on $\lambda$. Hence $\hat{\beta}_G \in R_j$ for all $j \in \mathbf{S}$. By Proposition 1, every corresponding taskwise minimizer satisfies $\|\hat{\theta}_j - \hat{\beta}_G\|_{\mathbf{\Sigma}_j} = 0$, and therefore $\|\hat{\theta}_j - \theta_j^\star\|_{\mathbf{\Sigma}_j} = \|\hat{\beta}_G - \theta_j^\star\|_{\mathbf{\Sigma}_j}$. $\qquad \square$

**Lemma 3.** *The transformed regularized loss function $\bar{g}_j(\cdot)$ is $\sqrt{B}\lambda_j$-Lipschitz.*

*Proof.* Let

$$M_j := \mathbf{\Sigma}_{\mathbf{S}}^{-1/2}\mathbf{\Sigma}_j\mathbf{\Sigma}_{\mathbf{S}}^{-1/2}.$$

In pooled-whitened coordinates,

$$\bar{g}_j(w) = \inf_z \left\{\bar{f}_j(z) + \lambda_j\|z - w\|_{M_j}\right\},$$

where the infimum is over the pooled-whitened subspace. By Assumption 1, $M_j \preceq B\mathbf{I}_d$. Fix $w, w'$ and $\eta > 0$, and choose $z_\eta$ such that

$$\bar{g}_j(w') + \eta > \bar{f}_j(z_\eta) + \lambda_j \|z_\eta - w'\|_{M_j}.$$

Then

$$\bar{g}_j(w) - \bar{g}_j(w') \leq \lambda_j \big( \|z_\eta - w\|_{M_j} - \|z_\eta - w'\|_{M_j} \big) + \eta$$
$$\leq \lambda_j \|w - w'\|_{M_j} + \eta \leq \sqrt{B} \lambda_j \|w - w'\|_2 + \eta.$$

Letting $\eta \downarrow 0$ and exchanging $w, w'$ proves the claim. $\qquad\square$

### C.4. Proof of Theorem 2

*Proof.* Let $\mathcal{E}_{\text{safe}}$ be the event obtained by applying the uniform form of Proposition 2 to each task with failure probability $\kappa/(2m)$ and taking a union bound. Then $\mathbb{P}(\mathcal{E}_{\text{safe}}) \geq 1 - \kappa/2$, and on this event, with $\lambda = q\sqrt{d\zeta/n}$ and $q \gtrsim 1$, for every task $j \in [m]$,

$$\|\hat{\theta}_j - \theta_j^\star\|_{\boldsymbol{\Sigma}_j}^2 \lesssim \frac{d\zeta}{n} + \lambda^2 \lesssim q^2 \frac{d\zeta}{n}.$$

This proves part (a), including all outlier tasks, without any balancedness or relatedness assumption.

It remains to prove the sharper inlier bound in part (b). We analyze the estimator in two regimes depending on the magnitude of the task heterogeneity $\delta$ relative to the regularization parameter $\lambda = q\sqrt{\frac{d\zeta}{n}}$. Throughout this part, $B \lesssim \min(1/\varepsilon, m)$; choosing the hidden universal constant small enough gives the numerical conditions needed in Proposition 6, and also $N_{\mathbf{S}} = |\mathbf{S}|n \asymp mn$.

**Case 1: Proposition 6 Applies.** Suppose

$$\lambda > 2(\sqrt{B} + 1)\delta + 2c_0\sqrt{\frac{d\zeta}{n}} + 2c_0\sqrt{\frac{Bd\zeta}{N_{\mathbf{S}}}}.$$

Then Proposition 6 applies. Its proof constructs a transfer event $\mathcal{E}_{\text{transfer}}$ with $\mathbb{P}(\mathcal{E}_{\text{transfer}}) \geq 1 - \kappa/2$ under the present good-event budget. On $\mathcal{E}_{\text{safe}} \cap \mathcal{E}_{\text{transfer}}$, which has probability at least $1 - \kappa$, for all inlier tasks $j \in \mathbf{S}$:

$$\|\hat{\theta}_j - \theta_j^\star\|_{\boldsymbol{\Sigma}_j} = \|\hat{\beta}_G - \theta_j^\star\|_{\boldsymbol{\Sigma}_j}$$
$$\leq (\sqrt{B} + 1)\delta + c_0\sqrt{\frac{Bd\zeta}{N_{\mathbf{S}}}} + \frac{\varepsilon}{1 - \varepsilon} B\lambda.$$

Squaring and using $(a + b + c)^2 \lesssim a^2 + b^2 + c^2$, we obtain

$$\|\hat{\theta}_j - \theta_j^\star\|_{\boldsymbol{\Sigma}_j}^2 \lesssim B\delta^2 + \frac{Bd\zeta}{mn} + \left(\frac{\varepsilon}{1 - \varepsilon} B\right)^2 \lambda^2$$
$$\lesssim \min\left(B\delta^2, q^2\frac{d\zeta}{n}\right) + \frac{Bd\zeta}{mn} + q^2 B^2 \varepsilon^2 \frac{d\zeta}{n}.$$

The last inequality uses the displayed condition, which implies $B\delta^2 \lesssim \lambda^2 = q^2 d\zeta/n$.

**Case 2: Proposition 6 Does Not Apply.** If the displayed condition fails, then

$$\lambda \leq 2(\sqrt{B} + 1)\delta + 2c_0\sqrt{\frac{d\zeta}{n}} + 2c_0\sqrt{\frac{Bd\zeta}{N_{\mathbf{S}}}}.$$

Since $q \gtrsim 1$ is chosen large enough that $q \geq 8c_0$, we can absorb the middle term into the left-hand side and obtain

$$\frac{\lambda}{2} \lesssim \sqrt{B}\delta + \sqrt{\frac{Bd\zeta}{N_{\mathbf{S}}}},$$

where we used $B \geq 1$. Hence at least one of the following alternatives must hold:

$$\lambda \lesssim \sqrt{B}\delta \quad \text{or} \quad \lambda \lesssim \sqrt{\frac{Bd\zeta}{N_{\mathbf{S}}}}.$$

Equivalently,

$$q^2 \frac{d\zeta}{n} \lesssim B\delta^2 \quad \text{or} \quad q^2 \frac{d\zeta}{n} \lesssim \frac{Bd\zeta}{N_{\mathbf{S}}}.$$

On the already-defined event $\mathcal{E}_{\text{safe}}$,

$$\|\hat{\theta}_j - \theta_j^\star\|_{\boldsymbol{\Sigma}_j}^2 \lesssim \frac{d\zeta}{n} + \lambda^2 \lesssim q^2 \frac{d\zeta}{n}.$$

The preceding alternatives show that this safety bound is controlled by either the heterogeneity term or the pooling term, and therefore by the right-hand side claimed in part (b).

**Conclusion.** Combining the two cases, we conclude that with probability at least $1 - \kappa$, for all $j \in \mathbf{S}$,

$$\|\hat{\theta}_j - \theta_j^\star\|_{\boldsymbol{\Sigma}_j}^2 \lesssim \frac{Bd\zeta}{mn} + \min\left(B\delta^2, q^2 \frac{d\zeta}{n}\right) + q^2 B^2 \varepsilon^2 \frac{d\zeta}{n}.$$

The outlier tasks are already covered by the universal safety bound proved at the beginning. $\qquad\square$

## D. Proofs for the GLM

We now turn to the genuinely multi-task part of the GLM proof. The single-task $n$-sample GLM loss geometry was isolated in Section B; this section focuses on the inlier pooling estimator, the outlier perturbation step, and the theorem and population-risk proofs.

### D.1. Multi-Task Setup and Notation

Throughout the GLM transfer proof, all quantities involving the inlier average $\boldsymbol{\Sigma}_{\mathbf{S}}$ are interpreted after projecting onto $\text{Range}(\boldsymbol{\Sigma}_{\mathbf{S}})$. On this projected subspace, the displayed inverses are ordinary inverses. This entails no loss for prediction risk: Assumption 1 implies that any direction in $\text{Null}(\boldsymbol{\Sigma}_{\mathbf{S}})$ is also in $\text{Null}(\boldsymbol{\Sigma}_j)$ for every inlier task $j \in \mathbf{S}$. Moreover, since the GLM feasible set is $\mathbf{C} = \mathbf{B}(0, \xi)$, orthogonal projection onto $\text{Range}(\boldsymbol{\Sigma}_{\mathbf{S}})$ keeps feasible parameters inside $\mathbf{C}$. Thus, in the proof below, we work on this projected subspace and treat $\boldsymbol{\Sigma}_{\mathbf{S}}$ as full-rank.

For any subset of tasks $\mathbf{I} \subseteq [m]$, let

$$N_{\mathbf{I}} := \sum_{j \in \mathbf{I}} n_j, \qquad \mathbf{V}_{\mathbf{I}} := \sum_{j \in \mathbf{I}} \mathbf{V}_j, \qquad \boldsymbol{\Sigma}_{\mathbf{I}} := \frac{1}{N_{\mathbf{I}}} \mathbf{V}_{\mathbf{I}}.$$

In the balanced case analyzed in the GLM theorem, $n_j = n$ and hence $N_{\mathbf{S}} = n|\mathbf{S}|$ and $\boldsymbol{\Sigma}_{\mathbf{S}} = |\mathbf{S}|^{-1} \sum_{j \in \mathbf{S}} \boldsymbol{\Sigma}_j$. Define $\tau := \alpha_u/\alpha_\ell$ and write $m = \psi'$ for the GLM mean function. For each task, define

$$f_j(\beta) := \frac{1}{n} \sum_{i=1}^n \left(-y_{ji} x_{ji}^\top \beta + \psi(x_{ji}^\top \beta)\right)$$

and its matrix-weighted infimal convolution over $\mathbf{C}$,

$$g_j(\beta) := \left(f_j \star \lambda \|\cdot\|_{\boldsymbol{\Sigma}_j}\right)(\beta) = \inf_{u \in \mathbf{C}} \left\{f_j(u) + \lambda \|u - \beta\|_{\boldsymbol{\Sigma}_j}\right\}.$$

We further write

$$F_{\mathbf{S}}(\beta) := \sum_{j \in \mathbf{S}} f_j(\beta), \qquad G_{\mathbf{S}}(\beta) := \sum_{j \in \mathbf{S}} g_j(\beta), \qquad L(\beta) := \sum_{j \in \mathbf{S}^c} g_j(\beta), \qquad G(\beta) := G_{\mathbf{S}}(\beta) + L(\beta).$$

Let $\hat{\beta}_{F_{\mathbf{S}}}, \hat{\beta}_{G_{\mathbf{S}}}$, and $\hat{\beta}_G$ denote minimizers of these objectives over $\mathbf{C}$. As before, define the invariant region

$$R_j := \{\beta \in \mathbf{C} : f_j(\beta) = g_j(\beta)\}, \qquad j \in [m].$$

**Pooled Whitening Notation.** On $\mathrm{Range}(\boldsymbol{\Sigma}_{\mathbf{S}})$, we define the pooled whitening map by

$$\bar{v} := \boldsymbol{\Sigma}_{\mathbf{S}}^{1/2} v, \qquad v \in \mathbb{R}^d,$$

and the pooled-whitened covariates by

$$\tilde{x}_{ji} := \boldsymbol{\Sigma}_{\mathbf{S}}^{-1/2} x_{ji}.$$

The transformed feasible set is

$$\bar{\mathbf{C}} := \{\boldsymbol{\Sigma}_{\mathbf{S}}^{1/2} v : v \in \mathbf{C}\}.$$

For any objective $h$ above, we define its pooled-whitened version by

$$\bar{h}(\bar{v}) := h(\boldsymbol{\Sigma}_{\mathbf{S}}^{-1/2} \bar{v}),$$

where $\boldsymbol{\Sigma}_{\mathbf{S}}^{-1/2}$ denotes the inverse on $\mathrm{Range}(\boldsymbol{\Sigma}_{\mathbf{S}})$. Thus $\bar{\theta}, \bar{\beta}, \bar{\theta}^{\star}, \bar{\theta}_j^{\star}, \bar{\hat{\beta}}_{F_{\mathbf{S}}}, \bar{\hat{\beta}}_{G_{\mathbf{S}}}$, and $\bar{\hat{\beta}}_G$ are all instances of the same pooled whitening map. In particular, we write $\bar{F}_{\mathbf{S}}, \bar{G}_{\mathbf{S}}, \bar{L}$, and $\bar{G}$ for the transformed objectives.

### D.2. The Inlier Pooling Estimator

We now derive concentration bounds for the pooling estimator in Generalized Linear Models.

**Lemma 4** (Error Bound of Pooling Estimator: GLM). *With probability at least $1 - \frac{\kappa}{4}$, the minimizer $\hat{\beta}_{F_{\mathbf{S}}}$ of the inlier loss $F_{\mathbf{S}}$ satisfies:*

$$\|\hat{\beta}_{F_{\mathbf{S}}} - \theta^{\star}\|_{\boldsymbol{\Sigma}_{\mathbf{S}}} \leq \frac{c_0}{\alpha_\ell} \sqrt{\frac{d\zeta}{N_{\mathbf{S}}}} + \tau \sqrt{B} \delta.$$

*Consequently, for any $j \in \mathbf{S}$, we have the task-specific bound:*

$$\|\hat{\beta}_{F_{\mathbf{S}}} - \theta^{\star}\|_{\boldsymbol{\Sigma}_j} \leq \frac{c_0}{\alpha_\ell} \sqrt{\frac{Bd\zeta}{N_{\mathbf{S}}}} + B\tau \delta.$$

*Proof.* Using the pooled whitening notation introduced above, let $\bar{\hat{\beta}}_{F_{\mathbf{S}}} := \boldsymbol{\Sigma}_{\mathbf{S}}^{1/2} \hat{\beta}_{F_{\mathbf{S}}}$.

**Step 1: Gradient Bound via Strong Convexity.** The Hessian of the transformed loss satisfies $\nabla^2 \bar{F}_{\mathbf{S}}(\bar{\beta}) \succeq \alpha_\ell |\mathbf{S}| \mathbf{I}_d$, so $\bar{F}_{\mathbf{S}}$ is $\mu$-strongly convex with $\mu = \alpha_\ell |\mathbf{S}|$. Let $\bar{\hat{\beta}}_{F_{\mathbf{S}}}$ be the minimizer of $\bar{F}_{\mathbf{S}}$. Strong convexity implies

$$\left\langle \nabla \bar{F}_{\mathbf{S}}(\bar{\theta}^{\star}) - \nabla \bar{F}_{\mathbf{S}}(\bar{\hat{\beta}}_{F_{\mathbf{S}}}), \bar{\theta}^{\star} - \bar{\hat{\beta}}_{F_{\mathbf{S}}} \right\rangle \geq \mu \|\bar{\theta}^{\star} - \bar{\hat{\beta}}_{F_{\mathbf{S}}}\|_2^2.$$

Since $\bar{\hat{\beta}}_{F_{\mathbf{S}}}$ minimizes $\bar{F}_{\mathbf{S}}$ over the convex set $\bar{\mathbf{C}}$ and $\bar{\theta}^{\star} \in \bar{\mathbf{C}}$, the constrained first-order condition gives

$$\left\langle \nabla \bar{F}_{\mathbf{S}}(\bar{\hat{\beta}}_{F_{\mathbf{S}}}), \bar{\theta}^{\star} - \bar{\hat{\beta}}_{F_{\mathbf{S}}} \right\rangle \geq 0.$$

Therefore Cauchy–Schwarz yields

$$\|\bar{\hat{\beta}}_{F_{\mathbf{S}}} - \bar{\theta}^{\star}\|_2 \leq \frac{1}{\alpha_\ell |\mathbf{S}|} \|\nabla \bar{F}_{\mathbf{S}}(\bar{\theta}^{\star})\|_2.$$

**Step 2: Gradient Decomposition.** We calculate the gradient at the centroid $\bar{\theta}^{\star}$:

$$\nabla \bar{F}_{\mathbf{S}}(\bar{\theta}^{\star}) = \frac{1}{n} \sum_{j \in \mathbf{S}} \sum_{i=1}^n \left( m(\tilde{x}_{ji}^{\top} \bar{\theta}^{\star}) - y_{ji} \right) \tilde{x}_{ji}$$

$$= \underbrace{-\frac{1}{n} \sum_{j \in \mathbf{S}} \sum_{i=1}^n \varepsilon_{ji} \tilde{x}_{ji}}_{\text{Noise}} + \underbrace{\frac{1}{n} \sum_{j \in \mathbf{S}} \sum_{i=1}^n \left( m(x_{ji}^{\top} \theta^{\star}) - m(x_{ji}^{\top} \theta_j^{\star}) \right) \tilde{x}_{ji}}_{\text{Bias}},$$

where $\tilde{x}_{ji} = \boldsymbol{\Sigma}_{\mathbf{S}}^{-1/2} x_{ji}$ and $m = \psi'$ is the mean function.

**Step 3: Bounding the Noise Term.** The noise term is a sum of independent sub-Gaussian random vectors. Since

$$\sum_{j \in \mathbf{S}} \sum_{i=1}^{n} \tilde{x}_{ji} \tilde{x}_{ji}^\top = \mathbf{\Sigma}_{\mathbf{S}}^{-1/2} \left( \sum_{j \in \mathbf{S}} \mathbf{X}_j^\top \mathbf{X}_j \right) \mathbf{\Sigma}_{\mathbf{S}}^{-1/2} = N_{\mathbf{S}} \mathbf{I}_d,$$

the covariance of the sum is isotropic. Applying the Hanson-Wright inequality with failure level $\kappa/4$:

$$\|\text{Noise}\|_2 \le c_0 \frac{\sqrt{N_{\mathbf{S}} d\zeta}}{n} = c_0 \sqrt{\frac{|\mathbf{S}| d\zeta}{n}}.$$

**Step 4: Bounding the Bias Term.** By the Mean Value Theorem, $m(x_{ji}^\top \theta^\star) - m(x_{ji}^\top \theta_j^\star) = u_{ji} x_{ji}^\top (\theta^\star - \theta_j^\star)$, where $u_{ji} \in [\alpha_\ell, \alpha_u]$. Let $\mathbf{Q}_j := \frac{1}{n} \sum_{i=1}^{n} u_{ji} x_{ji} x_{ji}^\top$. Note that $\alpha_\ell \mathbf{\Sigma}_j \preceq \mathbf{Q}_j \preceq \alpha_u \mathbf{\Sigma}_j$. The bias term can be written as:

$$\text{Bias} = \sum_{j \in \mathbf{S}} \mathbf{\Sigma}_{\mathbf{S}}^{-1/2} \mathbf{Q}_j (\theta^\star - \theta_j^\star) = \sum_{j \in \mathbf{S}} \mathbf{\Sigma}_{\mathbf{S}}^{-1/2} \mathbf{Q}_j \mathbf{\Sigma}_{\mathbf{S}}^{-1/2} \mathbf{\Sigma}_{\mathbf{S}}^{1/2} (\theta^\star - \theta_j^\star).$$

We bound the norm of each term in the sum:

$$\begin{aligned}
\|\mathbf{\Sigma}_{\mathbf{S}}^{-1/2} \mathbf{Q}_j (\theta^\star - \theta_j^\star)\|_2 &\le \|\mathbf{\Sigma}_{\mathbf{S}}^{-1/2} \mathbf{Q}_j^{1/2}\|_{\text{op}} \|\mathbf{Q}_j^{1/2} (\theta^\star - \theta_j^\star)\|_2 \\
&\le \|\mathbf{\Sigma}_{\mathbf{S}}^{-1/2} \mathbf{Q}_j^{1/2}\|_{\text{op}} \sqrt{\alpha_u} \|\theta^\star - \theta_j^\star\|_{\mathbf{\Sigma}_j}.
\end{aligned}$$

Using $\mathbf{Q}_j \preceq \alpha_u \mathbf{\Sigma}_j$ and Assumption 1 ($\mathbf{\Sigma}_j \preceq B \mathbf{\Sigma}_{\mathbf{S}}$), we have:

$$\|\mathbf{\Sigma}_{\mathbf{S}}^{-1/2} \mathbf{Q}_j^{1/2}\|_{\text{op}}^2 = \|\mathbf{\Sigma}_{\mathbf{S}}^{-1/2} \mathbf{Q}_j \mathbf{\Sigma}_{\mathbf{S}}^{-1/2}\|_{\text{op}} \le \alpha_u \|\mathbf{\Sigma}_{\mathbf{S}}^{-1/2} \mathbf{\Sigma}_j \mathbf{\Sigma}_{\mathbf{S}}^{-1/2}\|_{\text{op}} \le \alpha_u B.$$

Moreover, by the global covariate normalization $\|\mathbf{\Sigma}_j\|_{\text{op}} \le 1$ from the Regularity Assumptions and the task-relatedness condition $\|\theta_j^\star - \theta^\star\|_2 \le \delta$, we have

$$\|\theta^\star - \theta_j^\star\|_{\mathbf{\Sigma}_j} \le \|\theta^\star - \theta_j^\star\|_2 \le \delta.$$

Thus, $\|\text{Bias term}_j\|_2 \le \sqrt{\alpha_u B} \sqrt{\alpha_u} \delta = \alpha_u \sqrt{B} \delta$. Summing over $j \in \mathbf{S}$:

$$\|\text{Bias}\|_2 \le \sum_{j \in \mathbf{S}} \alpha_u \sqrt{B} \delta = |\mathbf{S}| \alpha_u \sqrt{B} \delta.$$

**Step 5: Final Bound.** Combining the bounds:

$$\begin{aligned}
\|\hat{\beta}_{F_{\mathbf{S}}} - \theta^\star\|_{\mathbf{\Sigma}_{\mathbf{S}}} &\le \frac{1}{\alpha_\ell |\mathbf{S}|} \left( c_0 \sqrt{\frac{|\mathbf{S}| d\zeta}{n}} + |\mathbf{S}| \alpha_u \sqrt{B} \delta \right) \\
&= \frac{c_0}{\alpha_\ell} \sqrt{\frac{d\zeta}{n |\mathbf{S}|}} + \frac{\alpha_u}{\alpha_\ell} \sqrt{B} \delta \\
&= \frac{c_0}{\alpha_\ell} \sqrt{\frac{d\zeta}{N_{\mathbf{S}}}} + \tau \sqrt{B} \delta.
\end{aligned}$$

The task-specific bound follows from $\| \cdot \|_{\mathbf{\Sigma}_j} \le \sqrt{B} \| \cdot \|_{\mathbf{\Sigma}_{\mathbf{S}}}$. $\qquad \square$

**Proposition 7** (Equivalence of Pooling Estimators: GLM). *Suppose the regularization parameter satisfies the following lower bound:*

$$\lambda > \alpha_u (B\tau + 1) \delta + \frac{c_0 \alpha_u}{\alpha_\ell} \sqrt{\frac{B d\zeta}{N_{\mathbf{S}}}} + c_0 \sqrt{\frac{d\zeta}{n}}.$$

*Then, with probability at least $1 - \frac{\kappa}{2}$, we have $\hat{\beta}_{G_{\mathbf{S}}} = \hat{\beta}_{F_{\mathbf{S}}}$. Furthermore, $\hat{\beta}_{F_{\mathbf{S}}} \in R_j$ holds simultaneously for all $j \in \mathbf{S}$.*

*Proof.* Let $\mathcal{E}_{\text{pool}}^{\text{glm}}$ be the event from Lemma 4; it has probability at least $1 - \kappa/4$. Let $\mathcal{E}_{\text{inv}}^{\text{glm}}$ be the event that the taskwise GLM invariant-region noise bound from Proposition 3 holds for every $j \in \mathbf{S}$, each invoked with failure level $\kappa/(4m)$. Then $\mathbb{P}(\mathcal{E}_{\text{pool}}^{\text{glm}} \cap \mathcal{E}_{\text{inv}}^{\text{glm}}) \geq 1 - \kappa/2$.

The goal is to show that the unregularized pooling estimator $\hat{\beta}_{F_{\mathbf{S}}}$ minimizes the regularized objective $G_{\mathbf{S}}(\beta) = \sum_{j \in \mathbf{S}} g_j(\beta)$. For GLMs, $\nabla^2 f_j(\beta) = \frac{1}{n} \mathbf{X}_j^\top \mathbf{D}_j(\beta) \mathbf{X}_j \succeq \alpha_\ell \mathbf{\Sigma}_j$, where $\mathbf{D}_j(\beta)$ is the diagonal matrix of $\psi''$-values. Hence

$$\nabla^2 F_{\mathbf{S}}(\beta) = \sum_{j \in \mathbf{S}} \nabla^2 f_j(\beta) \succeq \alpha_\ell \sum_{j \in \mathbf{S}} \mathbf{\Sigma}_j = \alpha_\ell |\mathbf{S}| \mathbf{\Sigma}_{\mathbf{S}} \succ 0,$$

so $F_{\mathbf{S}}$ is strictly convex and $\hat{\beta}_{F_{\mathbf{S}}}$ is its unique minimizer.

By Proposition 3, the condition $\hat{\beta}_{F_{\mathbf{S}}} \in R_j$ is satisfied if:

$$\lambda > \alpha_u \|\theta_j^\star - \hat{\beta}_{F_{\mathbf{S}}}\|_{\mathbf{\Sigma}_j} + c_0 \sqrt{\frac{d\zeta}{n}}. \tag{11}$$

We bound the error term $\|\theta_j^\star - \hat{\beta}_{F_{\mathbf{S}}}\|_{\mathbf{\Sigma}_j}$ using the triangle inequality:

$$\|\theta_j^\star - \hat{\beta}_{F_{\mathbf{S}}}\|_{\mathbf{\Sigma}_j} \leq \|\theta_j^\star - \theta^\star\|_{\mathbf{\Sigma}_j} + \|\theta^\star - \hat{\beta}_{F_{\mathbf{S}}}\|_{\mathbf{\Sigma}_j}.$$

For the first term, using the bounded covariate assumption implies $\|\theta_j^\star - \theta^\star\|_{\mathbf{\Sigma}_j} \leq \delta$. For the second term, we apply Lemma 4:

$$\|\hat{\beta}_{F_{\mathbf{S}}} - \theta^\star\|_{\mathbf{\Sigma}_j} \leq B\tau\delta + \frac{c_0}{\alpha_\ell} \sqrt{\frac{Bd\zeta}{N_{\mathbf{S}}}}.$$

Combining these, the condition (11) becomes:

$$\lambda > \alpha_u \left( \delta + B\tau\delta + \frac{c_0}{\alpha_\ell} \sqrt{\frac{Bd\zeta}{N_{\mathbf{S}}}} \right) + c_0 \sqrt{\frac{d\zeta}{n}}.$$

This is exactly the assumed lower bound on $\lambda$. Thus, on $\mathcal{E}_{\text{pool}}^{\text{glm}} \cap \mathcal{E}_{\text{inv}}^{\text{glm}}$, $\hat{\beta}_{F_{\mathbf{S}}}$ lies in the invariant region $R_j$ for all $j \in \mathbf{S}$. For each $j \in \mathbf{S}$, define the positive slack

$$\Delta_j^{\text{glm}} := \lambda - \left( \alpha_u \|\theta_j^\star - \hat{\beta}_{F_{\mathbf{S}}}\|_{\mathbf{\Sigma}_j} + c_0 \sqrt{\frac{d\zeta}{n}} \right) > 0.$$

Since $\beta \mapsto \|\theta_j^\star - \beta\|_{\mathbf{\Sigma}_j}$ is continuous and the stochastic term in Proposition 3 does not depend on $\beta$, there exists $r_j > 0$ such that

$$\|\beta - \hat{\beta}_{F_{\mathbf{S}}}\|_2 < r_j \implies \alpha_u \|\theta_j^\star - \beta\|_{\mathbf{\Sigma}_j} + c_0 \sqrt{\frac{d\zeta}{n}} < \lambda.$$

Hence Proposition 3 implies $f_j(\beta) = g_j(\beta)$ throughout the Euclidean ball $B_{r_j}(\hat{\beta}_{F_{\mathbf{S}}})$. Let $r := \min_{j \in \mathbf{S}} r_j > 0$. Then

$$F_{\mathbf{S}}(\beta) = G_{\mathbf{S}}(\beta) \qquad \text{for all } \beta \in B_r(\hat{\beta}_{F_{\mathbf{S}}}).$$

Since $\hat{\beta}_{F_{\mathbf{S}}}$ is the unique minimizer of $F_{\mathbf{S}}$, it is a local minimizer of $G_{\mathbf{S}}$ as well. Finally, $G_{\mathbf{S}}$ is convex as a sum of infimal convolutions of convex functions, so any local minimizer is global. Hence $\hat{\beta}_{G_{\mathbf{S}}} = \hat{\beta}_{F_{\mathbf{S}}}$. $\qquad \square$

### D.3. A Perturbation Bound for Outliers

**Proposition 8** (Key Proposition: GLM). *Suppose Assumption 1 holds with $B < \frac{1}{4\tau\varepsilon}$ and the regularization parameter satisfies:*

$$\lambda > 2\alpha_u(\tau B + 1)\delta + 2c_0 \sqrt{\frac{d\zeta}{n}} + 2\tau c_0 \sqrt{\frac{Bd\zeta}{N_{\mathbf{S}}}}.$$

*Then, with probability at least $1 - \kappa$, the following holds simultaneously for all $j \in \mathbf{S}$:*

$$\|\hat{\beta}_G - \theta_j^\star\|_{\boldsymbol{\Sigma}_j} \leq (\tau B + 1)\delta + \frac{c_0}{\alpha_\ell}\sqrt{\frac{Bd\zeta}{N_{\mathbf{S}}}} + \frac{2\varepsilon B\lambda}{\alpha_\ell}.$$

*Furthermore, under the same event, we have*

$$\|\hat{\theta}_j - \hat{\beta}_G\|_{\boldsymbol{\Sigma}_j} = 0, \qquad \|\hat{\theta}_j - \theta_j^\star\|_{\boldsymbol{\Sigma}_j} = \|\hat{\beta}_G - \theta_j^\star\|_{\boldsymbol{\Sigma}_j}$$

*for all $j \in \mathbf{S}$.*

*Proof.* Using the pooled whitening notation introduced above, let $\bar{\hat{\beta}}_G := \boldsymbol{\Sigma}_{\mathbf{S}}^{1/2}\hat{\beta}_G$ and $\bar{\hat{\beta}}_{G_{\mathbf{S}}} := \boldsymbol{\Sigma}_{\mathbf{S}}^{1/2}\hat{\beta}_{G_{\mathbf{S}}}$ denote the minimizers of the transformed global objective $\bar{G} = \bar{G}_{\mathbf{S}} + \bar{L}$ and the inlier objective $\bar{G}_{\mathbf{S}}$, respectively. Let $\mathcal{E}_{\mathrm{glm}}$ be the good event constructed in Proposition 7: the GLM pooling error holds and the taskwise GLM invariant-region noise bounds hold for all $j \in \mathbf{S}$. The pooling part is allocated failure $\kappa/4$, and the taskwise invariant-region noise bounds are allocated failure $\kappa/(4m)$ per task, so

$$\mathbb{P}(\mathcal{E}_{\mathrm{glm}}) \geq 1 - \frac{\kappa}{2} \geq 1 - \kappa.$$

**Step 1: Pooling Estimator and Strong Convexity.** By Proposition 7, the pooling estimator coincides with the regularized inlier estimator, i.e., $\hat{\beta}_{F_{\mathbf{S}}} = \hat{\beta}_{G_{\mathbf{S}}}$. Furthermore, invoking Claim 2, for all $\beta$ in the neighborhood $\|\bar{\beta} - \bar{\hat{\beta}}_{G_{\mathbf{S}}}\|_2 \leq \frac{\lambda}{2\alpha_u\sqrt{B}}$, the transformed inlier loss is strongly convex:

$$\nabla^2\bar{G}_{\mathbf{S}}(\bar{\beta}) \succeq \alpha_\ell|\mathbf{S}|\mathbf{I}_d.$$

**Step 2: Perturbation via Outliers.** We bound the Lipschitz constant of the transformed outlier term $\bar{L}$. Using the assumption $B < \frac{1}{4\tau\varepsilon}$ and Lemma 5:

$$\mathrm{Lip}(\bar{L}) \leq m\varepsilon\sqrt{B}\lambda.$$

We apply the constrained perturbation lemma (Lemma 12) over the convex set $\bar{\mathbf{C}}$, with $f = \bar{G}_{\mathbf{S}}$, $g = \bar{L}$, $x_0 = \bar{\hat{\beta}}_{G_{\mathbf{S}}}$, $\rho = \alpha_\ell|\mathbf{S}|$, and $r = \frac{\lambda}{2\alpha_u\sqrt{B}}$. Since $|\mathbf{S}| \geq (1-\varepsilon)m$, and we may assume $B \geq 1$ without loss of generality, the condition $B < \frac{1}{4\tau\varepsilon}$ implies $\varepsilon \leq \frac{1}{4}$ and hence $\frac{\varepsilon}{1-\varepsilon} \leq \frac{1}{3}$. Moreover,

$$\frac{\mathrm{Lip}(\bar{L})}{\rho r} = \frac{2\alpha_u\varepsilon B}{\alpha_\ell} \cdot \frac{m}{|\mathbf{S}|} \leq \frac{2\tau\varepsilon B}{1-\varepsilon} < 1,$$

so the condition $\mathrm{Lip}(\bar{L}) < \rho r$ holds. Applying the lemma yields:

$$\|\bar{\hat{\beta}}_G - \bar{\hat{\beta}}_{G_{\mathbf{S}}}\|_2 \leq \frac{m\varepsilon\sqrt{B}\lambda}{\alpha_\ell|\mathbf{S}|} \leq \frac{\varepsilon}{1-\varepsilon}\frac{\sqrt{B}\lambda}{\alpha_\ell} \leq \frac{2\varepsilon\sqrt{B}\lambda}{\alpha_\ell},$$

where we used $|\mathbf{S}| \geq (1-\varepsilon)m$ and $\varepsilon \leq 1/4$. Converting back to the $\boldsymbol{\Sigma}_j$-norm using $\boldsymbol{\Sigma}_j \preceq B\boldsymbol{\Sigma}_{\mathbf{S}}$, we obtain:

$$\|\hat{\beta}_G - \hat{\beta}_{G_{\mathbf{S}}}\|_{\boldsymbol{\Sigma}_j} \leq \sqrt{B}\|\bar{\hat{\beta}}_G - \bar{\hat{\beta}}_{G_{\mathbf{S}}}\|_2 \leq \frac{2\varepsilon B\lambda}{\alpha_\ell}.$$

**Step 3: Total Error and Invariant Region.** Combining the perturbation bound with the pooling error (Lemma 4):

$$\|\hat{\beta}_G - \theta_j^\star\|_{\boldsymbol{\Sigma}_j} \leq \|\hat{\beta}_{G_{\mathbf{S}}} - \theta_j^\star\|_{\boldsymbol{\Sigma}_j} + \|\hat{\beta}_G - \hat{\beta}_{G_{\mathbf{S}}}\|_{\boldsymbol{\Sigma}_j}$$

$$\leq (\tau B + 1)\delta + \frac{c_0}{\alpha_\ell}\sqrt{\frac{Bd\zeta}{N_{\mathbf{S}}}} + \frac{2\varepsilon B\lambda}{\alpha_\ell}. \tag{12}$$

To prove the zero-seminorm personalization statement, we check the condition for the invariant region (Proposition 3). We need $\lambda > \alpha_u \|\hat{\beta}_G - \theta_j^\star\|_{\Sigma_j} + c_0\sqrt{\frac{d\zeta}{n}}$. Substituting the error bound from (12) into the right-hand side:

$$\text{RHS} \leq \alpha_u \left[ (\tau B + 1)\delta + \frac{c_0}{\alpha_\ell}\sqrt{\frac{Bd\zeta}{N_{\mathbf{S}}}} + \frac{2\varepsilon B\lambda}{\alpha_\ell} \right] + c_0\sqrt{\frac{d\zeta}{n}}$$

$$= \alpha_u(\tau B + 1)\delta + \tau c_0\sqrt{\frac{Bd\zeta}{N_{\mathbf{S}}}} + c_0\sqrt{\frac{d\zeta}{n}} + 2\tau\varepsilon B\lambda.$$

By the assumption $B < \frac{1}{4\tau\varepsilon}$, we have $2\tau\varepsilon B \leq \frac{1}{2}$. Thus, the $\lambda$ term on the right-hand side is at most $\frac{1}{2}\lambda$. The assumed lower bound on $\lambda$ then ensures that the remaining deterministic and stochastic terms are also bounded by $\frac{1}{2}\lambda$.

Therefore, on the event $\mathcal{E}_{\text{glm}}$, the condition holds. Proposition 3 then implies that every task-specific minimizer at center $\hat{\beta}_G$ satisfies $\|\hat{\theta}_j - \hat{\beta}_G\|_{\Sigma_j} = 0$ for all $j \in \mathbf{S}$. Hence the corresponding prediction-seminorm errors coincide, and the bound follows directly from (12). $\qquad\square$

**Lemma 5.** *The transformed regularized loss function $\bar{g}_j$ is $\sqrt{B}\lambda_j$-Lipschitz on the domain $\bar{\mathbf{C}} := \{\Sigma_{\mathbf{S}}^{1/2}\beta \mid \beta \in \mathbf{C}\}$.*

*Proof.* Let

$$M_j := \Sigma_{\mathbf{S}}^{-1/2}\Sigma_j\Sigma_{\mathbf{S}}^{-1/2}.$$

For $w \in \bar{\mathbf{C}}$, the constrained infimal convolution becomes

$$\bar{g}_j(w) = \inf_{z \in \bar{\mathbf{C}}} \left\{ \bar{f}_j(z) + \lambda_j\|z - w\|_{M_j} \right\}.$$

By Assumption 1, $M_j \preceq B\mathbf{I}_d$. Fix $w, w' \in \bar{\mathbf{C}}$ and $\eta > 0$, and choose $z_\eta \in \bar{\mathbf{C}}$ such that

$$\bar{g}_j(w') + \eta > \bar{f}_j(z_\eta) + \lambda_j\|z_\eta - w'\|_{M_j}.$$

Then

$$\bar{g}_j(w) - \bar{g}_j(w') \leq \lambda_j\left(\|z_\eta - w\|_{M_j} - \|z_\eta - w'\|_{M_j}\right) + \eta$$
$$\leq \lambda_j\|w - w'\|_{M_j} + \eta \leq \sqrt{B}\lambda_j\|w - w'\|_2 + \eta.$$

Letting $\eta \downarrow 0$ and exchanging $w, w'$ proves the claim on $\bar{\mathbf{C}}$. $\qquad\square$

**Claim 2.** *Suppose Assumption 1 holds with $B < \frac{1}{4\tau\varepsilon}$ and*

$$\lambda > 2\alpha_u(\tau B + 1)\delta + 2c_0\sqrt{\frac{d\zeta}{n}} + 2\tau c_0\sqrt{\frac{Bd\zeta}{N_{\mathbf{S}}}}.$$

*Conditioned on the event where Lemma 4, Proposition 7, and the taskwise GLM invariant-region noise bounds hold for all $j \in \mathbf{S}$, for any $\beta \in \mathbf{C}$ such that $\|\bar{\beta} - \bar{\hat{\beta}}_{G\mathbf{S}}\|_2 \leq \frac{\lambda}{2\alpha_u\sqrt{B}}$, we have:*

$$\nabla^2\bar{G}_{\mathbf{S}}(\bar{\beta}) \succeq \alpha_\ell|\mathbf{S}|\mathbf{I}_d.$$

*Proof.* Let $\eta := \frac{\lambda}{2\alpha_u\sqrt{B}}$. Consider any $\beta$ such that $\|\bar{\beta} - \bar{\hat{\beta}}_{G\mathbf{S}}\|_2 \leq \eta$. First, we bound the distance in the task-specific norm $\|\cdot\|_{\Sigma_j}$. Using the assumption $\Sigma_j \preceq B\Sigma_{\mathbf{S}}$:

$$\|\beta - \hat{\beta}_{G\mathbf{S}}\|_{\Sigma_j} \leq \sqrt{B}\|\beta - \hat{\beta}_{G\mathbf{S}}\|_{\Sigma_{\mathbf{S}}}$$
$$= \sqrt{B}\|\bar{\beta} - \bar{\hat{\beta}}_{G\mathbf{S}}\|_2$$
$$\leq \sqrt{B}\eta = \frac{\lambda}{2\alpha_u}.$$

Next, we apply the triangle inequality and the error bound for the pooling estimator (Lemma 4). Recall that under the favorable event, $\hat{\beta}_{G_{\mathbf{S}}} = \hat{\beta}_{F_{\mathbf{S}}}$. Thus:

$$\|\beta - \theta_j^\star\|_{\boldsymbol{\Sigma}_j} \leq \|\beta - \hat{\beta}_{G_{\mathbf{S}}}\|_{\boldsymbol{\Sigma}_j} + \|\hat{\beta}_{G_{\mathbf{S}}} - \theta_j^\star\|_{\boldsymbol{\Sigma}_j}$$

$$\leq \frac{\lambda}{2\alpha_u} + \left( (\tau B + 1)\delta + \frac{c_0}{\alpha_\ell}\sqrt{\frac{Bd\zeta}{N_{\mathbf{S}}}} \right).$$

To ensure $\beta$ lies in the invariant region $R_j$, Proposition 3 requires $\lambda > \alpha_u\|\beta - \theta_j^\star\|_{\boldsymbol{\Sigma}_j} + c_0\sqrt{\frac{d\zeta}{n}}$. Substituting the bound above, it suffices to show:

$$\lambda > \alpha_u \left( \frac{\lambda}{2\alpha_u} + (\tau B + 1)\delta + \frac{c_0}{\alpha_\ell}\sqrt{\frac{Bd\zeta}{N_{\mathbf{S}}}} \right) + c_0\sqrt{\frac{d\zeta}{n}}.$$

Simplifying the right-hand side:

$$\lambda > \frac{\lambda}{2} + \alpha_u(\tau B + 1)\delta + \tau c_0\sqrt{\frac{Bd\zeta}{N_{\mathbf{S}}}} + c_0\sqrt{\frac{d\zeta}{n}}.$$

Rearranging terms, this condition is equivalent to

$$\frac{\lambda}{2} > \alpha_u(\tau B + 1)\delta + \tau c_0\sqrt{\frac{Bd\zeta}{N_{\mathbf{S}}}} + c_0\sqrt{\frac{d\zeta}{n}},$$

which is exactly the assumed lower bound on $\lambda$.

Consequently, for all $\beta$ in the neighborhood, $\beta \in R_j$ holds for all $j \in \mathbf{S}$. This implies $G_{\mathbf{S}}(\beta) = F_{\mathbf{S}}(\beta)$ locally. Finally, we compute the Hessian lower bound. The Hessian of the transformed inlier loss is:

$$\nabla^2 \bar{G}_{\mathbf{S}}(\bar{\beta}) = \nabla^2 \bar{F}_{\mathbf{S}}(\bar{\beta})$$

$$= \boldsymbol{\Sigma}_{\mathbf{S}}^{-1/2} \left( \sum_{j \in \mathbf{S}} \nabla^2 f_j(\beta) \right) \boldsymbol{\Sigma}_{\mathbf{S}}^{-1/2}.$$

For GLM, $\nabla^2 f_j(\beta) = \frac{1}{n}\mathbf{X}_j^\top \mathbf{D}_j \mathbf{X}_j$, where $\mathbf{D}_j$ is a diagonal matrix with entries $\psi''(\cdot) \geq \alpha_\ell$. Thus, $\nabla^2 f_j(\beta) \succeq \alpha_\ell \boldsymbol{\Sigma}_j$. Summing over inliers:

$$\nabla^2 \bar{G}_{\mathbf{S}}(\bar{\beta}) \succeq \boldsymbol{\Sigma}_{\mathbf{S}}^{-1/2} \left( \sum_{j \in \mathbf{S}} \alpha_\ell \boldsymbol{\Sigma}_j \right) \boldsymbol{\Sigma}_{\mathbf{S}}^{-1/2}$$

$$= \alpha_\ell \boldsymbol{\Sigma}_{\mathbf{S}}^{-1/2}(|\mathbf{S}|\boldsymbol{\Sigma}_{\mathbf{S}})\boldsymbol{\Sigma}_{\mathbf{S}}^{-1/2}$$

$$= \alpha_\ell|\mathbf{S}|\mathbf{I}_d.$$

This completes the proof. $\qquad\square$

### D.4. Proof of Theorem 4 (In-sample MSE of GLM)

*Proof.* Let $\mathcal{E}_{\text{safe}}^{\text{glm}}$ be the event obtained by applying Proposition 4 taskwise with failure probability $\kappa/(2m)$. By a union bound, $\mathbb{P}(\mathcal{E}_{\text{safe}}^{\text{glm}}) \geq 1 - \kappa/2$, and on this event, for all $j \in [m]$,

$$\|\hat{\theta}_j - \theta_j^\star\|_{\boldsymbol{\Sigma}_j}^2 \lesssim \frac{d\zeta}{n} + \lambda^2 \lesssim q^2\frac{d\zeta}{n}.$$

This proves part (a), so the remainder proves the sharper inlier guarantee in part (b). Recall that $q \gtrsim 1$, so we may choose the implicit constant large enough that $q \geq 8c_0$. Also, since part (b) assumes $B \lesssim \min(1/\varepsilon, m)$, the favorable regime of Proposition 8 is available and $N_{\mathbf{S}} = |\mathbf{S}|n \asymp mn$.

**Case 1: Proposition 8 Applies.** Suppose

$$\lambda > 2\alpha_u(\tau B + 1)\delta + 2c_0\sqrt{\frac{d\zeta}{n}} + 2\tau c_0\sqrt{\frac{Bd\zeta}{N_\mathbf{S}}}.$$

Then Proposition 8 applies. Its proof constructs a transfer event $\mathcal{E}_{\text{transfer}}^{\text{glm}}$ with $\mathbb{P}(\mathcal{E}_{\text{transfer}}^{\text{glm}}) \geq 1 - \kappa/2$ under the present good-event budget. On $\mathcal{E}_{\text{safe}}^{\text{glm}} \cap \mathcal{E}_{\text{transfer}}^{\text{glm}}$, which has probability at least $1 - \kappa$, for all inlier tasks $j \in \mathbf{S}$,

$$\|\hat{\theta}_j - \theta_j^\star\|_{\mathbf{\Sigma}_j} \leq (\tau B + 1)\delta + \frac{c_0}{\alpha_\ell}\sqrt{\frac{Bd\zeta}{N_\mathbf{S}}} + \frac{2\varepsilon B\lambda}{\alpha_\ell}.$$

Squaring and using $(a + b + c)^2 \lesssim a^2 + b^2 + c^2$,

$$\|\hat{\theta}_j - \theta_j^\star\|_{\mathbf{\Sigma}_j}^2 \lesssim B^2\delta^2 + \frac{Bd\zeta}{N_\mathbf{S}} + q^2 B^2 \varepsilon^2 \frac{d\zeta}{n}.$$

Moreover, the displayed condition implies $\lambda \gtrsim B\delta$, hence

$$B^2\delta^2 \lesssim q^2 \frac{d\zeta}{n}.$$

Therefore,

$$\|\hat{\theta}_j - \theta_j^\star\|_{\mathbf{\Sigma}_j}^2 \lesssim \frac{Bd\zeta}{mn} + \min\left(B^2\delta^2, q^2\frac{d\zeta}{n}\right) + q^2 B^2 \varepsilon^2 \frac{d\zeta}{n}.$$

**Case 2: Proposition 8 Does Not Apply.** If the displayed condition fails, then

$$\lambda \leq 2\alpha_u(\tau B + 1)\delta + 2c_0\sqrt{\frac{d\zeta}{n}} + 2\tau c_0\sqrt{\frac{Bd\zeta}{N_\mathbf{S}}}.$$

Since $q \geq 8c_0$, we can absorb the middle term into the left-hand side and obtain

$$\frac{\lambda}{2} \leq 2\alpha_u(\tau B + 1)\delta + 2\tau c_0\sqrt{\frac{Bd\zeta}{N_\mathbf{S}}}.$$

Hence at least one of the following two alternatives must hold:

$$\lambda \lesssim B\delta \qquad \text{or} \qquad \lambda \lesssim \sqrt{\frac{Bd\zeta}{N_\mathbf{S}}}.$$

Equivalently,

$$q^2\frac{d\zeta}{n} \lesssim B^2\delta^2 \qquad \text{or} \qquad q^2\frac{d\zeta}{n} \lesssim \frac{Bd\zeta}{N_\mathbf{S}}.$$

In either subcase,

$$q^2\frac{d\zeta}{n} \lesssim \frac{Bd\zeta}{mn} + \min\left(B^2\delta^2, q^2\frac{d\zeta}{n}\right).$$

On the already-defined event $\mathcal{E}_{\text{safe}}^{\text{glm}}$, the GLM personalization bound gives

$$\|\hat{\theta}_j - \theta_j^\star\|_{\mathbf{\Sigma}_j}^2 \lesssim \frac{d\zeta}{n} + \lambda^2 \lesssim q^2\frac{d\zeta}{n}.$$

Combining the last two displays yields the same theorem-level bound in this regime.

This completes the proof. □

# E. Proof of Theorem 3 (Population MSE)

*Proof.* The theorem relates the in-sample MSE to the population MSE.

**Part 1: Via Comparability.** The first inequality $\mathcal{E}_j(\hat{\theta}_j) \leq \nu_j \mathcal{E}_j^{\text{in}}(\hat{\theta}_j)$ follows directly from the definition of the comparability constant $\nu_j$ (Definition 9) and the definition of the MSEs. Specifically, $\|\cdot\|_{\hat{\boldsymbol{\Sigma}}_j}^2 \leq \nu_j \|\cdot\|_{\boldsymbol{\Sigma}_j}^2$.

**Part 2: Via Intrinsic Dimension (Safety).** For the second inequality, we consider the case where $\nu_j$ might be large. Assume that the parameter domain is known to be bounded, $\mathbf{C} = \mathbf{B}(0, \xi)$, and that $\theta_j^\star \in \mathbf{C}$. Let

$$\hat{\theta}_j^\xi \in \operatorname*{argmin}_{\theta \in \mathbf{B}(0,\xi)} \|\theta - \hat{\theta}_j\|_{\hat{\boldsymbol{\Sigma}}_j}^2$$

be the $\|\cdot\|_{\boldsymbol{\Sigma}_j}$-norm projection of $\hat{\theta}_j$ onto $\mathbf{B}(0, \xi)$.

Since $\theta_j^\star \in \mathbf{B}(0, \xi)$, the projection property gives

$$\|\hat{\theta}_j^\xi - \theta_j^\star\|_{\hat{\boldsymbol{\Sigma}}_j}^2 \leq \|\hat{\theta}_j - \theta_j^\star\|_{\hat{\boldsymbol{\Sigma}}_j}^2 = \mathcal{E}_j^{\text{in}}(\hat{\theta}_j).$$

Moreover, both $\hat{\theta}_j^\xi$ and $\theta_j^\star$ belong to $\mathbf{B}(0, \xi)$, so

$$\|\hat{\theta}_j^\xi - \theta_j^\star\|_2 \leq 2\xi.$$

Let $U_j$ be the high-probability upper bound on $\|x_{ji}\|_2$. Applying Lemma 8 to the bounded estimator $\hat{\theta}_j^\xi$, we obtain

$$\mathcal{E}_j(\hat{\theta}_j^\xi) \lesssim \|\hat{\theta}_j^\xi - \theta_j^\star\|_{\boldsymbol{\Sigma}_j}^2 + \frac{U_j^2}{n}\|\hat{\theta}_j^\xi - \theta_j^\star\|_2^2.$$

Combining the previous two displays yields

$$\mathcal{E}_j(\hat{\theta}_j^\xi) \lesssim \mathcal{E}_j^{\text{in}}(\hat{\theta}_j) + \frac{\xi^2 U_j^2}{n}.$$

This confirms that even if the spectral ratio $\nu_j$ diverges, the projected estimator has population risk controlled by the in-sample risk plus a $1/n$-order safety term. $\qquad\square$

## F. Results for Linear Model with General Sample Size

This section generalizes our main results to the setting where sample sizes $n_j$ vary across tasks. We demonstrate that our matrix-weighted approach naturally adapts to heterogeneous information levels by scaling the regularization parameter inversely with the square root of the sample size.

Because the heterogeneous-sample-size objective aggregates the Gram matrices $\mathbf{V}_j$, the relevant inlier geometry is their sample-size-weighted average. Accordingly, throughout this section we slightly abuse notation and write

$$\boldsymbol{\Sigma}_\mathbf{S} := \frac{1}{N_\mathbf{S}} \sum_{j \in \mathbf{S}} \mathbf{V}_j = \sum_{j \in \mathbf{S}} \frac{n_j}{N_\mathbf{S}} \boldsymbol{\Sigma}_j.$$

The balancedness assumption in this section is interpreted with respect to this weighted inlier average, namely

$$\boldsymbol{\Sigma}_j \preceq B\boldsymbol{\Sigma}_\mathbf{S}, \qquad j \in [m].$$

When $n_j = n$ for all $j$, this coincides with the original definition $\boldsymbol{\Sigma}_\mathbf{S} = |\mathbf{S}|^{-1} \sum_{j \in \mathbf{S}} \boldsymbol{\Sigma}_j$.

**Task Relatedness.** We adopt the generalized task relatedness definition from Duan and Wang (2023), which scales the closeness condition by the sample sizes.

**Definition 2** (Task Relatedness for General Sample Size). There exist $\varepsilon, \delta_0 \geq 0$ and a subset of inliers $\mathbf{S} \subseteq [m]$ such that:

$$\min_{\theta \in \mathbb{R}^d} \max_{j \in \mathbf{S}} \left\{ \sqrt{n_j} \|\theta_j^\star - \theta\|_2 \right\} \leq \delta_0 \quad \text{and} \quad \sum_{j \in \mathbf{S}^c} \sqrt{n_j} \leq \varepsilon \frac{N_\mathbf{S}}{\sqrt{n^\star}},$$

where $N_\mathbf{S} := \sum_{j \in \mathbf{S}} n_j$ is the total sample size of inliers, and $n^\star := \max_{j \in \mathbf{S}} n_j$.

To simplify the bounds, we define the structural constant $\alpha$ as:

$$\alpha := \frac{\sqrt{n^\star} \sum_{j \in \mathbf{S}} \sqrt{n_j}}{N_\mathbf{S}}.$$

Note that in the equal sample size case ($n_j = n$), we have $n^\star = n$, $N_\mathbf{S} = |\mathbf{S}|n$, and $\alpha = 1$, recovering the standard definition.

### F.1. Algorithm Adaptation

Recall the task-specific loss $f_j(\theta) := \frac{1}{2n_j}\|\mathbf{Y}_j - \mathbf{X}_j\theta\|_2^2$. For heterogeneous sample sizes, we weigh the loss by the sample size $w_j = n_j$ and scale the regularization parameter as $\lambda_j = \frac{\lambda}{\sqrt{n_j}}$, where $\lambda$ is a global tuning parameter. The resulting objective function is:

$$\mathcal{L}(\Theta) = \sum_{j=1}^m \left( \frac{1}{2}\|\mathbf{Y}_j - \mathbf{X}_j\theta_j\|_2^2 + \lambda\sqrt{n_j}\|\theta_j - \beta\|_{\mathbf{\Sigma}_j} \right). \tag{13}$$

We set $\mathbf{C} = \mathbb{R}^d$.

### F.2. MSE Bound

We present the main guarantee. Let $\zeta = \log(16m/\kappa)$. We select $\lambda = q\sqrt{d\zeta}$ for a sufficiently large universal constant $q \gtrsim 1$.

**Theorem 5** (MSE Bound: General Sample Size). *If we run Algorithm 1 with $q \gtrsim 1$, then with probability at least $1 - \kappa$:*

*(a) **Safety:** For all tasks $j \in [m]$, regardless of relatedness:*

$$\mathcal{E}_j^{\text{in}}(\hat{\theta}_j) \lesssim q^2 \frac{d\zeta}{n_j}.$$

*(b) **Adaptivity:** If the $(\varepsilon, \delta_0)$-task relatedness holds and the weighted balancedness assumption above is satisfied with $B \lesssim \min(1/\varepsilon, m)$, then for all inlier tasks $j \in \mathbf{S}$:*

$$\mathcal{E}_j^{\text{in}}(\hat{\theta}_j) \lesssim \frac{n^\star}{n_j} \frac{Bd\zeta}{N_\mathbf{S}} + \frac{1}{n_j} \min\left( \alpha^2 B^2 \delta_0^2, \, q^2 d\zeta \right) + \frac{\varepsilon^2 B^2 \lambda^2}{n^\star}.$$

**Discussion.** Theorem 5 confirms that our method's properties—Safety, Adaptivity, and Robustness—are preserved under sample size heterogeneity. The safety statement in part (a) does not require task relatedness or weighted balancedness; when $B$ is large or infinite, only the sharper transfer claim in part (b) becomes inactive. The bound interpolates between the independent rate ($d/n_j$) and a pooled variance term governed by ($n^\star/n_j$) $\cdot d/N_\mathbf{S}$. This extra factor reflects the use of a single global tuning constant $q$ across tasks with different information levels, and it disappears in the equal-sample-size regime. Importantly, strictly weaker spectral assumptions are required compared to Duan and Wang (2023), as we do not rely on minimum eigenvalue bounds.

**Population MSE.** In the task-wise i.i.d. random-design setting, the in-sample-to-population conversion in Theorem 3 is taskwise and applies verbatim to Theorem 5, after the same standard rescaling of $\kappa$ in $\zeta$. Thus, if $\bar{\mathbf{\Sigma}}_j \preceq \nu_j\mathbf{\Sigma}_j$, then $\mathcal{E}_j(\hat{\theta}_j) \leq \nu_j\mathcal{E}_j^{\text{in}}(\hat{\theta}_j)$; with a bounded parameter domain, the intrinsic-dimension safety alternative gives $\mathcal{E}_j(\hat{\theta}_j^\xi) \lesssim \mathcal{E}_j^{\text{in}}(\hat{\theta}_j) + \xi^2 U_j^2/n_j$.

## G. Proofs for the Linear Model with General Sample Size

We organize the heterogeneous-sample-size analysis in the same proof order as the equal-sample-size linear model: setup and notation, the inlier pooling estimator, the outlier perturbation step, and the final theorem proof.

### G.1. Setup and Notation

Recall the task-specific loss

$$f_j(\theta) = \frac{1}{2n_j}\|\mathbf{Y}_j - \mathbf{X}_j\theta\|_2^2,$$

the task-wise regularization level $\lambda_j = \lambda/\sqrt{n_j}$, and the weighted objective

$$\mathcal{L}(\Theta) = \sum_{j=1}^{m}\left(\frac{1}{2}\|\mathbf{Y}_j - \mathbf{X}_j\theta_j\|_2^2 + \lambda\sqrt{n_j}\|\theta_j - \beta\|_{\Sigma_j}\right).$$

We work on the unconstrained domain $\mathbf{C} = \mathbb{R}^d$ and define

$$g_j(\beta) := (f_j \star \lambda_j\|\cdot\|_{\Sigma_j})(\beta),$$
$$F_{\mathbf{S}}(\beta) := \sum_{j\in\mathbf{S}} n_j f_j(\beta),$$
$$G_{\mathbf{S}}(\beta) := \sum_{j\in\mathbf{S}} n_j g_j(\beta),$$
$$L(\beta) := \sum_{j\in\mathbf{S}^c} n_j g_j(\beta),$$
$$G(\beta) := G_{\mathbf{S}}(\beta) + L(\beta).$$

Let

$$\Sigma_{\mathbf{S}} := \frac{1}{N_{\mathbf{S}}}\sum_{j\in\mathbf{S}}\mathbf{V}_j,$$

Throughout this appendix, the balancedness assumption is interpreted with this weighted inlier covariance, namely $\Sigma_j \preceq B\Sigma_{\mathbf{S}}$ for all $j \in [m]$. As before, the invariant region is

$$R_j := \{\beta \in \mathbf{C} \mid f_j(\beta) = g_j(\beta)\}, \qquad j \in [m].$$

As in the equal-sample-size appendix, we work after restricting to $\text{Range}(\Sigma_{\mathbf{S}})$, so without loss of generality $\Sigma_{\mathbf{S}} \succ 0$.

**Pooled Whitening Notation.** We use the pooled whitening map

$$\bar{v} := \Sigma_{\mathbf{S}}^{1/2}v, \qquad v \in \mathbb{R}^d.$$

For any function $h : \mathbb{R}^d \to \mathbb{R}$, we define its pooled-whitened version by

$$\bar{h}(\bar{v}) := h(\Sigma_{\mathbf{S}}^{-1/2}\bar{v}).$$

Thus $\bar{\theta}, \bar{\beta}, \bar{\theta}^\star, \bar{\theta}_j^\star, \bar{\hat{\beta}}_{F_{\mathbf{S}}}, \bar{\hat{\beta}}_{G_{\mathbf{S}}}$, and $\bar{\hat{\beta}}_G$ are all instances of the same pooled whitening map. In particular, we write $\bar{F}_{\mathbf{S}}, \bar{G}_{\mathbf{S}}, \bar{L}$, and $\bar{G}$ for the transformed objectives.

### G.2. The Inlier Pooling Estimator

We first characterize the inlier-only estimator and identify a regime where the regularized inlier objective collapses to the same minimizer.

**Lemma 6** (Error Bound of Pooling Estimator)**.** *With probability at least $1 - \frac{\kappa}{4}$, the pooling estimator $\hat{\beta}_{F_{\mathbf{S}}}$ satisfies simultaneously for all $j \in \mathbf{S}$:*

$$\|\hat{\beta}_{F_{\mathbf{S}}} - \theta^\star\|_{\Sigma_j} \le \alpha\frac{B\delta_0}{\sqrt{n^\star}} + c_0\sqrt{\frac{Bd\zeta}{N_{\mathbf{S}}}}.$$

*Consequently, for each $j \in \mathbf{S}$,*

$$\|\hat{\beta}_{F_{\mathbf{S}}} - \theta_j^\star\|_{\Sigma_j} \lesssim \alpha\frac{B\delta_0}{\sqrt{n_j}} + \sqrt{\frac{Bd\zeta}{N_{\mathbf{S}}}}.$$

*Proof.* Recall $\hat{\beta}_{F_{\mathbf{S}}} = \mathbf{V}_{\mathbf{S}}^{-1}\sum_{k\in\mathbf{S}}\mathbf{X}_k^\top\mathbf{Y}_k$ and write

$$\|\hat{\beta}_{F_{\mathbf{S}}} - \theta^\star\|_{\Sigma_j} \le \text{Bias}_j + \text{Variance}_j.$$

**Bias Term.** Let $\eta_k^\star := \theta_k^\star - \theta^\star$ and $a := \sum_{k \in \mathbf{S}} \mathbf{V}_k \eta_k^\star$. Then $\mathrm{Bias}_j = \|\mathbf{V}_\mathbf{S}^{-1} a\|_{\boldsymbol{\Sigma}_j}$. Using $\boldsymbol{\Sigma}_j \preceq B\boldsymbol{\Sigma}_\mathbf{S} = \frac{B}{N_\mathbf{S}} \mathbf{V}_\mathbf{S}$, we have

$$\mathrm{Bias}_j^2 \leq \frac{B}{N_\mathbf{S}} \|\mathbf{V}_\mathbf{S}^{-1/2} a\|_2^2.$$

Next, by $\boldsymbol{\Sigma}_k \preceq B\boldsymbol{\Sigma}_\mathbf{S}$ we have $\mathbf{V}_k \preceq \frac{n_k B}{N_\mathbf{S}} \mathbf{V}_\mathbf{S}$, hence

$$\left\|\mathbf{V}_\mathbf{S}^{-1/2} \mathbf{V}_k \eta_k^\star\right\|_2 \leq \sqrt{\frac{n_k B}{N_\mathbf{S}}} \, \|\eta_k^\star\|_{\mathbf{V}_k}.$$

Therefore,

$$\left\|\mathbf{V}_\mathbf{S}^{-1/2} a\right\|_2 \leq \sum_{k \in \mathbf{S}} \left\|\mathbf{V}_\mathbf{S}^{-1/2} \mathbf{V}_k \eta_k^\star\right\|_2 \leq \sqrt{\frac{B}{N_\mathbf{S}}} \sum_{k \in \mathbf{S}} n_k \|\eta_k^\star\|_{\boldsymbol{\Sigma}_k}.$$

Under the bounded covariate assumption, $\|\eta_k^\star\|_{\boldsymbol{\Sigma}_k} \leq \|\eta_k^\star\|_2$, and the task-relatedness condition gives $\sqrt{n_k}\|\eta_k^\star\|_2 \leq \delta_0$. Hence

$$\mathrm{Bias}_j \leq \frac{B}{N_\mathbf{S}} \sum_{k \in \mathbf{S}} \sqrt{n_k}\, \delta_0 = \alpha \frac{B\delta_0}{\sqrt{n^\star}}.$$

**Variance Term.** Using Hanson–Wright with failure level $\kappa/(4m)$ for each $j \in \mathbf{S}$, a union bound, and $\boldsymbol{\Sigma}_j \preceq \frac{B}{N_\mathbf{S}} \mathbf{V}_\mathbf{S}$,

$$\mathrm{Variance}_j \leq c_0 \sqrt{\mathrm{tr}(\boldsymbol{\Sigma}_j \mathbf{V}_\mathbf{S}^{-1}) \zeta} \leq c_0 \sqrt{\frac{Bd\zeta}{N_\mathbf{S}}}.$$

For the task-specific error, apply triangle inequality:

$$\|\hat{\beta}_{F_\mathbf{S}} - \theta_j^\star\|_{\boldsymbol{\Sigma}_j} \leq \|\hat{\beta}_{F_\mathbf{S}} - \theta^\star\|_{\boldsymbol{\Sigma}_j} + \|\theta^\star - \theta_j^\star\|_{\boldsymbol{\Sigma}_j}.$$

The first term is bounded above. For the second, $\|\theta^\star - \theta_j^\star\|_{\boldsymbol{\Sigma}_j} \leq \|\theta^\star - \theta_j^\star\|_2 \leq \delta_0/\sqrt{n_j}$ by definition of task relatedness. Since $\alpha B \geq 1$, this yields

$$\|\hat{\beta}_{F_\mathbf{S}} - \theta_j^\star\|_{\boldsymbol{\Sigma}_j} \lesssim \alpha \frac{B\delta_0}{\sqrt{n_j}} + \sqrt{\frac{Bd\zeta}{N_\mathbf{S}}}.$$

$\square$

**Proposition 9** (Equivalence of Pooling Estimators: General Sample Size)**.** *There is a universal constant $C_{\mathrm{gs}}$ such that the following holds. Suppose for all $j \in \mathbf{S}$,*

$$\frac{\lambda}{\sqrt{n_j}} > C_{\mathrm{gs}} \left(\alpha B \frac{\delta_0}{\sqrt{n_j}} + c_0 \sqrt{\frac{Bd\zeta}{N_\mathbf{S}}} + c_0 \sqrt{\frac{d\zeta}{n_j}}\right).$$

*Then, with probability at least $1 - \frac{\kappa}{2}$, $\hat{\beta}_{F_\mathbf{S}} \in R_j$ for all $j \in \mathbf{S}$, and $\hat{\beta}_{F_\mathbf{S}}$ minimizes $G_\mathbf{S}$. In particular, we may choose $\hat{\beta}_{G_\mathbf{S}} = \hat{\beta}_{F_\mathbf{S}}$.*

*Proof.* Let $\mathcal{E}_{\mathrm{pool}}^{\mathrm{gs}}$ be the event from Lemma 6, which has probability at least $1 - \kappa/4$. Let $\mathcal{E}_{\mathrm{inv}}^{\mathrm{gs}}$ be the event that the taskwise invariant-region noise bound from Proposition 1 holds for all $j \in \mathbf{S}$, each invoked with failure level $\kappa/(4m)$. Then $\mathbb{P}(\mathcal{E}_{\mathrm{pool}}^{\mathrm{gs}} \cap \mathcal{E}_{\mathrm{inv}}^{\mathrm{gs}}) \geq 1 - \kappa/2$. On this intersection, it suffices to verify for each $j \in \mathbf{S}$:

$$\frac{\lambda}{\sqrt{n_j}} > \|\theta_j^\star - \hat{\beta}_{F_\mathbf{S}}\|_{\boldsymbol{\Sigma}_j} + c_0 \sqrt{\frac{d\zeta}{n_j}}.$$

We bound the first term by triangle inequality:

$$\|\theta_j^\star - \hat{\beta}_{F_\mathbf{S}}\|_{\boldsymbol{\Sigma}_j} \leq \|\theta_j^\star - \theta^\star\|_{\boldsymbol{\Sigma}_j} + \|\theta^\star - \hat{\beta}_{F_\mathbf{S}}\|_{\boldsymbol{\Sigma}_j}.$$

Since $\|\mathbf{\Sigma}_j\|_{\mathrm{op}} \leq 1$ and $\sqrt{n_j}\|\theta_j^\star - \theta^\star\|_2 \leq \delta_0$, we have

$$\|\theta_j^\star - \theta^\star\|_{\mathbf{\Sigma}_j} \leq \frac{\delta_0}{\sqrt{n_j}}.$$

By Lemma 6,

$$\|\theta^\star - \hat{\beta}_{F_{\mathbf{S}}}\|_{\mathbf{\Sigma}_j} \leq \alpha \frac{B\delta_0}{\sqrt{n^\star}} + c_0\sqrt{\frac{Bd\zeta}{N_{\mathbf{S}}}} \leq \alpha \frac{B\delta_0}{\sqrt{n_j}} + c_0\sqrt{\frac{Bd\zeta}{N_{\mathbf{S}}}}.$$

Combining the last three displays and using $\alpha B \geq 1$ (we may assume $B \geq 1$), the assumed lower bound on $\lambda/\sqrt{n_j}$ implies $\hat{\beta}_{F_{\mathbf{S}}} \in R_j$ for all $j \in \mathbf{S}$. For each $j \in \mathbf{S}$, define the positive slack

$$\Delta_j^{\mathrm{gs}} := \frac{\lambda}{\sqrt{n_j}} - \left(\|\theta_j^\star - \hat{\beta}_{F_{\mathbf{S}}}\|_{\mathbf{\Sigma}_j} + c_0\sqrt{\frac{d\zeta}{n_j}}\right) > 0.$$

Since $\beta \mapsto \|\theta_j^\star - \beta\|_{\mathbf{\Sigma}_j}$ is continuous and the stochastic term in Proposition 1 does not depend on $\beta$, there exists $r_j > 0$ such that

$$\|\beta - \hat{\beta}_{F_{\mathbf{S}}}\|_2 < r_j \quad \Longrightarrow \quad \frac{\lambda}{\sqrt{n_j}} > \|\theta_j^\star - \beta\|_{\mathbf{\Sigma}_j} + c_0\sqrt{\frac{d\zeta}{n_j}}.$$

Hence Proposition 1 implies $f_j(\beta) = g_j(\beta)$ throughout the Euclidean ball $B_{r_j}(\hat{\beta}_{F_{\mathbf{S}}})$. Let $r := \min_{j \in \mathbf{S}} r_j > 0$. Then

$$F_{\mathbf{S}}(\beta) = G_{\mathbf{S}}(\beta) \qquad \text{for all } \beta \in B_r(\hat{\beta}_{F_{\mathbf{S}}}).$$

Since $F_{\mathbf{S}}$ is strictly convex with Hessian $N_{\mathbf{S}}\mathbf{\Sigma}_{\mathbf{S}} \succ 0$, $\hat{\beta}_{F_{\mathbf{S}}}$ is its unique minimizer, and therefore a local minimizer of $G_{\mathbf{S}}$ as well. By convexity, this local minimizer is global, so $\hat{\beta}_{F_{\mathbf{S}}}$ minimizes $G_{\mathbf{S}}$. This proves the claimed statement on $\mathcal{E}_{\mathrm{pool}}^{\mathrm{gs}} \cap \mathcal{E}_{\mathrm{inv}}^{\mathrm{gs}}$, whose probability is at least $1 - \kappa/2$. $\qquad \square$

### G.3. A Perturbation Bound for Outliers

We next control the passage from the inlier objective $G_{\mathbf{S}}$ to the full objective $G = G_{\mathbf{S}} + L$, where $L$ denotes the aggregate outlier contribution introduced in the setup.

**Claim 3.** *Assume $\varepsilon < \frac{1}{3B}$. With the same constant $C_{\mathrm{gs}}$ as in Proposition 9, suppose*

$$\frac{\lambda}{\sqrt{n_j}} > C_{\mathrm{gs}}\left(\alpha B \frac{\delta_0}{\sqrt{n_j}} + c_0\sqrt{\frac{Bd\zeta}{N_{\mathbf{S}}}} + c_0\sqrt{\frac{d\zeta}{n_j}}\right)$$

*for all $j \in \mathbf{S}$. On the event where Proposition 9, Lemma 6, and the taskwise invariant-region noise bounds hold, for all $\beta$ such that $\|\bar{\beta} - \hat{\bar{\beta}}_{G_{\mathbf{S}}}\|_2 \leq \frac{\lambda}{2\sqrt{Bn^\star}}$, we have*

$$\nabla^2 \bar{G}_{\mathbf{S}}(\bar{\beta}) \succeq N_{\mathbf{S}}\mathbf{I}_d.$$

*Proof.* The proof follows the same logic as the equal-sample-size case, but we verify the weighted scaling explicitly. Fix $\beta$ such that $\|\bar{\beta} - \hat{\bar{\beta}}_{G_{\mathbf{S}}}\|_2 \leq \frac{\lambda}{2\sqrt{Bn^\star}}$. For any $j \in \mathbf{S}$, using $\mathbf{\Sigma}_j \preceq B\mathbf{\Sigma}_{\mathbf{S}}$:

$$\|\beta - \hat{\beta}_{G_{\mathbf{S}}}\|_{\mathbf{\Sigma}_j} \leq \sqrt{B}\|\beta - \hat{\beta}_{G_{\mathbf{S}}}\|_{\mathbf{\Sigma}_{\mathbf{S}}} = \sqrt{B}\|\bar{\beta} - \hat{\bar{\beta}}_{G_{\mathbf{S}}}\|_2$$

$$\leq \frac{\lambda}{2\sqrt{n^\star}} \leq \frac{\lambda}{2\sqrt{n_j}}.$$

By Proposition 9, we have $\hat{\beta}_{G_{\mathbf{S}}} = \hat{\beta}_{F_{\mathbf{S}}}$. Hence, by Lemma 6,

$$\|\hat{\beta}_{G_{\mathbf{S}}} - \theta_j^\star\|_{\mathbf{\Sigma}_j} \lesssim \alpha \frac{B\delta_0}{\sqrt{n_j}} + \sqrt{\frac{Bd\zeta}{N_{\mathbf{S}}}}.$$

Hence

$$\|\beta - \theta_j^\star\|_{\Sigma_j} \lesssim \frac{\lambda}{2\sqrt{n_j}} + \alpha \frac{B\delta_0}{\sqrt{n_j}} + \sqrt{\frac{Bd\zeta}{N_{\mathbf{S}}}}.$$

Proposition 1 (applied with task-wise $\lambda_j = \lambda/\sqrt{n_j}$) requires

$$\frac{\lambda}{\sqrt{n_j}} > \|\beta - \theta_j^\star\|_{\Sigma_j} + c_0 \sqrt{\frac{d\zeta}{n_j}}.$$

The displayed assumption, with $C_{\text{gs}}$ chosen sufficiently large, leaves enough slack after the $\lambda/(2\sqrt{n_j})$ term to verify this condition. Therefore $\beta \in R_j$ for all $j \in \mathbf{S}$, so $G_{\mathbf{S}}(\beta) = F_{\mathbf{S}}(\beta)$ locally. Consequently, $\nabla^2 \bar{G}_{\mathbf{S}}(\bar{\beta}) = \nabla^2 \bar{F}_{\mathbf{S}}(\bar{\beta}) = N_{\mathbf{S}} \mathbf{I}_d$. $\quad\square$

**Proposition 10** (Key Proposition: General Sample Size). *Assume $\varepsilon < \frac{1}{3B}$. With the same universal constant $C_{\text{gs}}$, suppose the regularization parameter satisfies*

$$\frac{\lambda}{\sqrt{n_j}} > C_{\text{gs}} \left( \alpha B \frac{\delta_0}{\sqrt{n_j}} + c_0 \sqrt{\frac{Bd\zeta}{N_{\mathbf{S}}}} + c_0 \sqrt{\frac{d\zeta}{n_j}} \right)$$

*for all $j \in \mathbf{S}$, then with probability at least $1 - \kappa$:*

1. *The pooled center satisfies*

$$\|\hat{\beta}_G - \theta_j^\star\|_{\Sigma_j} \lesssim \alpha \frac{B\delta_0}{\sqrt{n_j}} + \sqrt{\frac{Bd\zeta}{N_{\mathbf{S}}}} + \frac{\varepsilon B\lambda}{\sqrt{n^\star}}.$$

2. *Furthermore,*

$$\|\hat{\theta}_j - \hat{\beta}_G\|_{\Sigma_j} = 0, \qquad \|\hat{\theta}_j - \theta_j^\star\|_{\Sigma_j} = \|\hat{\beta}_G - \theta_j^\star\|_{\Sigma_j}$$

*for all $j \in \mathbf{S}$.*

*Proof.* The proof strategy mirrors Proposition 6, adapted for weighted sums. Let $\mathcal{E}_{\text{gs}}$ be the good event from Proposition 9: the pooling estimator bound and all taskwise invariant-region noise bounds hold. The pooled event is allocated failure $\kappa/4$, and the taskwise events are allocated failure $\kappa/(4m)$ each, so $\mathbb{P}(\mathcal{E}_{\text{gs}}) \geq 1 - \kappa/2 \geq 1 - \kappa$. Using the pooled whitening notation introduced above, let

$$\bar{\hat{\beta}}_G := \Sigma_{\mathbf{S}}^{1/2} \hat{\beta}_G, \qquad \bar{\hat{\beta}}_{G_{\mathbf{S}}} := \Sigma_{\mathbf{S}}^{1/2} \hat{\beta}_{G_{\mathbf{S}}}.$$

**Step 1: Strong Convexity.** The aggregate inlier loss is $F_{\mathbf{S}}(\beta) = \sum_{j \in \mathbf{S}} \frac{1}{2}\|\mathbf{Y}_j - \mathbf{X}_j \beta\|_2^2$. By Proposition 9, we may set $\hat{\beta}_{G_{\mathbf{S}}} = \hat{\beta}_{F_{\mathbf{S}}}$ on the high-probability event. The Hessian of the transformed inlier loss is

$$\nabla^2 \bar{F}_{\mathbf{S}}(\bar{\beta}) = \Sigma_{\mathbf{S}}^{-1/2} \left( \sum_{j \in \mathbf{S}} \mathbf{X}_j^\top \mathbf{X}_j \right) \Sigma_{\mathbf{S}}^{-1/2} = \Sigma_{\mathbf{S}}^{-1/2} (N_{\mathbf{S}} \Sigma_{\mathbf{S}}) \Sigma_{\mathbf{S}}^{-1/2} = N_{\mathbf{S}} \mathbf{I}_d.$$

Thus, $\bar{F}_{\mathbf{S}}$ is $N_{\mathbf{S}}$-strongly convex. By Claim 3, this strong convexity holds for $\bar{G}_{\mathbf{S}}$ in the relevant neighborhood.

**Step 2: Lipschitz Constant of Outliers.** We bound the Lipschitz constant of the transformed outlier term $\bar{L}(\cdot) = \sum_{j \in \mathbf{S}^c} n_j \bar{g}_j(\cdot)$. From Lemma 3, $\bar{g}_j$ is $\sqrt{B}\lambda_j$-Lipschitz. Thus, $n_j \bar{g}_j$ is $n_j \sqrt{B}(\lambda/\sqrt{n_j}) = \sqrt{n_j}\sqrt{B}\lambda$-Lipschitz. Summing over outliers:

$$\text{Lip}(\bar{L}) \leq \sum_{j \in \mathbf{S}^c} \sqrt{B}\lambda\sqrt{n_j} = \sqrt{B}\lambda \sum_{j \in \mathbf{S}^c} \sqrt{n_j}.$$

Using the task-relatedness definition ($\sum_{\mathbf{S}^c} \sqrt{n_j} \leq \varepsilon \frac{N_{\mathbf{S}}}{\sqrt{n^\star}}$):

$$\text{Lip}(\bar{L}) \leq \frac{\varepsilon \sqrt{B}\lambda N_{\mathbf{S}}}{\sqrt{n^\star}}.$$

**Step 3: Perturbation Bound.**  We apply Lemma 11 with $\rho = N_{\mathbf{S}}$ and radius

$$r = \frac{\lambda}{2\sqrt{B n^\star}},$$

for which Claim 3 guarantees local strong convexity of $\bar{G}_{\mathbf{S}}$. The perturbation condition requires $\mathrm{Lip}(\bar{L}) < \rho r$. Using the bound above,

$$\mathrm{Lip}(\bar{L}) \leq \frac{\varepsilon \sqrt{B} \lambda N_{\mathbf{S}}}{\sqrt{n^\star}} = 2\varepsilon B \rho r.$$

Since $\varepsilon < \frac{1}{3B}$, the condition holds. The lemma yields

$$\|\hat{\bar{\beta}}_G - \hat{\bar{\beta}}_{G_{\mathbf{S}}}\|_2 \leq \frac{\mathrm{Lip}(\bar{L})}{\rho} \leq \frac{\varepsilon\sqrt{B}\lambda}{\sqrt{n^\star}}.$$

Converting to the $\mathbf{\Sigma}_j$-norm,

$$\|\hat{\beta}_G - \hat{\beta}_{G_{\mathbf{S}}}\|_{\mathbf{\Sigma}_j} \leq \sqrt{B}\|\hat{\beta}_G - \hat{\beta}_{G_{\mathbf{S}}}\|_{\mathbf{\Sigma}_{\mathbf{S}}} \leq \frac{\varepsilon B \lambda}{\sqrt{n^\star}}.$$

**Step 4: Combining Errors.**  Combining this with the pooling estimator error (Lemma 6):

$$
\begin{aligned}
\|\hat{\beta}_G - \theta_j^\star\|_{\mathbf{\Sigma}_j} &\leq \|\hat{\beta}_G - \hat{\beta}_{G_{\mathbf{S}}}\|_{\mathbf{\Sigma}_j} + \|\hat{\beta}_{G_{\mathbf{S}}} - \theta_j^\star\|_{\mathbf{\Sigma}_j} \\
&= \|\hat{\beta}_G - \hat{\beta}_{G_{\mathbf{S}}}\|_{\mathbf{\Sigma}_j} + \|\hat{\beta}_{F_{\mathbf{S}}} - \theta_j^\star\|_{\mathbf{\Sigma}_j} \\
&\lesssim \frac{\varepsilon B \lambda}{\sqrt{n^\star}} + \alpha \frac{B \delta_0}{\sqrt{n_j}} + \sqrt{\frac{B d\zeta}{N_{\mathbf{S}}}}.
\end{aligned}
$$

**Step 5: Invariant Region for the Full Estimator.**  It remains to justify the zero-seminorm personalization statement for inlier tasks. Proposition 1, with taskwise regularization $\lambda_j = \lambda/\sqrt{n_j}$, requires

$$\frac{\lambda}{\sqrt{n_j}} > \|\hat{\beta}_G - \theta_j^\star\|_{\mathbf{\Sigma}_j} + c_0 \sqrt{\frac{d\zeta}{n_j}}.$$

The deterministic part of the displayed assumption, with $C_{\mathrm{gs}}$ sufficiently large, controls the terms $\alpha B \delta_0/\sqrt{n_j}$, $\sqrt{B d\zeta/N_{\mathbf{S}}}$, and $c_0\sqrt{d\zeta/n_j}$. The perturbation term is absorbed by the small-contamination condition: since $n_j \leq n^\star$,

$$\frac{\varepsilon B \lambda}{\sqrt{n^\star}} \leq \varepsilon B \frac{\lambda}{\sqrt{n_j}} \leq \frac{1}{3} \frac{\lambda}{\sqrt{n_j}}.$$

Thus $\hat{\beta}_G \in R_j$ for every $j \in \mathbf{S}$, which implies that every task-specific minimizer at center $\hat{\beta}_G$ satisfies $\|\hat{\theta}_j - \hat{\beta}_G\|_{\mathbf{\Sigma}_j} = 0$. Hence the corresponding prediction-seminorm errors coincide, so the pooled-center error bound above also gives the stated bound for $\hat{\theta}_j$. $\qquad\square$

### G.4. Proof of Theorem 5

*Proof.*  Recall $\lambda = q\sqrt{d\zeta}$. Since $q \gtrsim 1$, we may choose the implicit constant so that $q \geq 8c_0$. Let $\mathcal{E}_{\mathrm{safe}}^{\mathrm{gs}}$ be the event obtained by applying the personalization bound taskwise with failure probability $\kappa/(2m)$. Then $\mathbb{P}(\mathcal{E}_{\mathrm{safe}}^{\mathrm{gs}}) \geq 1 - \kappa/2$, and on this event, for every task $j \in [m]$,

$$\|\hat{\theta}_j - \theta_j^\star\|_{\mathbf{\Sigma}_j}^2 \lesssim \frac{d\zeta}{n_j} + \frac{\lambda^2}{n_j} \lesssim q^2 \frac{d\zeta}{n_j}.$$

This proves part (a).

**Case 1: Proposition 10 Applies.** Because the inequality in Proposition 10 is hardest when $n_j = n^\star$, it suffices to assume

$$\frac{\lambda}{\sqrt{n^\star}} > C_{\mathrm{gs}} \left( \alpha B \frac{\delta_0}{\sqrt{n^\star}} + c_0 \sqrt{\frac{Bd\zeta}{N_{\mathbf{S}}}} + c_0 \sqrt{\frac{d\zeta}{n^\star}} \right).$$

Indeed, after multiplying both sides by $\sqrt{n_j}$, the only $n_j$-dependent term is the pooling-variance contribution $c_0 \sqrt{n_j Bd\zeta/N_{\mathbf{S}}}$, which is largest at $n_j = n^\star$. Then Proposition 10 applies. Its proof constructs a transfer event $\mathcal{E}_{\mathrm{transfer}}^{\mathrm{gs}}$ with $\mathbb{P}(\mathcal{E}_{\mathrm{transfer}}^{\mathrm{gs}}) \geq 1 - \kappa/2$ under the present good-event budget. On $\mathcal{E}_{\mathrm{safe}}^{\mathrm{gs}} \cap \mathcal{E}_{\mathrm{transfer}}^{\mathrm{gs}}$, which has probability at least $1 - \kappa$, for every $j \in \mathbf{S}$,

$$\|\hat{\theta}_j - \theta_j^\star\|_{\mathbf{\Sigma}_j} \lesssim \alpha \frac{B\delta_0}{\sqrt{n_j}} + \sqrt{\frac{Bd\zeta}{N_{\mathbf{S}}}} + \frac{\varepsilon B\lambda}{\sqrt{n^\star}}.$$

Squaring yields

$$\|\hat{\theta}_j - \theta_j^\star\|_{\mathbf{\Sigma}_j}^2 \lesssim \alpha^2 B^2 \frac{\delta_0^2}{n_j} + \frac{Bd\zeta}{N_{\mathbf{S}}} + \frac{\varepsilon^2 B^2 \lambda^2}{n^\star}.$$

Moreover, the displayed condition implies $\alpha B\delta_0 \lesssim \lambda = q\sqrt{d\zeta}$, so the heterogeneity term may be replaced, up to constants, by

$$\frac{1}{n_j} \min \left( \alpha^2 B^2 \delta_0^2, \ q^2 d\zeta \right).$$

Since $n_j \leq n^\star$, the pooling term satisfies

$$\frac{Bd\zeta}{N_{\mathbf{S}}} \leq \frac{n^\star}{n_j} \frac{Bd\zeta}{N_{\mathbf{S}}}.$$

Therefore, on $\mathcal{E}_{\mathrm{safe}}^{\mathrm{gs}} \cap \mathcal{E}_{\mathrm{transfer}}^{\mathrm{gs}}$,

$$\|\hat{\theta}_j - \theta_j^\star\|_{\mathbf{\Sigma}_j}^2 \lesssim \frac{n^\star}{n_j} \frac{Bd\zeta}{N_{\mathbf{S}}} + \frac{1}{n_j} \min \left( \alpha^2 B^2 \delta_0^2, \ q^2 d\zeta \right) + \frac{\varepsilon^2 B^2 \lambda^2}{n^\star}.$$

**Case 2: Proposition 10 Does Not Apply.** If the displayed condition fails, then

$$\frac{\lambda}{\sqrt{n^\star}} \leq C_{\mathrm{gs}} \left( \alpha B \frac{\delta_0}{\sqrt{n^\star}} + c_0 \sqrt{\frac{Bd\zeta}{N_{\mathbf{S}}}} + c_0 \sqrt{\frac{d\zeta}{n^\star}} \right).$$

Since $q \gtrsim 1$ is chosen large enough relative to $C_{\mathrm{gs}} c_0$, we absorb the last term into the left-hand side and obtain

$$\frac{\lambda}{2\sqrt{n^\star}} \lesssim \alpha B \frac{\delta_0}{\sqrt{n^\star}} + \sqrt{\frac{Bd\zeta}{N_{\mathbf{S}}}}.$$

Therefore at least one of the following alternatives must hold:

$$\frac{\lambda}{\sqrt{n^\star}} \lesssim \alpha B \frac{\delta_0}{\sqrt{n^\star}} \qquad \text{or} \qquad \frac{\lambda}{\sqrt{n^\star}} \lesssim \sqrt{\frac{Bd\zeta}{N_{\mathbf{S}}}}.$$

The first alternative implies, for every $j \in \mathbf{S}$,

$$q^2 \frac{d\zeta}{n_j} = \frac{\lambda^2}{n_j} \lesssim \alpha^2 B^2 \frac{\delta_0^2}{n_j}.$$

The second alternative implies

$$q^2 \frac{d\zeta}{n^\star} \lesssim \frac{Bd\zeta}{N_{\mathbf{S}}} \qquad \Longrightarrow \qquad q^2 \frac{d\zeta}{n_j} = \frac{n^\star}{n_j} q^2 \frac{d\zeta}{n^\star} \lesssim \frac{n^\star}{n_j} \frac{Bd\zeta}{N_{\mathbf{S}}}.$$

Hence, in either subcase, the safety rate obeys

$$q^2 \frac{d\zeta}{n_j} \lesssim \frac{n^\star}{n_j} \frac{Bd\zeta}{N_{\mathbf{S}}} + \frac{1}{n_j} \min \left( \alpha^2 B^2 \delta_0^2, \ q^2 d\zeta \right).$$

On the already-defined event $\mathcal{E}^{\text{gs}}_{\text{safe}}$, the personalization bound with $\lambda_j = \lambda/\sqrt{n_j}$ gives

$$\|\hat{\theta}_j - \theta_j^\star\|_{\Sigma_j} \lesssim \lambda_j + \sqrt{\frac{d\zeta}{n_j}} \lesssim q\sqrt{\frac{d\zeta}{n_j}},$$

so

$$\|\hat{\theta}_j - \theta_j^\star\|_{\Sigma_j}^2 \lesssim q^2 \frac{d\zeta}{n_j}.$$

Combining the last two displays yields

$$\|\hat{\theta}_j - \theta_j^\star\|_{\Sigma_j}^2 \lesssim \frac{n^\star}{n_j} \frac{Bd\zeta}{N_\mathbf{S}} + \frac{1}{n_j} \min\left(\alpha^2 B^2 \delta_0^2,\ q^2 d\zeta\right).$$

Adding the nonnegative outlier-robustness term $\varepsilon^2 B^2 \lambda^2 / n^\star$ gives the claimed bound in part (b). $\qquad\square$

## H. Details for Sample Complexity of Second Moment Concentration

We discuss in more detail the sample complexity of covariance comparability, initially covered in Section 4. We generically investigate the sample complexity required for the empirical second moment and the expected second moment to become comparable for a distribution $x \sim \mathcal{P}$.

Let $x_1, \ldots, x_n$ be $n$ i.i.d. samples drawn from $\mathcal{P}$. We define the empirical second moment matrix $\Sigma$ and the population second moment matrix $\bar{\Sigma}$ as:

$$\Sigma := \frac{1}{n} \sum_{i=1}^n x_i x_i^\top, \quad \text{and} \quad \bar{\Sigma} := \mathbb{E}_{x \sim \mathcal{P}}[xx^\top].$$

Our goal is to find a constant $\nu > 0$ such that, with high probability over the $n$ samples, the following spectral ordering holds:

$$\nu^{-1}\Sigma \preceq \bar{\Sigma} \preceq \nu\Sigma.$$

Furthermore, we aim to determine the sample complexity $n$ required to achieve $\nu = \tilde{\mathcal{O}}(1)$.

In Section 4.2, we demonstrated that for many standard distributions (which we categorize as **Type 1**), the sample complexity is $n = \tilde{\Omega}(d)$. We briefly summarize these before discussing the more general **Type 2** distributions.

**Type 1 Distributions.** For this class, if $n = \tilde{\Omega}(d)$, comparability holds with $\nu = \mathcal{O}(1)$. This class is characterized by properties discussed in (Oliveira, 2016):

Every coordinate of $\bar{\Sigma}^{-\frac{1}{2}}x$ has 9th moments bounded by $\mathcal{O}(1)$.

A notable subset includes strongly sub-Gaussian vectors, where $v^\top x$ is sub-Gaussian with proxy variance $cv^\top \bar{\Sigma}v$ for some $c = \mathcal{O}(1)$.

This encompasses Gaussian, uniform distributions, and cases with independently sub-Gaussian coordinates.

**Type 2 Distributions.** We now present the sample complexity for a more general sub-exponential class. We proceed under the assumption that the norm $\|x\|_2$ concentrates.

**Definition 3** (High-probability bound of $\|x\|_2$)**.** We assume there exists a parameter $U > 0$ such that for some absolute constant $c > 0$:

$$\mathbb{P}\left[\frac{\|x\|_2}{U} > t\right] \leq \exp(-ct).$$

Note that if $\|x\|_{\psi_1} \leq C_{\psi_1}$, we typically have $U = \tilde{\mathcal{O}}(\sqrt{d}C_{\psi_1})$. However, $U$ can be significantly smaller than $d$, depending on the *intrinsic dimension* (often related to $\text{tr}(\bar{\Sigma})$).

**Definition 4** (Minimum non-zero eigenvalue). Let $\gamma$ be the minimum non-zero eigenvalue of the population covariance $\bar{\Sigma}$.

Our main result for Type 2 distributions states that we achieve comparability with a constant factor $\nu = 6$ provided $n = \tilde{\Omega}(U^2/\gamma)$.

**Lemma 7** (Sample complexity of comparability). *Assume the high-probability bound on $\|x\|_2$ holds. If*

$$n \gtrsim \frac{U^2}{\gamma} \log^3\left(\frac{nU}{\gamma\kappa}\right),$$

*then with probability at least $1 - \kappa$,*

$$\frac{1}{6}\check{\Sigma} \preceq \Sigma \preceq 6\bar{\Sigma}.$$

This indicates that the minimum eigenvalue $\gamma$ affects the sample complexity requirement but does not degrade the constant factor in the spectral bound once the sample size is sufficient.

**Proof of Lemma 7** Without loss of generality, we assume $\bar{\Sigma}$ is full-rank. If not, the analysis restricts to the subspace spanned by the support of $\mathcal{P}$. We define a truncation radius

$$K := AU \log\left(\frac{nU}{\gamma\kappa}\right)$$

for a sufficiently large constant $A$. We define the truncated distribution $\check{\mathcal{P}}$ as the distribution of $x$ conditioned on the event $\mathcal{E}_{\text{trunc}} := \{\|x\|_2 \leq K\}$. Let $\check{\Sigma} := \mathbb{E}_{x\sim\check{\mathcal{P}}}[xx^\top]$.

Consider the event $\mathcal{E}_{\text{samples}} := \{\forall i \in [n], \|x_i\|_2 \leq K\}$. By the union bound and the definition of $U$:

$$\mathbb{P}(\mathcal{E}_{\text{samples}}^c) \leq \sum_{i=1}^n \mathbb{P}(\|x_i\|_2 > K) \leq n \exp\left(-c\frac{K}{U}\right) \leq \kappa,$$

provided $A$ is chosen large enough. Conditioned on $\mathcal{E}_{\text{samples}}$, the samples $x_1, \ldots, x_n$ can be treated as i.i.d. draws from the truncated distribution $\check{\mathcal{P}}$.

We apply the Matrix Bernstein inequality to the whitened truncated variables. Let $z_i := \check{\Sigma}^{-1/2}x_i$. Under truncation, we verify the operator norm bound:

$$\|\check{\Sigma}^{-1/2}x_i x_i^\top \check{\Sigma}^{-1/2}\|_2 = \|\check{\Sigma}^{-1/2}x_i\|_2^2 \leq \frac{1}{\lambda_{\min}(\check{\Sigma})}\|x_i\|_2^2 \leq \frac{2}{\gamma}K^2,$$

where we used the fact that $\check{\Sigma} \succeq \frac{1}{2}\bar{\Sigma} \succeq \frac{\gamma}{2}I$ (proven in Lemma 9 below). The Matrix Bernstein inequality implies that with high probability:

$$\|\check{\Sigma}^{-\frac{1}{2}}(\Sigma - \check{\Sigma})\check{\Sigma}^{-\frac{1}{2}}\| \leq C\sqrt{\frac{K^2 \log n}{n\gamma}} + C\frac{K^2 \log n}{n\gamma}.$$

Given the sample complexity condition $n \gtrsim \frac{K^2}{\gamma} \log n$, we can ensure the RHS is bounded by $1/2$. This implies:

$$\frac{1}{2}\check{\Sigma} \preceq \Sigma \preceq \frac{3}{2}\check{\Sigma}. \tag{14}$$

Using Lemma 9, we know that $\bar{\Sigma}$ and $\check{\Sigma}$ are close:

$$\bar{\Sigma} - \frac{\gamma}{2}I \preceq \check{\Sigma} \preceq 2\bar{\Sigma}.$$

Specifically, since $\bar{\Sigma} \succeq \gamma I$, the additive error implies multiplicative comparability (e.g., $\frac{1}{2}\bar{\Sigma} \preceq \check{\Sigma}$). Combining this with Equation (14), we obtain the desired result:

$$\frac{1}{6}\bar{\Sigma} \preceq \frac{1}{2}\check{\Sigma} \preceq \Sigma \preceq \frac{3}{2}\check{\Sigma} \preceq 3\bar{\Sigma}.$$

∎

We now present a lemma useful for establishing MSE bounds (Theorem 3).

**Lemma 8.** *Let $K$ be defined as in the proof of Lemma 7. With probability at least $1 - 2\kappa$,*

$$\bar{\boldsymbol{\Sigma}} \preceq c\left(\boldsymbol{\Sigma} + \frac{K^2}{n}\mathbf{I}\right)$$

*for some absolute constant $c > 1$.*

*Proof.* We again consider the high-probability event $\mathcal{E}_{\text{samples}}$ where all samples satisfy $\|x_i\|_2 \leq K$. Conditioned on this event, we essentially sample from $\tilde{\mathcal{P}}$. By applying a one-sided Matrix Bernstein inequality (or similar concentration bounds for bounded random vectors as in Lemma 7), we obtain:

$$\check{\boldsymbol{\Sigma}} \preceq c'\left(\boldsymbol{\Sigma} + \frac{K^2}{n}\mathbf{I}\right)$$

for some constant $c' > 1$. From Lemma 9, we have $\bar{\boldsymbol{\Sigma}} \preceq 2\check{\boldsymbol{\Sigma}}$. Combining these yields the result. $\qquad\square$

Finally, we provide the rigorous "peeling" argument that bounds the difference between the full and truncated population covariances.

**Lemma 9** (Truncation tail bound). *Let $\check{\boldsymbol{\Sigma}} = \mathbb{E}[xx^\top \mid \|x\|_2 \leq K]$, where $K$ is defined as in the proof of Lemma 7 (i.e., $K = AU\log\left(\frac{nU}{\gamma\kappa}\right)$). For a sufficiently large constant $A$,*

$$\|\bar{\boldsymbol{\Sigma}} - \mathbb{P}(\|x\|_2 \leq K)\check{\boldsymbol{\Sigma}}\|_2 \leq \frac{\gamma}{2}.$$

*Consequently, $\check{\boldsymbol{\Sigma}} \succeq \frac{1}{2}\bar{\boldsymbol{\Sigma}} \succeq \frac{\gamma}{2}\mathbf{I}$.*

*Proof.* Let $E$ be the indicator event $\{\|x\|_2 \leq K\}$. We can decompose the expectation:

$$\bar{\boldsymbol{\Sigma}} = \mathbb{E}[xx^\top] = \mathbb{P}(E)\mathbb{E}[xx^\top \mid E] + \mathbb{E}[xx^\top\mathbb{I}(E^c)] = \mathbb{P}(E)\check{\boldsymbol{\Sigma}} + \Delta.$$

We bound the residual matrix $\Delta = \mathbb{E}[xx^\top\mathbb{I}(\|x\|_2 > K)]$ using a peeling argument. Define shells $S_j := \{x : K+(j-1)U < \|x\|_2 \leq K + jU\}$ for $j \geq 1$.

$$\Delta \preceq \mathbb{E}[\|x\|_2^2\mathbb{I}(\|x\|_2 > K)]\mathbf{I}$$
$$= \sum_{j=1}^{\infty}\mathbb{E}[\|x\|_2^2\mathbb{I}(x \in S_j)]\mathbf{I}$$
$$\preceq \sum_{j=1}^{\infty}(K + jU)^2\mathbb{P}(\|x\|_2 > K + (j-1)U)\mathbf{I}.$$

Using the tail assumption $\mathbb{P}(\|x\|_2 > tU) \leq e^{-ct}$, we have:

$$\mathbb{P}(\|x\|_2 > K + (j-1)U) \leq \exp\left(-c\frac{K + (j-1)U}{U}\right) = e^{-cK/U}e^{-c(j-1)}.$$

Substituting this back:

$$\|\Delta\|_2 \leq e^{-cK/U}\sum_{j=1}^{\infty}(K + jU)^2e^{-c(j-1)}.$$

Since $K \gg U$, the term dominates for small $j$. Recall $K = AU\log(nU/\gamma)$, so $e^{-cK/U} = (\frac{\gamma}{nU})^{cA}$. For sufficiently large $A$, the pre-factor $(\frac{\gamma}{nU})^{cA}$ suppresses the polynomial terms in the sum, ensuring:

$$\|\Delta\|_2 \leq \frac{\gamma}{2}.$$

Thus, $\mathbb{P}(E)\check{\boldsymbol{\Sigma}} \succeq \bar{\boldsymbol{\Sigma}} - \frac{\gamma}{2}\mathbf{I} \succeq \frac{\gamma}{2}\mathbf{I}$. Since $\mathbb{P}(E) \leq 1$, this implies $\check{\boldsymbol{\Sigma}} \succeq \frac{\gamma}{2}\mathbf{I}$. $\qquad\square$

## I. Technical Lemmas

**Lemma 10.** *If $f : \mathbb{R}^d \to \mathbb{R}$ is convex, $\inf_{\boldsymbol{x} \in \mathbb{R}^d} f(\boldsymbol{x}) > -\infty$, and $R : \mathbb{R}^d \to \mathbb{R}$ is convex and L-Lipschitz with respect to a norm $\| \cdot \|$ for some $L \geq 0$, then*

$$g(\theta) := \inf_{x \in \mathbf{C}} \{ f(x) + R(\theta - x) \}$$

*is convex and L-Lipschitz with respect to $\| \cdot \|$.*

*Proof.* For any $\theta, \theta' \in \mathbf{C}$ and any $\varepsilon > 0$, there exists $x_0 \in \mathbf{C}$ such that

$$g(\theta') + \varepsilon > f(x_0) + R(\theta' - x_0).$$

Then we have

$$
\begin{aligned}
g(\theta) - g(\theta') &\leq f(x_0) + R(\theta - x_0) - (f(x_0) + R(\theta' - x_0) - \varepsilon) \\
&\leq R(\theta - x_0) - R(\theta' - x_0) + \varepsilon \\
&\leq L\|\theta - \theta'\| + \varepsilon.
\end{aligned}
$$

Since $\varepsilon$ is arbitrary, we get the wanted result.

$\square$

**Lemma 11** (Lemma F.2 from Duan and Wang 2023). *Let $f : \mathbb{R}^d \to \mathbb{R}$ be a convex function and $\boldsymbol{x}_0 = \operatorname{argmin}_{\boldsymbol{x}} f(\boldsymbol{x})$. Suppose there exist $\rho > 0$ and $r > 0$ such that $\nabla^2 f(\boldsymbol{x}) \succeq \rho \boldsymbol{I}, \forall \boldsymbol{x} \in B(\boldsymbol{x}_0, r)$. We have*

$$\|\boldsymbol{f}\|_2 \geq \rho \min \{\|\boldsymbol{x} - \boldsymbol{x}_0\|_2, r\}, \quad \forall \boldsymbol{f} \in \partial f(\boldsymbol{x}), \quad \boldsymbol{x} \in \mathbb{R}^d.$$

*If $g : \mathbb{R}^d \to \mathbb{R}$ is convex and $\lambda'$-Lipschitz for some $\lambda' < \rho r$, then $f(\boldsymbol{x}) + g(\boldsymbol{x})$ has a unique minimizer and it belongs to $B(\boldsymbol{x}_0, \lambda'/\rho)$.*

**Lemma 12** (Constrained Perturbation Bound). *Let $\mathbf{C} \subseteq \mathbb{R}^d$ be closed and convex. Let $f : \mathbb{R}^d \to \mathbb{R}$ be convex and differentiable, and let $x_0 \in \arg\min_{x \in \mathbf{C}} f(x)$. Suppose there exist $\rho > 0$ and $r > 0$ such that*

$$\nabla^2 f(x) \succeq \rho I_d, \qquad \forall x \in \mathbf{C} \cap B(x_0, r).$$

*Let $g : \mathbb{R}^d \to \mathbb{R}$ be convex and $\lambda'$-Lipschitz with respect to $\| \cdot \|_2$. If $\lambda' < \rho r$, then every minimizer*

$$\hat{x} \in \arg\min_{x \in \mathbf{C}} \{ f(x) + g(x) \}$$

*satisfies*

$$\|\hat{x} - x_0\|_2 \leq \frac{\lambda'}{\rho}.$$

*Proof.* Fix any minimizer $\hat{x} \in \arg\min_{x \in \mathbf{C}} \{ f(x) + g(x) \}$, and set $t := \|\hat{x} - x_0\|_2$. If $t = 0$, the claim is immediate. Let $u := (\hat{x} - x_0)/t$ and $s := \min\{t, r\}$. Since $\mathbf{C}$ is convex, the segment $x_0 + au, 0 \leq a \leq t$, lies in $\mathbf{C}$. The one-dimensional function $\phi(a) := f(x_0 + au)$ is convex on $[0, t]$, and satisfies $\phi''(a) \geq \rho$ for $0 \leq a \leq s$. Therefore $\phi'(t) - \phi'(0) \geq \rho s$, which gives

$$\langle \nabla f(\hat{x}) - \nabla f(x_0), \hat{x} - x_0 \rangle \geq \rho \min\{t, r\} t.$$

By the constrained first-order optimality condition for $\hat{x}$, there exist $v \in \partial g(\hat{x})$ and $n \in N_{\mathbf{C}}(\hat{x})$ such that

$$\nabla f(\hat{x}) + v + n = 0.$$

Since $x_0 \in \mathbf{C}$, $\langle n, x_0 - \hat{x} \rangle \leq 0$. Since $x_0$ minimizes the convex differentiable function $f$ over $\mathbf{C}$, we also have $\langle \nabla f(x_0), \hat{x} - x_0 \rangle \geq 0$. Thus

$$\langle \nabla f(\hat{x}) - \nabla f(x_0), \hat{x} - x_0 \rangle \leq \langle -v, \hat{x} - x_0 \rangle \leq \|v\|_2 t \leq \lambda' t,$$

where the last inequality uses the $\lambda'$-Lipschitz property of $g$. Combining the two displays and dividing by $t > 0$, we obtain

$$\rho \min\{t, r\} \leq \lambda'.$$

Because $\lambda' < \rho r$, the minimum cannot be $r$, and hence $t < r$. Therefore $\rho t \leq \lambda'$, proving $t \leq \lambda'/\rho$. $\square$

**Lemma 13** (Lemma F.3. from Duan and Wang 2023)**.** *Let $f$ be convex and differentiable. If $\lambda > \|\nabla f(x)\|_2$, then*

$$f(x) = f \star (\lambda \| \cdot \|_2)(x) \quad and \quad \underset{y \in \mathbb{R}^d}{\arg\min} \{f(y) + \lambda\|x - y\|_2\} = x.$$

*Proof.* Observe that

$$f(y) + \lambda\|x - y\|_2 \geq f(x) + \langle \nabla f(x), y - x \rangle + \lambda\|x - y\|_2$$
$$\geq f(x) + (\lambda - \|\nabla f(x)\|_2)\|x - y\|_2 > f(x).$$

$\square$

**Lemma 14** (Constrained Invariant Region)**.** *Let $f : \mathbb{R}^d \to \mathbb{R}$ be convex and differentiable, let $\mathbf{C} \subseteq \mathbb{R}^d$ be a convex set, and let $x \in \mathbf{C}$. If $\lambda > \|\nabla f(x)\|_2$, then*

$$f(x) = \inf_{y \in \mathbf{C}} \{f(y) + \lambda\|x - y\|_2\}, \qquad \arg\min_{y \in \mathbf{C}} \{f(y) + \lambda\|x - y\|_2\} = \{x\}.$$

*Proof.* Fix any $y \in \mathbf{C}$. By convexity and differentiability of $f$,

$$f(y) \geq f(x) + \langle \nabla f(x), y - x \rangle.$$

Therefore

$$f(y) + \lambda\|x - y\|_2 \geq f(x) + \langle \nabla f(x), y - x \rangle + \lambda\|x - y\|_2 \geq f(x) + (\lambda - \|\nabla f(x)\|_2)\|x - y\|_2.$$

If $y \neq x$, the last term is strictly larger than $f(x)$ because $\lambda > \|\nabla f(x)\|_2$. For $y = x$, the objective value is exactly $f(x)$. Hence $x$ is the unique minimizer over $\mathbf{C}$, and the constrained infimal convolution value at $x$ equals $f(x)$. $\square$

### I.1. Concentration Inequalities

**Lemma 15** (Corollary E.1 from Wang 2026)**.** *Let $\{x_i\}_{i=1}^n$ be i.i.d. random elements in a separable Hilbert space $\mathbb{H}$ with $\mathbf{\Sigma} = \mathbb{E}(x_i \otimes x_i)$ being trace class. Define $\hat{\mathbf{\Sigma}} = \frac{1}{n}\sum_{i=1}^n x_i \otimes x_i$. Choose any constant $\gamma \in (0, 1)$ and define an event $\mathcal{A} = \{(1 - \gamma)(\mathbf{\Sigma} + \lambda\mathbf{I}) \preceq \hat{\mathbf{\Sigma}} + \lambda\mathbf{I} \preceq (1 + \gamma)(\mathbf{\Sigma} + \lambda\mathbf{I})\}$.*

*1. If $\|x_i\|_{\mathbb{H}} \leq M$ holds almost surely for some constant $M$, then there exists a constant $C \geq 1$ determined by $\gamma$ such that $\mathbb{P}(\mathcal{A}) \geq 1 - \delta$ holds so long as $\delta \in (0, 1/14]$ and $\lambda \geq \frac{CM^2 \log(n/\delta)}{n}$.*

*2. If $\|\langle x_i, v \rangle\|_{\psi_2}^2 \leq \kappa\mathbb{E}|\langle x_i, v \rangle|^2$ holds for all $v \in \mathbb{H}$ and some constant $\kappa$, then there exists a constant $C$ determined by $\kappa$ such that $\mathbb{P}(\mathcal{A}) \geq 1 - \delta$ holds so long as $\delta \in (0, 1/e]$ and $\lambda \geq \frac{C\operatorname{Tr}(\mathbf{\Sigma})\log(1/\delta)}{\gamma^2 n}$.*

**Lemma 16** (Lemma E.1 from Wang 2026)**.** *Suppose that $x \in \mathbb{R}^d$ is a zero-mean random vector with $\|x\|_{\psi_2} \leq 1$. There exists a universal constant $c_0 > 0$ such that for any symmetric and positive semi-definite matrix $\mathbf{\Sigma} \in \mathbb{R}^{d \times d}$,*

$$\mathbb{P}\left(x^\top \mathbf{\Sigma} x \leq c_0 \operatorname{Tr}(\mathbf{\Sigma})t\right) \geq 1 - e^{-r(\mathbf{\Sigma})t}, \quad \forall t \geq 1$$

*Here $r(\mathbf{\Sigma}) = \operatorname{Tr}(\mathbf{\Sigma})/\|\mathbf{\Sigma}\|_2$ is the effective rank of $\mathbf{\Sigma}$.*

