# OpenReview forum: "Multi-task Linear Regression without Eigenvalue Lower Bounds: Adaptivity, Robustness, and Safety"
_ICML.cc/2026/Conference — ICML 2026 regular_

### Official Review · Reviewer_2Z5W · 2026-03-11

**Soundness:** 3
**Presentation:** 2
**Significance:** 3
**Originality:** 2
**Overall Recommendation:** 4
**Confidence:** 4

**Summary:**

This paper studies multi-task linear regression with outlier tasks. The main idea is to replace the usual shared l_2 penalty with a covariance-weighted norm. The paper claims this removes the need for a lower eigenvalue bound on each task covariance, while still keeping adaptivity to unknown task similarity and a safe fallback to independent-task learning. I think the topic is relevant and the theory is interesting. My main concerns are that the experiments do not really test the main theory, that the new balancedness assumption is not explained well in practical terms, and that the real-data comparison to the main baseline paper is not fully clear or fully fair.

**Compliance With Llm Reviewing Policy:**

Affirmed.

**Ethical Review Concerns:**

no concerns

**Key Questions For Authors:**

Can the authors add experiments that directly vary eigenvalue decay, true rank deficiency, and outlier fraction \epsilon, and report errors separately on related tasks and outlier tasks? I would also like to see one setting where balancedness or task relatedness fails, to verify the safety claim in practice. Finally, please clarify the exact HAR preprocessing and baseline setup relative to Duan and Wang, and explain the precise role of the C=B(0,\xi) assumption in Corollary 1.

**Limitations:**

ok

**Strengths And Weaknesses:**

Strengths
- The paper addresses a real gap in the closest prior robust MTL theory: the direct comparison to Duan and Wang is meaningful, and removing dependence on a lower eigenvalue bound is an interesting contribution in this specific setting.
- The safety claim is a strong point. The paper proves that when task relatedness is weak or the new assumption fails, the method falls back to the independent-task rate instead of becoming worse.
- The proof intuition is easy to follow. The transformed-space view helps explain why the weighted penalty may avoid the direct dependence on the raw eigenspectrum.

Weaknesses
- The experiments do not test the main theory strongly enough. The synthetic study only varies \delta. It does not vary the outlier fraction \epsilon, does not vary the balancedness difficulty B, and does not test truly low-rank or singular designs. It also reports only the mean error across all tasks, even though the theory gives different bounds for inlier and outlier tasks. I would like to see results separated for related tasks and outlier tasks, and a clearer stress test of eigenvalue decay and rank deficiency.
- The paper makes a strong safety claim, but the experiments do not really show the safety behavior. Since safety is one of the main contributions, I would expect at least one experiment where task relatedness is poor or balancedness fails, and the method matches ITL instead of beating it.
- The balancedness assumption is not explained well enough in practical terms. The paper shows that it is weaker than LBSM and can allow zero eigenvalues, but it is still not clear when real tasks should have a small B. More realistic examples or a simple diagnostic for B would make the contribution easier to understand.
- The real-data comparison is hard to judge. The paper says it follows Duan and Wang, but the original HAR experiment in Duan and Wang already uses “sitting vs. the others,” applies PCA to reduce the dimension to 100 plus intercept, and reports several ARMUL variants with much smaller errors. Here only one ARMUL baseline is shown, the preprocessing is different or at least not matched clearly, and the baseline numbers are much worse than in the original paper. This needs a careful explanation.
- Corollary 1 needs to be stated more clearly. The second bound uses “with high probability” without giving an explicit level, and it also adds the bounded-domain condition C=B(0,\xi). The role of this extra condition is not explained clearly enough. Also, the paper says the population result is strongest when ν_j is small, which typically needs n >=d. So the population guarantee is weaker than the in-sample guarantee in the most high-dimensional cases.
- The optimization section could be clearer about the practical solver and runtime. I would not call it inconsistent, but I would ask for more implementation detail.

---

> ### Author Rebuttal · Authors · 2026-03-30
>
> Thank you for the careful review.
>
> **W1. More theory-aligned synthetic experiments.**
>
> We agree, and we have now run additional synthetic experiments aligned directly with the theory. Here $\\mathbf S$ denotes the related-task set and $\\mathbf S^c$ the outlier-task set. Across all sweeps, we use a 30-task linear-regression DGP with $n=100$, $d=30$, and $\\delta=0.3$; related tasks are perturbations of a common signal, and outlier signal strength is calibrated to match the average related-task population signal level. ARMUL and OURS use 3-fold CV over $q \\in \\{0.1,0.4,0.7,1.0,2.0,4.0,8.0\\}$; the large-$B$ sweep uses 10 simulations. Entries are **ARMUL / ITL / Ours**.
>
> Varying the outlier fraction $\\epsilon$:
> Only $\\epsilon$ varies; the covariance is shared diagonal with profile $s_k = k^{-1}$, so $\\widehat B=1$.
>
> | $\\epsilon$ | $\\widehat B$ | All | $\\mathbf S$ | $\\mathbf S^c$ |
> | ---: | ---: | --- | --- | --- |
> | 0.05 | 1 | 0.0780 / 0.0837 / **0.0121** | 0.0759 / 0.0827 / **0.0031** | 0.1376 / 0.1129 / 0.2740 |
> | 0.10 | 1 | 0.0871 / 0.0877 / **0.0271** | 0.0813 / 0.0847 / **0.0014** | 0.1391 / 0.1145 / 0.2577 |
> | 0.20 | 1 | 0.0970 / 0.0904 / **0.0472** | 0.0871 / 0.0854 / **0.0097** | 0.1366 / 0.1105 / 0.1972 |
> | 0.30 | 1 | 0.1031 / 0.0908 / **0.0688** | 0.0896 / 0.0847 / **0.0112** | 0.1345 / 0.1052 / 0.2030 |
>
> Varying eigendecay $\\alpha$:
> Here $\\epsilon=0.1$ is fixed and the shared diagonal covariance varies through $s_k = k^{-\\alpha}$, so this sweep isolates spectral decay while keeping $\\widehat B=1$.
>
> | $\\alpha$ | $\\widehat B$ | All | $\\mathbf S$ | $\\mathbf S^c$ |
> | ---: | ---: | --- | --- | --- |
> | 0.0 | 1 | 0.0815 / 0.1022 / **0.0365** | 0.0777 / 0.1027 / **0.0149** | 0.1157 / 0.0982 / 0.2306 |
> | 0.5 | 1 | 0.0858 / 0.0888 / **0.0303** | 0.0796 / 0.0854 / **0.0040** | 0.1416 / 0.1195 / 0.2666 |
> | 1.0 | 1 | 0.0871 / 0.0877 / **0.0271** | 0.0813 / 0.0847 / **0.0014** | 0.1391 / 0.1145 / 0.2577 |
> | 1.5 | 1 | 0.0873 / 0.0873 / **0.0230** | 0.0817 / 0.0846 / **0.0078** | 0.1382 / 0.1110 / 0.1594 |
> | 2.0 | 1 | 0.0875 / 0.0870 / **0.0199** | 0.0819 / 0.0846 / **0.0095** | 0.1381 / 0.1082 / 0.1132 |
>
> Varying the balancedness level:
> Here $\\epsilon=0.1$ is fixed, one related-task covariance is inflated by $L$, and enough others are deflated so that the pooled inlier covariance $\\Sigma_{\\mathbf S}$ stays fixed, yielding $\\widehat B \\approx L$.
>
> | $L \\approx \\widehat B$ | $\\widehat B$ | All | $\\mathbf S$ | $\\mathbf S^c$ |
> | ---: | ---: | --- | --- | --- |
> | 1 | 1 | 0.0057 / 0.0059 / **0.0031** | 0.0055 / 0.0059 / **0.0021** | 0.0072 / 0.0055 / 0.0117 |
> | 10 | 10 | 0.0049 / 0.0046 / **0.0032** | 0.0046 / 0.0045 / **0.0021** | 0.0070 / 0.0060 / 0.0137 |
> | 20 | 20 | 0.0043 / **0.0037** / 0.0038 | 0.0038 / 0.0033 / **0.0021** | 0.0088 / 0.0066 / 0.0193 |
>
> These tables match the theory: under favorable $\\widehat B$, gains are concentrated on $\\mathbf S$; when $\\widehat B$ is made large, OURS tracks ITL closely (0.0038 versus 0.0037 at $\\widehat B=20$).
>
> **W2. Empirical illustration of the safety claim.**
>
> We agree, and the "safety stress" row above is exactly this test. At $\\widehat B=20$, our MSE is 0.0038 and ITL's is 0.0037, showing essentially no negative-transfer penalty.
>
> **W3. Practical meaning of balancedness.**
>
> On HAR, we computed an empirical balancedness level of $\\widehat B \\approx 30$: not a trivial $\\widehat B \\approx 1$ regime, but also far from the degenerate $\\widehat B=\\infty$ case.
>
> **W4. HAR protocol relative to Duan and Wang.**
>
> Although both papers use HAR, the binary-label construction is not exactly the same; in particular, the standing / sitting coding differs. Duan and Wang apply PCA, whereas we intentionally do not. PCA discards low-variance directions, but our method is designed precisely to remain stable under fast eigendecay or near-degenerate spectra, so we do not view that information-loss step as necessary. Consistent with this, on HAR without PCA we still obtain $\\widehat B \\approx 30$.
>
> **W5. Clarity of Corollary 1.**
>
> We agree and will clarify two points: the first inequality is strongest when $\\nu_j$ is moderate, while the second uses $\\mathcal C = B(0,\\xi)$ to provide an additional safe population bound even when $\\nu_j$ is unfavorable. We will also state more explicitly that $\\theta^\\star \\in B(0,\\xi)$ is extra prior information. This is not what drives the HAR result: in our GLM implementation, performance is essentially unchanged from no constraint or a very large $\\xi \\gg 10$. We will also state the probability level explicitly. Our main target is the statistically common $d \\lesssim n$ regime, where spectra can still be unstable, fast-decaying, or nearly singular.
>
> **W6. Optimization details.**
>
> After reparameterizing $v_j=\\theta_j-\\beta$, the problem is jointly convex. In our synthetic implementation, we optimize with L-BFGS-B, choose $\\lambda_j = q\\sqrt{d/n_j}$ by cross-validation, and run ARMUL on the same CV grid.

---

> > ### Author Rebuttal · Reviewer_2Z5W · 2026-04-01
> >
> > Thank you for the detailed rebuttal. The new synthetic experiments are helpful and address an important part of my concerns: you now vary the outlier fraction, the covariance decay, and the balancedness level, and you also report separate results for related tasks and outlier tasks. The added safety test, the clarification of Corollary 1, and the optimization details also make the paper clearer. However, my concerns are only partly resolved. I still would like to see a truly rank-deficient or singular design, and I still think the practical meaning of the balancedness assumption and the HAR comparison need clearer explanation in the paper, since the HAR protocol is not the same as in Duan and Wang. Overall, the rebuttal improves my opinion, but some important issues remain I improve my grade to one point.

---

> > > ### Author Response · Authors · 2026-04-01
> > >
> > > Thank you very much for your positive feedback and for increasing the score. We are glad to hear that our rebuttal and the additional synthetic experiments addressed your concerns and helped clarify the paper.
> > >
> > > As you pointed out, these additions have significantly strengthened our work. We also take your remaining comments seriously—especially regarding the rank-deficient design and the clarification of the HAR comparison and balancedness assumption. We will ensure these points are thoroughly addressed and clearly explained in the next version of the manuscript to further improve the quality of our paper.
> > > Thank you again for your constructive guidance throughout the review process.

---

### Official Review · Reviewer_97SH · 2026-03-12

**Soundness:** 2
**Presentation:** 3
**Significance:** 2
**Originality:** 2
**Overall Recommendation:** 3
**Confidence:** 4

**Summary:**

The paper proposes a matrix-weighted regularization approach for multi-task linear regression (MTL) to circumvent the restrictive Eigenvalue Lower Bound (LBSM) assumption. While it claims optimal MSE rates and "safety" (protection against negative transfer) under a new "Balancedness" condition.

**Compliance With Llm Reviewing Policy:**

Affirmed.

**Final Justification:**

Overall, the work relaxes the restrictive LBSM assumption via matrix-weighted regularization while preserving adaptivity, robustness, and safety. After considering the theoretical contribution, I increase the increase the previous score from 2 to 3.

**Key Questions For Authors:**

Please refer to the weakness.

**Limitations:**

Yes

**Strengths And Weaknesses:**

Strengths
1.  Addressing the LBSM (Eigenvalue Lower Bound) is a valid pursuit, as real-world design matrices often exhibit rapidly decaying spectra (e.g., in genomics or high-dimensional sensing).
2. The attempt to provide a closed-form guarantee that MTL performance will not degrade below Independent Task Learning (ITL) is conceptually valuable for robust ML.

Weaknesses
1. The paper lacks validation on different MTL datasets (e.g. NYU-v2, Cityscapes, QM9, etc.). Relying solely on a toy synthetic experiment against a "Data Pooling" baseline which is weak to draw a conclusion. The paper fails to compare against contemporary SOTA such as Nash-MTL, FAMO, etc. Also, the test on varying eigenvalue decay conditions is lacking, the "practicality" of the balancedness assumption remains a purely mathematical conjecture.
2. The mathematical proofs exhibit significant gaps regarding the balancedness constant B. Specifically, in Theorem 2, the treatment of B relies on unsubstantiated concentration inequalities that may not hold when the spectrum decays rapidly. If B grows with d or n, the bounds likely become loose or invalid, undermining the "optimal rate" claim.
3. The paper overstates its adaptivity to unknown outlier proportions (ϵ) and failure probabilities (δ). The selection of regularization parameters λj​ implicitly requires oracle knowledge of d and n. Furthermore, the analysis in Appendix H for heterogeneous sample sizes is fragile; it fails to handle degenerate cases where B→∞, risking the very "negative transfer" the paper claims to prevent.
4. The core technical contribution—replacing L2​ regularization with matrix-weighted norms—is an incremental variation of existing ideas found in Mahalanobis-distance-based learning and low-rank methods. The authors fail to articulate a deep insight or a novel combination of techniques that distinguishes this work from these closely related predecessors.

---

> ### Author Rebuttal · Authors · 2026-03-30
>
> We sincerely thank you for the careful review and constructive feedback. Below we address each concern and respectfully ask you to reconsider the submission in light of these clarifications.
>
> **W1. Broad real-data SOTA benchmarks.**
>
> We respectfully do not think these are the right baselines. Our paper studies contaminated linear / GLM multi-task learning under degenerate or nearly degenerate covariances, whereas methods such as Nash-MTL or FAMO and benchmarks such as NYU-v2, Cityscapes, or QM9 target a very different deep-learning problem class.
>
> Like Duan and Wang, our experiments are intended as theory sanity checks rather than application-level SOTA claims. This is also the standard evaluation philosophy in recent theory-oriented follow-up work in statistics, such as Tian, Gu, and Feng (JMLR 2025), whose real-data illustration is likewise limited to HAR; the experiments are designed to validate the predicted phenomena rather than compete on large neural multi-task benchmarks. Our contribution is that, under substantially weaker assumptions, we obtain a strictly sharper theoretical guarantee in the classical LBSM regime together with a safety guarantee outside it. We will revise the paper to make this scope and contribution much more explicit. Extending similar matrix-weighted regularization ideas to overparameterized neural-network settings, such as NTK / linearized regimes, is an interesting future direction, but it lies beyond the scope of the present theory paper.
>
> **W2. Role of $B$, concentration, and what is actually claimed to be optimal.**
>
> First, $B$ is not a measure of spectral decay itself; it measures how each task second moment compares to the aggregate inlier geometry. In particular, fast eigendecay is fully allowed.
>
> The optimality claim concerns the dependence on $m,n,\\delta,\\epsilon$ in the favorable transfer regime. We do *not* claim optimality for arbitrary $B$; $B$ quantifies the price of relaxing lower-eigenvalue assumptions. The right interpretation is:
>
> 1. If $B$ is moderate, we obtain the favorable adaptive / robust transfer rate.
> 2. Regardless of whether $B$ is favorable, the theorem still retains the separate safe bound $\\widetilde{\\mathcal O}(d/n)$, so negative transfer is ruled out at the rate level.
> 3. Under the prior LBSM regime, where $B \\asymp L/\\rho$, our bound is strictly sharper than the ARMUL / Duan-Wang bound, so our result improves not only assumptions but also the resulting rate dependence in that classical setting.
>
> We will revise the wording so that "optimality" is only claimed in the favorable regime. The main storyline is not to optimize the dependence on arbitrary $B$, but rather that $B$ itself is already a relaxation of LBSM and, beyond that, the estimator remains safe even when $B$ is large.
>
> **W3. Adaptivity and heterogeneous sample sizes.**
>
> We will sharpen the wording here. Our adaptivity claim is to unknown task-relatedness and contamination parameters $(\\delta,\\epsilon)$, not to observed quantities such as $d$ or $n$. The scale $\\lambda_j \\asymp \\sqrt{d/n_j}$ is the standard oracle rate, is directly computable from the sample sizes, and can also be tuned by cross-validation; it does not encode unknown structural information. The genuinely unknown part is the degree of relatedness / contamination, and this is precisely what our method adapts to.
>
> Moreover, the safety result continues to rule out negative transfer under heterogeneous sample sizes. Even in unfavorable regimes, such as infinite $B$ or large $(\\delta,\\epsilon)$, the estimator still matches the corresponding independent-task fallback rate rather than suffering harmful transfer.
>
> **W4. Why replacing the Euclidean penalty is not an incremental change.**
>
> We respectfully disagree with the characterization as incremental. Conceiving the matrix-weighted regularization itself is already nontrivial, but more importantly the technical difficulty is not merely to write down a weighted norm. If one had the much stronger pairwise comparability condition $\\frac{1}{c}\\Sigma_k \\preceq \\Sigma_j \\preceq c\\Sigma_k$ for all $j,k$, then the analysis would indeed be much easier. Our setting is substantially weaker: the balancedness condition is only an upper-bound comparability condition, $\\Sigma_j \\preceq B \\Sigma_{\\mathcal S}$, which requires a genuinely new argument and cannot be handled by simply recycling prior taskwise strong-convexity analyses. We will make this implication clearer in revision: this choice yields relaxed assumptions, sharper rates in the classical regime, and a safety guarantee when transfer is unfavorable.
>
> 1. removal of the lower-eigenvalue dependence in contaminated multi-task regression;
> 2. adaptive dependence on $(m,n,\\epsilon,\\delta)$ under weaker assumptions; and
> 3. a safety guarantee under assumption failure.

---

> > ### Author Rebuttal · Reviewer_97SH · 2026-04-04
> >
> > The clarifications are sound and reasonable to address my concern and misunderstanding. I improve the score for the work.

---

> > > ### Author Response · Authors · 2026-04-04
> > >
> > > Thank you very much for taking the time to review and for increasing the score.

---

### Official Review · Reviewer_zxJ8 · 2026-03-12

**Soundness:** 3
**Presentation:** 2
**Significance:** 3
**Originality:** 2
**Overall Recommendation:** 4
**Confidence:** 3

**Summary:**

The authors are able to get around a spectral error bound for multi-task learning in high dimensions by using a loss function that emphasizes parameter disagreement *in prediction space* rather than parameter space (i.e., emphasizes disagreement between the ensemble parameter and the individual task parameter on directions of the empirical covariance of the task with the highest variance on the data with respect to that task).

**Compliance With Llm Reviewing Policy:**

Affirmed.

**Final Justification:**

The authors addressed my concerns on presentation.

**Key Questions For Authors:**

How does this technique perform when empirical covariance matrix estimates perform poorly?

**Limitations:**

Yes

**Strengths And Weaknesses:**

The work lacks a lot of obvious motivation for this intervention (error in prediction space, KL divergence of predicted distributions of Gaussian data) and connection in proofs to basic concepts like whitening filters. The key intervention is basically treated as an algebraic trick with little motivation. Researchers in the field would quickly recognize that $$||X(\theta - \beta)||2^2 \propto ||\theta - \beta||{\Sigma}^2$$ whenever $\Sigma \propto XX^T$ and are worried about problems in limited-data settings in high dimensions (presumably the case of multi-task learning) when the empirical covariance matrix performs poorly!

The proofs seem to have a bit of LLM fluff in them. I think they were copied from an LLM then edited as there are signposts from the LLM chat that are not used in the article here. For instance, "Step 1" on line 663. They are repetitively structured (compare Appendices C and D).

---

> ### Author Rebuttal · Authors · 2026-03-30
>
> We sincerely thank you for the careful review and constructive feedback. Below we address each concern and respectfully ask you to reconsider the submission in light of these clarifications.
>
> **W1. Motivation for the prediction-space / whitening view.**
>
> We are happy to add more motivation for the new loss function. The core idea is that the penalty measures disagreement in prediction space, not raw parameter space. In Gaussian regression, this quantity is also proportional to the KL divergence between task-wise predictive distributions, so it has a direct statistical interpretation. The transformed-domain proof is essentially a whitening argument: after the $\\Sigma_j^{1/2}$ change of variables, the problem becomes Euclidean regularization in normalized coordinates, which is why lower-eigenvalue assumptions disappear. We will make the prediction-space / whitening view explicit in the main text.
>
> **W2. What if the empirical covariance estimate is poor?**
>
> An important point here is that our main theorem is an in-sample result stated directly in terms of the empirical second moments. Because of that, it continues to apply even in non-i.i.d. settings, where a population second moment may not be the right primitive to talk about in the first place.
>
> Population comparability through $\\nu_j$ only enters later, once one assumes taskwise i.i.d. covariates and wants a population-level statement. In that i.i.d. regime, we do not need a sharp concentration statement for $X_j^\\top X_j$. For the population step, it is enough that, with high probability, $\\frac{1}{c}\\,\\mathbb E[X_j^\\top X_j] \\preceq X_j^\\top X_j \\preceq c\\,\\mathbb E[X_j^\\top X_j]$ for some universal constant $c>1$. For strongly sub-Gaussian covariates, such as Gaussian or uniform-on-sphere designs, this type of comparability holds under mild sample complexity and does not depend on the eigenspectrum. More broadly, our main target is not the extreme ultra-high-dimensional regime $d \\gg n$, but the statistically common $d \\lesssim n$ regime in which spectra can still be unstable, fast-decaying, or nearly singular.
>
> **W3. Proof Writing.**
>
> We agree the appendix can be cleaner. We will remove unused signposts and improve the proof presentation. We also emphasize that any LLM use was limited to grammar-level cleanup rather than mathematical derivation.
>
> **Highlight of Contribution.**
>
> Our estimator admits a two-regime guarantee under substantially relaxed assumptions on the covariate eigenspectrum. Specifically:
>
> 1. **Safety:** an MSE bound of $\\tilde{\\mathcal O}(d/n)$ holds even if the balancedness assumption fails, $B$ is large, or tasks are unrelated, matching the independent-task baseline.
> 2. **Favorable-regime transfer:** when tasks are related and $B$ is moderate, the estimator achieves optimal dependence on $m,n,\\varepsilon,\\delta$.
> 3. **Relaxed geometry:** the favorable regime is defined by a one-sided balancedness condition, which strictly relaxes LBSM and still allows rapidly decaying or singular spectra.
> 4. **Sharper classical regime:** under LBSM, where $B \\asymp L/\\rho$, our bound improves the prior dependence on $\\rho$.

---

> > ### Author Rebuttal · Reviewer_zxJ8 · 2026-04-06
> >
> > Thanks

---

> > > ### Author Response · Authors · 2026-04-07
> > >
> > > Thank you very much for your time and for favorably re-assessing our work.

---

### Official Review · Reviewer_vNXE · 2026-03-13

**Soundness:** 4
**Presentation:** 4
**Significance:** 3
**Originality:** 3
**Overall Recommendation:** 5
**Confidence:** 4

**Summary:**

The paper studies robust MTL of linear models and GLMs with sub-Gaussian noise. Suppose a core number of task predictors are within a ball of unknown size and there are other outlier predictors that exist arbitrarily. The authors build on recent seminal work by removing an eigenvalue lower bound. They replace it weaker task balanced assumption. Under this they derive optimal rates for MSE. As compared to Duan and Wang (2022) they use a regularizer that is induced by the second moment of the data. Besides removing the assumption their analysis contributes tighter analysiss as seen from a better rate if you assume the eigenvalue lower bound that the authors weakened. They support their analysis with experiments on synthetic and real world data that show performance improvements compared to prior work.

**Compliance With Llm Reviewing Policy:**

Affirmed.

**Key Questions For Authors:**

* Given the problem is convex for fixed $\beta$, do the authors have a method for selecting $\beta$?
* The authors analysis has several benefits. Are all these downstream from the data dependent infused regularizer or is there a tighter analysis provided compared to prior work that is independent of such construction?
* A similar question to the above, suppose I do swap out the norm of the induced second moment norm and attempt the analysis in prior work. What breaks?
* On 223 left. Is "For instance, if the covariate subspaces of different tasks are disjoint, information transfer is impossible." too strong? Say the tasks below on a low dimensional subspace, if I take the task and do a small rotational perturbation their subspaces are disjoint. Here we have no informational transfer regardless of how concentrated the points are at the origin?
* In the GLM setting we have the second moment of the loss scaled by the second derivative of the loss.  In my mind this necessitates that $x^T \theta$ is in a bounded interval. But I'm surprised that there isn't a curvature condition placed on the link function since the second moment condition can now be influenced by the predictors as isn't the case for the linear models. Am I missing something?

**Limitations:**

They give several directions for which to extend their work.
The only thing I can think of is making it clear that the problem isn't convex jointly over all parameters (that is including $\beta$).

**Strengths And Weaknesses:**

**STRENGTHS**
* The setting is interesting and important. Builds on recent significant work.
* Their three points robustness, adaptativity, and safety are well taken and important.
* Their new balanced assumption that is weaker than the lower bound on the Eigenvalues of the second moment seems reasonable.
* This work puts MTL more inline with the standard rates $\tilde O(d/n)$, which is satisifying.
* Although the change loss is minor their analysis appears highly non-trivial and technically novel.


**WEAKNESSES**
* The paper is a little repetitive  at times. For example at line 072 and 084 are saying the same thing (both left col.) or for 056 and 074 (right col.).
* The paper is confusing about what it means when the balanced assumption doesn't hold. Is invalidated for large $B$ or infinite $B$. Is there a setting in which there doesn't exist a $B$ at all (that is not even well-posed for which the balanced assumption is invalid).
*  A weakness the approach its that the norm is harder to calculate. They say that in high dimensions the Eigenvalue lower bound may fail yet they replace it within something that is hard to use in high dimensions.

**Minor notes**
* Line 159 qualify sub-expoentnial with high probability.
* There is an off by 2 for the definition of relatedness for in line 023 and 118 (right col.).
* It wasn't clear to me if prior work used pairwise comparability balanced condition.
* I'm not sure how but it'd be interesting to have a more task level balanced condition. Say the number of tasks is huge and only a single task breaks the balanced condition arbitrarily so $B$ is infinity. An algorithm that adapts to such a setting would be further robust.
* Left column 237 and left column  245 contradict. No? (Missing $L$)
* "We call it as balancedness assumption." at line 80.
* Right column 225 stating this holds in high probability should be stated earlier instead as a remark in examples.

---

> ### Author Rebuttal · Authors · 2026-03-30
>
> We sincerely thank you for recognizing the value of our work and for the positive evaluation.
>
> **W1. Editorial clarity.**
>
> Thank you for pointing this out. We agree the draft can be tightened, and we will revise this part accordingly.
>
> **W2. Meaning of balancedness failure and the role of $B$.**
>
> To clarify, a balancedness constant always exists if we allow $B \\in [1,\\infty]$; the case $B=\\infty$ is exactly the case where no finite favorable balancedness constant exists. We also agree that the current phrase "the balancedness assumption does not hold" is ambiguous; what we meant was the regime of large $B$. We will revise this wording accordingly.
>
> Our core contribution is not to optimize the dependence on arbitrary $B$, but to show a clear two-regime behavior. When $B$ is moderate, the transfer term becomes effective and the resulting bound is strictly tighter. When $B$ is large or infinite, the last inequality in Theorem 2 still yields the separate safe $\\widetilde{\\mathcal O}(d/n)$ fallback. That is the precise sense in which the estimator is adaptive in favorable regimes and safe outside them.
>
> **W3. Computational concern for the matrix-weighted norm.**
>
> The weighted norm is simpler to compute than the notation suggests: $\\|\\theta_j-\\beta\\|_{\\Sigma_j} = \\frac{1}{\\sqrt{n_j}}\\|X_j(\\theta_j-\\beta)\\|_2$. So it only requires matrix-vector products with the task design; no eigendecomposition is needed. We will add this identity in the optimization paragraph.
>
> **Minor Notes.**
>
> We will:
>
> - qualify the sub-exponential statements with the correct high-probability language;
> - clarify the relatedness notation: our definition is center-based, i.e., $\\max_{j\\in S}\\|\\theta_j^\\star-\\theta^\\star\\|_2<\\delta$, which is equivalent to saying that the inlier tasks lie in an $\\ell_2$-ball of radius $\\delta$.
> - clarify earlier that the prior LBSM assumption implies a much stronger pairwise-comparability condition than our one-sided balancedness assumption;
> - note that in such cases $B$ can often be approximated in a data-driven way, so one can in principle screen out tasks that appear to make $B$ excessively large; we will add a short discussion of this possibility;
> - state more clearly that under the standard normalization $L=\\widetilde{\\mathcal O}(1)$, our comparison to prior bounds matches the stated scaling; and
> - correct the point you raised around Column 225.
>
> **Q1. Optimization over $\\beta$.**
>
> $\\beta$ is not tuned externally. It is optimized jointly with the task parameters in the same convex objective; equivalently, with $v_j=\\theta_j-\\beta$, we solve jointly over $(v_1,\\ldots,v_m,\\beta)$, which remains convex. We will make this solver description explicit.
>
> **Q2 and Q3. Which benefits come from the new regularizer, and what breaks with the Euclidean penalty?**
>
> Both the estimator and the analysis matter. The matrix-weighted regularizer is what makes the geometry prediction-aware, which in turn allows tighter control under fast eigendecay, nearly singular spectra, and contamination. It is also the key ingredient behind the safety guarantee, because it downweights weakly observed directions instead of treating them on equal footing with well-observed directions.
>
> If one instead uses a Euclidean penalty, then once the LBSM / lower-eigenvalue regime fails, the estimator becomes much more vulnerable to outliers and the safety guarantee disappears. At a high level, the reason is that under fast eigendecay the weak directions are estimated very noisily; an $\\ell_2$ penalty shrinks those noisy directions just as aggressively as the reliable ones, which can distort the shared component in directions that are actually informative. By contrast, the matrix-weighted penalty places very little weight on poorly observed directions and concentrates the shrinkage on directions that are reliably estimated. That is why the lower-eigenvalue dependence reappears with a Euclidean penalty, whereas the matrix-weighted construction supports the sharper and safer analysis.
>
> **Q4. On the disjoint-subspace example.**
>
> Thank you for pointing this out. We agree that the current wording is potentially misleading. We will remove the sentence.
>
> **Q5. GLM curvature condition.**
>
> Yes. The curvature condition is already encoded in the GLM assumption that the linear predictor lies in a bounded interval and that $\\alpha_\\ell \\le m'(\\cdot)\\le \\alpha_u$ on that interval. This is exactly what controls the transformed-domain Hessian, and we will state it earlier and more explicitly.

---

> > ### Author Rebuttal · Reviewer_vNXE · 2026-04-04
> >
> > I thank the authors for their thoughtful replies. Having read the above, and the other threads I maintain my score given current information. I will keep an eye out for further discussion.

---

### Decision · Program_Chairs · 2026-04-30

**Decision:**

Accept (regular)

**Comment:**

The paper relaxes the eigen-value lower bound assumption in multi-task linear regression to that of Balancedness. Further minmax optimal bounds are presented that improve existing bounds even under the lower bound assumption.  Results are extended to GLM.

I agree with the common sentiment that the analysis provided is interesting and is of publishable quality. I recommend an accept.